# Constrained Linear Thompson Sampling

**Aditya Gangrade**
Boston University
gangrade@bu.edu

**Venkatesh Saligrama**
Boston University
srv@bu.edu

## Abstract

We study safe linear bandits (SLBs), where an agent selects actions from a convex set to maximize an unknown linear objective subject to unknown linear constraints in each round. Existing methods for SLBs provide strong regret guarantees, but require solving expensive optimization problems. To address this, we propose Constrained Linear Thompson Sampling (COLTS), a sampling-based framework that selects actions by solving perturbed linear programs, which significantly reduces computational costs while matching the regret and risk of prior methods. We develop two main variants: S-COLTS, which ensures zero risk and $\widetilde{O}(\sqrt{d^3 T})$ regret given a safe action, and R-COLTS, which achieves $\widetilde{O}(\sqrt{d^3 T})$ regret and risk with no instance information. In simulations, these methods match or outperform state of the art SLB approaches while substantially improving scalability. On the technical front, we introduce a novel coupled noise design that ensures frequent 'local optimism' about the true optimum, and a scaling-based analysis to handle the per-round variability of constraints.

## 1 Introduction

Stochastic bandit problems are a fundamental model for optimising unknown objectives through repeated trials. While single-objective bandit theory is well-developed, real-world learners must also deal with *unknown constraints* at every round of interaction. For instance, in *dose-finding* [AKR21], *micro-grid control* [FLZY22], and *fair recommendation* [Cho+24], a learner must choose actions that maximise reward while never crossing unknown toxicity, voltage, or exposure limits (see §B).

The *safe linear bandit* (SLB) problem models these scenarios in a linear programming (LP) setting: a learner selects actions $\{a_t\}$ from a convex domain $\mathcal{A}$ to optimize an unknown objective vector $\theta_* \in \mathbb{R}^d$ subject to unknown constraints of the form $\Phi_* a \le \alpha$, where $\Phi_* \in \mathbb{R}^{m \times d}$. After each action, the learner observes noisy feedback of the objective $\theta_*^\top a + \text{noise}$ and the constraints $\Phi_* a + \text{noise}$, thus acquiring information to guide future actions. Performance in SLBs is measured via the

$$\text{\textit{regret,}} \ \mathbf{R}_T := \sum_{t \le T} \left( \theta_*^\top (a_* - a_t) \right)_+, \quad \text{and} \quad \text{\textit{risk,}} \ \mathbf{S}_T := \sum_{t \le T} \left( \max_i \left( \Phi_* a_t - \alpha \right)^i \right)_+, \qquad (1)$$

where $a_*$ is the optimal action under the true (but unknown) constraints, and $(\cdot)_+ := \max(\cdot, 0)$. There are two main notions of safety in SLBs:

• *Hard constraint enforcement*, which requires that with high probability, $\mathbf{S}_T = 0$ for all $T$. This is only achievable if the learner has prior access to a *known safe action* $a_{\mathsf{safe}}$.

• *Soft constraint enforcement*, which requires $\mathbf{S}_T = o(T)$ with high probability (whp). This is a weaker requirement, but does not need prior information.

A series of recent work [e.g. GCS24; PGB24; AAT19; MAAT21] offers OFUL-style algorithms for SLBs with strong theoretical guarantees. However, these often require the solution of nontrivial optimisation problems (second-order conic programs, and sometimes NP-hard problems) in each round. Our motivation lies in improving this computational cost.

39th Conference on Neural Information Processing Systems (NeurIPS 2025).

Table 1: COMPARISON OF SLB METHODS. 'Known $a_{\mathsf{safe}}$' means that the method requires an action known a priori to be safe. $\Delta(a) := \theta_*^\top(a_* - a)$ is the reward gap of an action $a$, and $\Gamma(a) := \min_i(\alpha - \Phi_* a)_+^i$ is its safety margin. $\mathcal{R}(a) := 1 + (\Delta(a)/\Gamma(a))$ if $\Gamma(a) > 0$, and $\infty$ otherwise. LP is the computation needed to optimize a linear objective with $m$ linear constraints over $\mathcal{A}$ to constant approximation. SOCP is the same with $m$ second-order conic constraints. We write 'NP-hard' if implementing the method needs a solver for an NP-hard problem. OPT-PESS refers to most frequentist hard enforcement methods discussed in §1.1, which have similar costs and bounds; SAFE-LTS is due to [MAAT21]; DOSS and the lower bound are due to [GCS24].

| Algorithm | Assumptions | Regret | Risk | Compute at $t$ |
|---|---|---|---|---|
| OPT-PESS | Known $a_{\mathsf{safe}}$ | $\mathcal{R}(a_{\mathsf{safe}}) \cdot \widetilde{O}(\sqrt{d^2 T})$ | 0 | NP-hard |
| Relaxed OPT-PESS | Known $a_{\mathsf{safe}}$ | $\mathcal{R}(a_{\mathsf{safe}}) \cdot \widetilde{O}(\sqrt{d^3 T})$ | 0 | $d \cdot \text{SOCP} \cdot \log(t)$ |
| SAFE-LTS | Known $a_{\mathsf{safe}}$ | $\mathcal{R}(a_{\mathsf{safe}}) \cdot \widetilde{O}(\sqrt{d^3 T})$ | 0 | $\text{SOCP} \cdot \log(t)$ |
| **S-COLTS** | Known $a_{\mathsf{safe}}$ | $\mathcal{R}(a_{\mathsf{safe}}) \cdot \widetilde{O}(\sqrt{d^3 T})$ | 0 | $\text{LP} \cdot \log(t)$ |
| DOSS | Feasibility | $\widetilde{O}(\sqrt{d^2 T})$ | $\widetilde{O}(\sqrt{d^2 T})$ | NP-hard |
| **R-COLTS** | Feasibility | $\widetilde{O}(\sqrt{d^3 T})$ | $\widetilde{O}(\sqrt{d^3 T})$ | $\text{LP} \cdot \log^2(t)$ |
| LOWER-BOUND | Feasibility | $\max(\mathbf{R}_T, \mathbf{S}_T) = \Omega(\sqrt{T})$, *no matter the instance*; | | |

**Contributions.** We introduce a *sampling-based* approach, *COnstrained Linear Thompson Sampling* (COLTS), which adds carefully chosen noise to estimates of both the objective and constraint parameters, and selects actions according to this perturbed program. This allows us to maintain the same order of regret and risk bounds as prior methods, while substantially reducing the complexity of each round. However, just perturbing the program as above does not directly yield good actions, since the perturbed program may be infeasible, or its optimum may be unsafe. We therefore develop two augmentations of COLTS, which address the SLB problem under distinct regimes:

• **S-COLTS** assumes a *given safe action* $a_{\mathsf{safe}}$. Actions are picked by first solving a perturbed LP (while ensuring that $a_{\mathsf{safe}}$ is feasible), and then scaling its optimum towards $a_{\mathsf{safe}}$ to ensure safety. This yields *zero* risk, and regret $\mathcal{R}(a_{\mathsf{safe}}) \cdot \widetilde{O}(\sqrt{d^3 T})$ (see §2, or Table 1 for definition of $\mathcal{R}(a)$).

• **R-COLTS** requires only *feasibility* of the true problem, and operates by sampling $O(\log T)$ perturbed programs, and setting $a_t$ to be the optimiser of the one with largest value. This resampling directly yields optimism, leading to instance-independent $\widetilde{O}(\sqrt{d^3 T})$ regret and risk bounds. We additionally argue that under Slater's condition, and with extra exploration, a similar regret and risk guarantee follows without resampling, and so solving only one optimisation per round.

Table 1 summarizes our results in comparison to prior work. We highlight that our results match previously attainable regret and risk bounds with only $O(\text{LP})$ computation per round. This yields the first efficient method for soft enforcement, and significantly speeds up hard constraint enforcement. A simulation study (§6,§J) further validates these claims. Contextual extensions are discussed in §E.

**Technical Innovations.** The random perturbations in our approach cause two challenges that break existing analyses of linear TS: (i) the feasible region fluctuates at each round; and (ii) the true optimum $a_*$ can become infeasible under perturbed constraints. We address these via two key innovations:

A) *Coupled Noise Design.* Independent perturbations of objectives and constraints are difficult to analyze and yield undesirable exponential factors ($e^{\Omega(m)}$). We instead *couple* the perturbations by adding a single random vector $\psi$ to the objective estimate and $-\psi$ to each row of the constraint estimate. This coupling ensures a high *local optimism rate*: with constant probability, the perturbed program is feasible at the true optimum $a_*$, achieving regret bounds scaling only with $\log(m)$. Empirical studies (§6,J) confirm the advantages of coupled noise.

B) *Scaling and Resampling.* The fluctuating constraints disable both existing analysis frameworks for linear TS: the 'unsaturation' approach of [AG13] and the 'optimism' approach of [AL17]. To analyze S-COLTS, we adapt the unsaturation framework with a new scaling-based trick allowing comparisons across distinct feasible regions. For R-COLTS, we instead use resampling to directly generate optimistic and feasible actions, bypassing these analytic barriers entirely.

## 1.1 Related Work

**Safe Bandits.** Safe bandits have been studied under two main notions of constraint enforcement: *soft* [CGS22; GCS24] and *hard* [AAT19; MAAT21; PGBJ21; PGB24; HTA23; HTA24]. Soft enforcement achieves regret and risk bounds of $\widetilde{O}(\sqrt{d^2 T})$, with improved instance-specific guarantees for polytopal domains. Hard enforcement achieves zero risk, and regret bounds of $\widetilde{O}(\mathcal{R}(a_{\mathsf{safe}})\sqrt{d^2 T})$ but given a safe action $a_{\mathsf{safe}}$. Efficient variants of these methods instead achieve weaker regret bounds

of $\widetilde{O}(\mathcal{R}(a_{\mathsf{safe}})\sqrt{d^3 T})$. In contrast to safe bandits, *bandits with knapsacks* [BKS13; AD16] control aggregate constraints, which is unsuitable for roundwise safety enforcement (see §C).

**Computational Complexity.** Existing efficient hard-enforcement methods rely on frequentists confidence sets for constraints, which induce $m$ expensive second-order conic (SOC) constraints during action selection [PGBJ21; PGB24; AAT19; MAAT21]. Most variants require solving $2d$ such problems per round. Our approach, S-COLTS, instead only optimises over linear constraints while maintaining near-optimal guarantees. The scaling approach inherent to S-COLTS is related to the prior ROFUL method [HTA24], although this uses the inefficient method DOSS as a subroutine.

Notably, no computationally efficient methods have previously been proposed for soft enforcement. The main point of comparison, DOSS need to solve $(2d)^{m+1}$ LPs each round [GCS24]. R-COLTS resolves this gap by sampling $O(\log(t))$ perturbed programs each round. Under mild conditions (Slater's condition), one can further reduce to a single LP per round. See §C for more details.

**Thompson Sampling (TS).** Frequentist bounds for linear TS were first established by Agrawal & Goyal [AG13] through an 'unsaturation' approach, while Abeille & Lazaric [AL17] developed a related 'global optimism' approach. Neither approach extends to SLBs due the per-round fluctuation of the perturbed constraints, and the ensuing variability of the 'feasible regions' for each round (see §C for more details). We overcome these challenges through our coupled noise design, ensuring frequent optimism, and a novel scaling trick to compare solutions across distinct feasible regions.

The only existing sampling-based treatment of unknown constraints is due to Chen et al. [CGS22] for multi-armed settings, who use posterior quantiles to enforce constraints. Although their method does not scale to continuous action sets, our resampling approach can be interpreted as an efficient, scalable analogue for simultaneously enforcing constraints and optimizing reward indices.

## 2    Problem Definition and Background

*Notation.* For a vector $v$, $\|v\|$ denotes its $\ell_2$-norm. For a PSD matrix $M$, $\|v\|_M := \|M^{1/2}v\|$. $\mathbb{S}^d$ is the unit sphere in $\mathbb{R}^d$. For a matrix $M$, $M^i$ is the $i$th row of $M$. $\mathbf{1}_m$ is the all ones vector in $\mathbb{R}^m$. Also see §A for an extensive glossary of notation used in the paper.

**Setup.** An instance of a SLB problem is defined by an objective $\theta_* \in \mathbb{R}^d$, a constraint matrix $\Phi_* \in \mathbb{R}^{m \times d}$, constraint levels $\alpha \in \mathbb{R}^m$, a compact *convex* domain $\mathcal{A} \subset \mathbb{R}^d$, and $\delta \in (0, 1)$. $\mathcal{A}, \alpha, \delta$ are known to the learner, but $\theta_*$ and $\Phi_*$ are not. The program of interest is $\max \theta_*^\top a$ s.t. $\Phi_* a \leq \alpha, a \in \mathcal{A}$, assumed to be feasible. $a_*$ denotes a(ny) maximiser of this program. The *reward gap* of $a \in \mathcal{A}$ is $\Delta(a) := \theta_*^\top (a_* - a)$, and its *safety margin* is $\Gamma(a) = \min_i (\alpha - \Phi_* a)_+^i$. For infeasible $a$, $\Gamma(a) = 0$, and $\Delta$ may be negative. We set $\mathcal{R}(a) = 1 + \Delta(a)/\Gamma(a)$ if $\Gamma(a) > 0$, and $\infty$ otherwise.

**Play.** We index rounds by $t$. At each $t$, the learner picks $a_t \in \mathcal{A}$, and receives the feedback $R_t = \theta_*^\top a_t + w_t^R$, and $S_t = \Phi_* a_t + w_t^S$, where $w_t^R \in \mathbb{R}$ and $w_t^S \in \mathbb{R}^m$ are noise processes. $C_t$ denotes algorithmic randomness at round $t$. The historical filtration is $\mathfrak{H}_{t-1} := \sigma(\{(a_s, R_s, S_s, C_s)\}_{s<t})$, and $\mathfrak{G}_t := \sigma(\mathfrak{H}_{t-1} \cup \{(a_t, C_t)\})$. The action $a_t$ must be adapted to $\sigma(\mathfrak{H}_{t-1} \cup \sigma(\{C_t\}))$.

**The Soft Enforcement SLB problem** demands algorithms that ensure, with high probability, that both the metrics $\mathbf{R}_T$ and $\mathbf{S}_T$ (see (1)) grow sublinearly with $T$.

**The Hard Enforcement SLB problem** demands algorithms that ensure, with high probability, that $\mathbf{S}_T = 0$ and $\mathbf{R}_T = o(T)$. This is enabled by a safe starting point $a_{\mathsf{safe}}$ such that $\Gamma(a_{\mathsf{safe}}) > 0$.

**Standard Assumptions.** We assume the following standard conditions [e.g. APS11] on the instance $(\theta_*, \Phi_*, \mathcal{A})$, and noise. All subsequent results only hold under these assumptions.
- *Boundedness*: $\|\theta_*\| \leq 1$, for each row $i$, $\|\Phi_*^i\| \leq 1$, and $\mathcal{A} \subset \{a : \|a\| \leq 1\}$.
- *SubGaussian noise*: $w_t := (w_t^R, (w_t^S)^\top)^\top$ is centred and 1-subGaussian given $\mathfrak{G}_t$, i.e., $\mathbb{E}[w_t|\mathfrak{G}_t] = 0$, and $\forall \lambda \in \mathbb{R}^{m+1}, \mathbb{E}[\exp(\lambda^\top w_t)|\mathfrak{G}_t] \leq \exp(\|\lambda^2\|/2)$.

To simplify the form of our bounds, we also assume that $\log(m/\delta) = o(d)$ when stating theorems.

**Background.** The (1-)RLS estimates for $\theta_*, \Phi_*$ given the history $\mathfrak{H}_{t-1}$ are

$$\hat{\theta}_t = \arg\min_{\hat{\theta}} \sum_{s<t} (\hat{\theta}^\top a_s - R_s)^2 + \|\hat{\theta}\|^2, \text{ and } \hat{\Phi}_t = \arg\min_{\hat{\Phi}} \sum_{s<t} \|\hat{\Phi} a_s - S_s\|^2 + \sum_i \|\hat{\Phi}^i\|^2.$$

The standard *confidence sets* [APS11] for $(\theta_*, \Phi_*)$ are

$$\mathcal{C}_t^\theta(\delta) = \{\widetilde{\theta} : \|\widetilde{\theta} - \hat{\theta}_t\|_{V_t} \leq \omega_t(\delta)\}, \text{ and } \mathcal{C}_t^\Phi(\delta) = \{\widetilde{\Phi} : \forall \text{ rows } i, \|\widetilde{\Phi}^i - \hat{\Phi}_t^i\|_{V_t} \leq \omega_t(\delta)\},$$

where $V_t := I + \sum_{s<t} a_s a_s^\top$, and $\omega_t(\delta) := 1 + \sqrt{1/2 \log((m+1)/\delta) + 1/4 \log(\det V_t)}$. A key standard result states that these confidence sets are *consistent* [APS11].

**Lemma 1.** *Let the consistency event at time $t$ be $\mathsf{Con}_t(\delta) := \{\theta_* \in \mathcal{C}_t^\theta(\delta), \Phi_* \in \mathcal{C}_t^\Phi(\delta)\}$, and let $\mathsf{Con}(\delta) := \bigcap_{t \geq 1} \mathsf{Con}_t(\delta)$. Under the standard assumptions, for all $\delta \in (0,1)$, $\mathbb{P}(\mathsf{Con}(\delta)) \geq 1 - \delta$.*

## 3 The Constrained Linear Thompson Sampling Approach

We begin by describing the COLTS framework. In the frequentist viewpoint, TS is a randomised method for bandits that, at each $t$, perturbs an estimate of the unknown objective, in a manner sensitive to the historical information $\mathfrak{H}_{t-1}$, and then picks actions by optimising this perturbed objective.

Naturally, then, we will perturb the estimates $\hat{\theta}_t, \hat{\Phi}_t$, for which we use a law $\mu$ on $\mathbb{R}^{1 \times d} \times \mathbb{R}^{m \times d}$. For $(\eta, H) \sim \mu$, independent of $\mathfrak{H}_{t-1}$, we define the perturbed parameters

$$\widetilde{\theta}(\eta, t)^\top := \hat{\theta}_t^\top + \omega_t(\delta)\eta V_t^{-1/2} \text{ and } \widetilde{\Phi}(H, t) := \hat{\Phi}_t + \omega_t H V_t^{-1/2}. \tag{2}$$

Notice that these perturbations are aligned with $\mathfrak{H}_{t-1}$ only via the scaling by $\omega_t(\delta)V_t^{-1/2}$. The underlying thesis of the COLTS approach is that for well-chosen $\mu$, the action

$$a(\eta, H, t) = \arg\max\{\widetilde{\theta}(\eta, t)^\top a : \widetilde{\Phi}(H, t)a \leq \alpha, a \in \mathcal{A}\}, \tag{3}$$

if it exists, is a good choice to play, in that it is either underexplored, or nearly safe and optimal (N.B. we treat $\arg\max$ as a point function that picks any one optimal solution). Two major issues arise with this view. Firstly, the set $\mathcal{A} \cap \{\widetilde{\Phi}(H, t)a \leq \alpha\}$ may be empty for certain $H$, meaning $a(\eta, H, t)$ need not exist. Secondly, in hard enforcement, $a(\eta, H, t)$ need not actually be safe, and so cannot directly be used. Thus, the main questions are 1) what $\mu$ we should use, 2) how we should augment the COLTS principle to design effective algorithms, and 3) how we can analyse these algorithms to prove effectiveness. These questions occupy the rest of this paper.

$B$**-Concentration.** Before proceeding, we note that very large $\eta, H$ can wash out all of the signal in $\hat{\theta}_t$ and $\hat{\Phi}_t$. We introduce the following definition to quantifiably limit their size.

**Definition 2.** *Let $B : (0, 1] \to \mathbb{R}_{\geq 0}$ be a nonincreasing map. A law $\mu$ on $\mathbb{R}^{1 \times d} \times \mathbb{R}^{m \times d}$ is said to satisfy $B$-concentration if $\forall \xi \in (0, 1], \mu\left(\{\max(\|\eta\|, \max_{i \in [1:m]} \|H^i\|) \geq B(\xi)\}\right) \leq \xi$.*

As an example, if each $\eta, H^i$ were normal, then $B(\xi) = \sqrt{d\log((m+1)/\xi)}$. Henceforth, we will assume that $\mu$ satisfies $B$-concentration for some map $B$, and define quantities in terms of this $B$. This condition has the following useful consequence (§F).

**Lemma 3.** *Let $\delta_t := \delta/t(t+1)$, and for $B : (0, 1] \to \mathbb{R}_{\geq 0}$, let $B_t = 1 + \max(1, B(\delta_t))$. Let*

$$M_t(a) := B_t \omega_t(\delta)\|a_t\|_{V_t^{-1}}.$$

*Let $\{(\eta_t, H_t)\}$ be a sequence of perturbation noise such that at each $t$, $(\eta_t, H_t) \sim \mu$ independently of $\mathfrak{H}_{t-1}$. If $\mu$ satisfies $B$-concentration, then with probability at least $1 - 2\delta$,*

$$\forall t, a, \max\left(|(\theta_* - \widetilde{\theta}(\eta_t, t))^\top a|, \max_i |(\widetilde{\Phi}(H_t, t)^i - \Phi_*^i)a|\right) \leq M_t(a).$$

*Further, $\sum_{t \leq T} M_t(a_t) \leq B_T \omega_T(\delta) \cdot O(\sqrt{dT}) \leq B_T \widetilde{O}(\sqrt{d^2 T})$.*

## 4 Hard Constraint Enforcement via Scaling-COLTS

We turn to the problem of hard constraint enforcement of minimising $\mathbf{R}_T$ while ensuring that w.h.p., $\mathbf{S}_T = 0$, using a safe action $a_{\mathsf{safe}}$ such that $\Gamma(a_{\mathsf{safe}}) > 0$. We will extend COLTS with a 'scaling heuristic,' that was first proposed in the context of SLBs by Hutchinson et al. [HTA24], who used it to design a (ineffient) method ROFUL.

To begin, our method, S-COLTS, draws noise $(\eta_t, H_t) \sim \mu$, and computes the preliminary action $b_t := a(\eta_t, H_t, t)$, assuming for now that this exists. As argued in §3, this action $b_t$ either has low-regret, or is informative. Of course, this $b_t$ need not be safe—we only know via Lemma 3 that $\Phi_* b_t \leq \alpha + M_t(b_t)\mathbf{1}_m$—and so cannot be used

---

**Algorithm 1** Scaling-COLTS (S-COLTS$(\mu, \delta)$)

1: **Input**: $a_{\mathsf{safe}}, \Gamma_0 \in [\Gamma(a_{\mathsf{safe}})/2, \Gamma(a_{\mathsf{safe}})]$.
2: **for** $t = 1, 2, \ldots$ **do**
3:     Draw $(\eta_t, H_t) \sim \mu$
4:     **if** $M_t(a_{\mathsf{safe}}) > \Gamma_0/3$ OR $a(\eta_t, H_t, t)$ does not exist **then**
5:         $a_t \leftarrow a_{\mathsf{safe}}$.
6:     **else**
7:         $b_t \leftarrow a(\eta_t, H_t, t)$
8:         Compute $a_t$ as in (4).
9:     Play $a_t$, observe $R_t, S_t$, update $\mathfrak{H}_t$.

---

for hard enforcement. However, the action $a_{\mathsf{safe}}$ is safe, with a large slack of at least $\Gamma(a_{\mathsf{safe}})$ in each constraint. Via linearity, and the convexity of $\mathcal{A}$, this means we can *scale back* $b_t$ towards $a_{\mathsf{safe}}$ to find a safe action, i.e., play $a_t$ of the form $(1 - \rho_t)a_{\mathsf{safe}} + \rho_t b_t$ for some $\rho_t \in [0, 1]$. If $\rho_t$ is not too small, this maintains fidelity with respect to the informative direction $b_t$, while retaining safety.

Ensuring that $1 - \rho_t$ is small relies on the margin $\Gamma(a_{\mathsf{safe}})$ of $a_{\mathsf{safe}}$. Indeed, notice that

$$\Phi_*(\rho b_t + (1 - \rho)a_{\mathsf{safe}}) \leq \alpha + (\rho M_t(b_t) - (1 - \rho)\Gamma(a_{\mathsf{safe}}))\mathbf{1}_m,$$

and so there is a safe $\rho$ satisfying $1 - \rho \leq M_t(b_t)/\Gamma(a_{\mathsf{safe}})$. Of course, we do not know $\Gamma(a_{\mathsf{safe}})$, and so cannot directly set $\rho_t$ this way. However, by repeatedly playing $a_{\mathsf{safe}}$ (and using adaptive bounds), we can find a value $\Gamma_0$ such that $\Gamma_0 \in [\Gamma(a_{\mathsf{safe}})/2, \Gamma(a_{\mathsf{safe}})]$ using only $\widetilde{O}(\Gamma(a_{\mathsf{safe}})^{-2})$ rounds. We give an account of this method in §H.1, and henceforth just assume that we know such a $\Gamma_0$.

Define $\widetilde{\theta}_t = \widetilde{\theta}(\eta_t, t)$ and $\widetilde{\Phi}_t = \widetilde{\Phi}(H_t, t)$. We observe that if $M_t(a_{\mathsf{safe}}) \leq \Gamma_0/3$, then, whp,

$$\widetilde{\Phi}_t a_{\mathsf{safe}} \leq \Phi_* a_{\mathsf{safe}} + M_t(a_{\mathsf{safe}})\mathbf{1}_m \leq \alpha - (\Gamma(a_{\mathsf{safe}}) - {}^{\Gamma_0}/3)\mathbf{1}_m \leq \alpha - {}^{2\Gamma(a_{\mathsf{safe}})}/3\mathbf{1}_m.$$

Thus, the constraints induced by $\widetilde{\Phi}_t$ are feasible (since $a_{\mathsf{safe}}$ meets them), and so, critically for S-COLTS, the action $b_t = a(\eta_t, H_t, t)$ exists. To play a safe action, we set

$$a_t = \mathfrak{a}_t(\rho_t), \ \textit{where } \mathfrak{a}_t(\rho) := (1 - \rho)a_{\mathsf{safe}} + \rho b_t, \ \textit{and} \tag{4}$$

$$\rho_t := \max\{\rho \in [0, 1] : \hat{\Phi}_t \mathfrak{a}_t(\rho) + \omega_t(\delta)\|\mathfrak{a}_t(\rho)\|_{V_t^{-1}}\mathbf{1}_m \leq \alpha\}.$$

Importantly, $M_t(a_{\mathsf{safe}}) \leq \Gamma_0/3$ yields $1 - \rho_t \leq 3M_t(b_t)/\Gamma(a_{\mathsf{safe}})$ (§H), giving similar fidelity to $b_t$ as if we knew $\Gamma(a_{\mathsf{safe}})$. If $M_t(a_{\mathsf{safe}}) > \Gamma_0/3$, we simply play $a_{\mathsf{safe}}$.

The only design variable left undetermined is the perturbation law $\mu$. In §4.1, we first describe an *unsaturation* condition on $\mu$ that induces low regret. Then, in §4.2, we give a general construction of unsaturated laws. This operationalises the S-COLTS design, with regret bounds described in §4.3.

## 4.1 Analysis of S-COLTS

Since safety of S-COLTS directly follows from (4), we main challenge is controlling regret. In this section, we show that if $\mu$ satisfies an *unsaturation* condition, then S-COLTS incurs $\widetilde{O}(\sqrt{T})$ regret. We will being by describing this unsaturation condition, and show how this is operationalised via a novel *look-back* analysis of TS, which is first presented without unknown constraints for the sake of clarity. Next we will discuss how this is augmented via a *scaling strategy* to handle the shifting constraints in S-COLTS.

**Unsaturation.** Following [AG13], we say that an action $a$ is *unsaturated* at time $t$ if $\Delta(a) \leq M_t(a)$. The core idea is that playing unsaturated actions is either informative (large $M_t(a)$), or low-regret (small $M_t(a)$). Thus, with large $\rho_t$, every time the proposed $b_t$ is unsaturated, the learning process should make progress. Of course, this $b_t$ will not always be unsaturated due to the perturbations in $\widetilde{\theta}_t, \widetilde{\Phi}_t$, but it suffices for $b_t$ to be unsaturated often enough. This motivates the following definition.

**Definition 4.** *Let $\mu$ be a $B$-concentrated law. Define the* unsaturation event *at time $t$ as*

$$\mathsf{U}_t(\delta) := \{(\eta, H) : a(\eta, H, t) \textit{ exists, and } \Delta(a(\eta, H, t)) \leq M_t(a(\eta, H, t)).$$

*For $\chi \in (0, 1]$, we say that $\mu$-satisfies $\chi$-unsaturation if for all $t$ such that $\delta/(t(t + 1)) \leq \chi/2$,*

$$\mathbb{P}[\mathsf{U}_t(\delta)|\mathfrak{H}_{t-1}]\mathbb{1}_{\mathsf{Con}_t(\delta)} = \mathbb{E}[\mu(\mathsf{U}_t(\delta))|\mathfrak{H}_{t-1}]\mathbb{1}_{\mathsf{Con}_t(\delta)} \geq (\chi/2)\mathbb{1}_{\mathsf{Con}_t(\delta)}.$$

In words, $\chi$-unsaturation means that at all $t$, given the past, $b_t$ is unsaturated with chance at least $\chi/2$.

### 4.1.1 Using Unsaturation: The Look-Back method without Unknown Constraints

For the sake of clarity, let us first consider how we can analyze TS without unknown constraints using this unsaturation definition. We shall do so via a novel 'look-back' technique, which operationalises commonly supplied intuition for how TS works [e.g. AL17], and thus offers a more intuitive argument than the prior approach based upon studying the minimum-norm unsaturated action [AG13].

Without unknown constraints, S-COLTS collapses to standard TS by setting $a_t = b_t$ (and we will only use $a_t$ below). Now, suppose $a_t$ were always unsaturated. Then observe that we get the bound

$$\mathbf{R}_T = \sum \Delta(a_t) \leq \sum M_t(a_t) = \widetilde{O}(B_T\sqrt{d^2 T}).$$

However, in reality, $a_t$ is often not unsaturated. To handle this, we will 'look back' at the *last time* $s < t$ that $a_s$ was unsaturated. Specifically, define

$$\tau(t) := \inf\{s < t : \Delta(a_s) \leq M_s(a_s)\}, \quad \inf \emptyset := 0.$$

This $\tau(t)$ is the last $s$ with an unsaturated $a_s$, and so, per the unsaturation heuristic, is the last time the learner made progress. The main idea of looking back is to control $\Delta(a_t)$ in terms of the information available at $\tau(t)$ to exploit the 'steady' learning at such time steps. To this end, we will bound $\Delta(a_t)$ in terms of $a_{\tau(t)}$. To lower density of notation, we will write $\tau$ instead of $\tau(t)$ unless necessary.

**Introducing $a_\tau$.** Notice that since $\Delta(a) = \theta_*^\top(a_* - a)$, we can introduce $a_\tau$ into the control thus:

$$\Delta(a_t) = \Delta(a_\tau) + \theta_*^\top(a_\tau - a_t) \leq M_\tau(a_\tau) + \theta_*^\top(a_\tau - a_t),$$

where we used the unsaturation of $a_\tau$. Bounding this requires us to control the second term. For this, we use the resource that $a_t$ optimises $\widetilde{\theta}_t$, and so $\widetilde{\theta}_t^\top(a_\tau - a_t) \leq 0$. Thus,

$$\Delta(a_t) \leq M_\tau(a_\tau) + (\theta_* - \widetilde{\theta}_t)^\top(a_\tau - a_t) + \widetilde{\theta}_t^\top(a_\tau - a_t),$$

where the third term can be dropped. Further note that the second term decomposes as two terms of the form $(\theta_* - \widetilde{\theta}_t)^\top a$, which is precisely the object controlled in Lemma 3. Using this gives

$$\Delta(a_t) \leq M_\tau(a_\tau) + M_t(a_\tau) + M_t(a_t). \tag{5}$$

Now, the final term accumulates to $\sum M_t(a_t)$, which by Lemma 3 is $\widetilde{O}(\sqrt{T})$. Thus, to control regret, we only need to bound the accumulation of these look back terms $\sum_t M_{\tau(t)}(a_{\tau(t)}) + M_t(a_{\tau(t)})$.

**Controlling Look-Back Accumulation.** Our main resource for controlling this is $\chi$-unsaturation. Consider just the first term $\sum_t M_{\tau(t)}(a_{\tau(t)})$ (the second follows similarly), and let $T_1 \leq T_2 \leq \cdots$ denote the (stopping) times at which $a_t$ was unsaturated. Then notice that

$$\sum_{t \leq T} M_{\tau(t)}(a_{\tau(t)}) = \sum_{t=1}^{T_1} M_{\tau(t)}(a_{\tau(t)}) + \sum_{t=T_1+1}^{T_2} M_{\tau(t)}(a_{\tau(t)}) + \cdots = \sum_{i \geq 1}(T_i - T_{i-1})M_{T_{i-1}}(a_{T_{i-1}}),$$

where we set $T_0 = 0$ and $M_0(\cdot) = 1$ for consistency. But notice that due to the frequency of unsaturation, each $T_i - T_{i-1}$ is only $O(\log(T)/\chi)$ with high probability. This suggests the bound

$$\sum_t M_{\tau(t)}(a_{\tau(t)}) \leq \widetilde{O}(\chi^{-1}) \sum_i M_{T_i}(a_{T_i}) \leq \widetilde{O}(\chi^{-1}) \sum_{\text{all } t \leq T} M_t(a_t) = \widetilde{O}(\chi^{-1} B_T \sqrt{d^2 T}),$$

where in the second inequality, we used the nonnegativity of $M_t$. Using a more refined martingale analysis described in §H.3, this yields the following key structural tool for the look-back method.

**Lemma 5.** *If $\mu$ satisfies $\chi$-unsaturation, then with probability at least $1 - \delta$, for all $T \geq \sqrt{2/\chi}$,*

$$\sum_{t \leq T} M_{\tau(t)}(a_{\tau(t)}) + M_t(a_{\tau(t)}) \leq 10 B_T \omega_T(\delta_T) \cdot \chi^{-1} \cdot \left(\log(1/\delta) + \sum_{t \leq T} \|a_t\|_{V_t^{-1}}\right).$$

Since (Lemma 3) $\omega_T \sum \|a_t\|_{V_t^{-1}} = \widetilde{O}(d^2\sqrt{T})$, this results in a proof that TS with a $\chi$-unsaturated perturbation law admits the bound $\mathbf{R}_T = \widetilde{O}(\chi^{-1} B_T \sqrt{d^2 T})$ with no unknown constraints.

#### 4.1.2 Looking Back with Unknown Constraints: The Analysis of S-COLTS

In the above analysis, we critically used the optimality of the action for the perturbed program, and the frequency of unsaturation. Coming back to S-COLTS, let us recall that our action is the convex combination $a_t = \rho_t b_t + (1 - \rho_t)a_{\mathsf{safe}} \neq b_t$, and that unsaturation means that $\Delta(b_t) \leq M_t(b_t)$.

**Handling the Scaling.** Our first course of action, then, is to observe via linearity that

$$\Delta(a_t) = \rho_t \Delta(b_t) + (1 - \rho_t)\Delta(a_{\mathsf{safe}}),$$

and regret control requires us to bound the sum of these across $t$. Notice that even if $b_t$ were always unsaturated, we would only get the bound $\sum \rho_t M_t(b_t) + \sum(1 - \rho_t)\Delta(a_{\mathsf{safe}})$, and nominally, neither term is controlled by $\sum M_t(a_t)$. We handle this by the following, which relies on the largeness of $\rho_t$.

**Lemma 6.** *At any $t$ such that $M_t(a_{\mathsf{safe}}) \leq \Gamma_0/3$, it holds that*

$$(1 - \rho_t)\Gamma(a_{\mathsf{safe}}) \leq 6 M_t(a_t) \quad and \quad \rho_t M_t(b_t) \leq 2 M_t(a_t).$$

This follows from a more detailed result, Lemma 20 in §H.2. Observe as a consequence that $\sum(1 - \rho_t)\Delta(a_{\mathsf{safe}}) \leq O(\Delta(a_{\mathsf{safe}})/\Gamma(a_{\mathsf{safe}})) \cdot \sum M_t(a_t)$ is well-controlled (and, of course, $\Delta(a_{\mathsf{safe}})/\Gamma(a_{\mathsf{safe}}) = \mathcal{R}(a_{\mathsf{safe}}) - 1$). So, the main object of study is $\sum \rho_t \Delta(b_t)$.

**Looking-Back with Unknown Constraints.** We slightly adjust the look-back time so that $\rho_\tau M_\tau(b_\tau) \leq 2M_\tau(a_\tau)$ holds:

$$\tau(t) := \inf\{s < t : M_s(a_{\sf safe}) \leq \Gamma_0/3, \Delta(b_s) \leq M_s(b_s)\}.$$

Following §4.1.1 directly suggests that we should proceed via

$$\rho_t \Delta(b_t) \leq \rho_t(\Delta(b_\tau) + (\theta_* - \widetilde{\theta}_t)^\top (b_\tau - b_t) + \widetilde{\theta}_t^\top (b_\tau - b_t)).$$

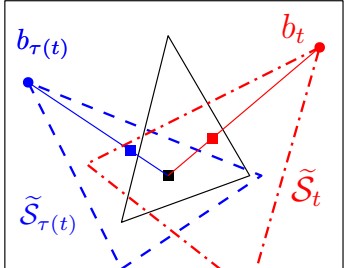

However, a major issue emerges here: notice that $b_t$ optimises the constrained program $\max\{\theta_t^\top a : \widetilde{\Phi}_t a \leq \alpha\}$. However, the look-back action $b_\tau$ only satisfies the constraint $\{\widetilde{\Phi}_\tau a \leq \alpha\}$, and so $b_\tau$ may not be feasible for the perturbed constraints at time $t$. This prevents us from simply dropping the final term above, and breaks the analysis.

We address this by using essentially the same idea as that underlying S-COLTS itself: move $b_\tau$ towards $a_{\sf safe}$ to find a point that is feasible for $\widetilde{\Phi}_t$, but has regret similar to $b_\tau$. Concretely, we set

Figure 1: A schematic of the analysis. $\widetilde{S}_s := \{\widetilde{\Phi}_s a \leq \alpha\}$ are the perturbed feasible regions. $a_t$ (red box) is a mixture of $b_t$ and $a_{\sf safe}$ (black box). $b_{\tau(t)}$ is infeasibe for $\widetilde{S}_t$, and we instead mix it with $a_{\sf safe}$ to produce $\bar{b}_{\tau(t)\to t} \in \widetilde{S}_t$ (blue box).

$$\bar{b}_{\tau\to t} := \sigma_{\tau\to t} b_\tau + (1 - \sigma_{\tau\to t})a_{\sf safe}, \quad \text{where } \sigma_{\tau\to t} := \Gamma_0/(\Gamma_0 + 3M_t(b_\tau) + 3M_\tau(b_\tau)).$$

Importantly, notice that $\sigma_{\tau\to t}$ essentially acts the same way as $\rho_\tau$! Indeed, part of our analysis in §H.2 essentially shows that Lemma 6 holds with $\sigma_{\tau\to t}M_\tau(b_\tau)$ and $(1 - \sigma_{\tau\to t})$ instead of $\rho_\tau$.

In any case, the main point is thus: by two applications of Lemma 3, $\widetilde{\Phi}_\tau b_\tau \leq \alpha \implies \widetilde{\Phi}_t \bar{b}_{\tau\to t} \leq \alpha$. So, instead of $b_\tau$, our look-back analysis will focus on $\bar{b}_{\tau\to t}$. Applying this gives

$$\rho_t \Delta(b_t) \leq \rho_t \Delta(\bar{b}_{\tau\to t}) + \rho_t(\theta_* - \widetilde{\theta}_t)^\top (\bar{b}_{\tau\to t} - b_t) + \rho_t \widetilde{\theta}_t^\top (\bar{b}_{\tau\to t} - b_t),$$

where the final term is negative, and the middle term is bounded by $M_t(\bar{b}_{\tau\to t}) + 2M_t(a_t)$. For the first term, notice by linearity and unsaturation of $b_\tau$ that

$$\Delta(\bar{b}_{\tau\to t}) = \sigma_{\tau\to t}\Delta(b_t) + (1 - \sigma_{\tau\to t})\Delta(a_{\sf safe}) \leq \sigma_{\tau\to t}M_\tau(b_\tau) + (1 - \sigma_{\tau\to t})\Delta(a_{\sf safe}).$$

This leaves us with a bound in terms of $\sigma_{\tau\to t}$ and $M_t(b_\tau)$, the core resource for which is the observation that $\sigma_{\tau\to t}$ acts essentially as $\rho_\tau$, which lets us write, e.g., that $\sigma_{\tau\to t}M_\tau(b_\tau) \leq 2M_\tau(a_\tau)$. For the sake of brevity, we leave this analysis to §H.2, and only state the main resulting bound.

**Lemma 7.** *Let $M_0(a) := 1$. If $\mu$ is $B$-concentrated, then with probability at least $1 - 3\delta$,*

$$\forall t : M_t(a_{\sf safe}) \leq \Gamma_0/3, \Delta(a_t) \leq 6\mathcal{R}(a_{\sf safe})(M_t(a_t) + M_t(a_{\tau(t)}) + M_{\tau(t)}(a_{\tau(t)})).$$

Importantly notice that the bound is in terms of $a_{\tau(t)}$, instead of $b_{\tau(t)}$. Of course, this puts us in the same situation as in (5) in §4.1.1, but with an extra $6\mathcal{R}(a_{\sf safe})$ factor in the bound. Via Lemma 5, and an analysis of the number of times $M_t(a_{\sf safe}) > \Gamma_0/3$ can hold, this yields a bound of the form $\mathbf{R}_T = \widetilde{O}(\chi^{-1}\mathcal{R}(a_{\sf safe})B_T\sqrt{d^2T})$.

**Novelty Relative to Prior Work.** As previously mentioned, our look-back approach is a novel, and more intuitive, modification of the seminal analysis of unconstrained TS by Agrawal and Goyal [AG12; AG13]. More importantly, with unknown constraints, we had to handle *fluctuating constraint sets*: our look-back analysis broke since $b_\tau$ could be infeasible for $\widetilde{\Phi}_t$, which we addressed by scaling $b_\tau$. This issue would also break an analysis based directly on the approach of [AG12]. The novelty relative to [HTA24] is similar: like $b_\tau$, $a_*$ may also be infeasible for $\Phi_t$, which breaks their analysis.

## 4.2 The Coupled Noise Design

§4.1 shows that $\chi$-unsaturation yields control on the regret. To operationalise this, we need to design well-concentrated laws with good unsaturation. In single-objective TS, unsaturation is enabled via anticoncentration of the $\eta$s, and a good balance is attained by, e.g., $\text{Unif}(\sqrt{3d}\mathbb{S}^d)$ or $\mathcal{N}(0, I_d)$.

A natural guess with unknown constraints is to sample both $\eta$ and each row of $H$ from such a law. However, the unsaturation rate under such a design is difficult to control well. The main issue arises from maintaining feasibility with respect to all $m$ constraints under perturbation, since each such perturbation gets an independent shot at shaving away some unsaturated actions, suggesting that $\chi$ decays as $e^{-\Omega(m)}$ and indeed, experimentally, increase in $m$ may lead to at least a polynomial decay in the unsaturated rate with such independent noise (see §J.3). We sidestep this issue by *coupling* the perturbations of the reward and constraints, as encapsulated below.

**Lemma 8.** *Let $\bar{B} \in \{(0,1] \to \mathbb{R}_{\geq 0}\}$ be a map, and $p \in (0,1]$. Let $\nu$ be a law on $\mathbb{R}^{d \times 1}$ such that*

$$\forall u \in \mathbb{R}^d, \nu(\{\zeta : \zeta^\top u \geq \|u\|\}) \geq p, \text{ and } \forall \xi \in (0,1], \nu(\{\zeta : \|\zeta\| > \bar{B}(\xi)\}) \leq \xi.$$

*Let $\mu$ be the law of $\zeta \mapsto (\zeta^\top, -\mathbf{1}_m \zeta^\top)$ for $\zeta \sim \nu$. Then $\mu$ is $p$-unsaturated and $\bar{B}$-concentrated.*

Our proof of this lemma, executed in §G, is based upon analysing the *local optimism event* at $a_*$:

$$\mathsf{L}_t(\delta) := \{(\eta, H) : \widetilde{\theta}(\eta, t)^\top a_* \geq \theta_*^\top a_*, \widetilde{\Phi}(H, t) a_* \leq \alpha\}. \tag{6}$$

Notice that $\mathsf{L}_t$ demands that the perturbation is such that $a_*$ remains feasible with respect to $\widetilde{\Phi}_t$, and its value at $\widehat{\theta}_t$ increases beyond $\theta_*^\top a_*$, in other words, the perturbed program is optimistic *at $a_*$*. Our proof first directly analyses $a_*$ under the perturbations to show that $\mathbb{P}[\mathsf{L}_t(\delta)|\mathfrak{H}_{t-1}]\mathbb{1}_{\mathsf{Con}_t(\delta)} \geq p\mathbb{1}_{\mathsf{Con}_t(\delta)}$, i.e., frequent local optimism. This enables an argument due to [AG13]: since $a_*$ is unsaturated ($\Delta(a_*) = 0$), and, w.h.p. the perturbed reward of any saturated action is dominated by that of $a_*$, it follows that $\mathsf{L}_t(\delta) \subset \mathsf{U}_t(\delta)$, yielding lower bounds on $\mu(\mathsf{U}_t(\delta))$.

We note that the conditions of Lemma 8 are the same as those used for unconstrained linear TS in prior work [AG13; AL17], and so this generic result extends this unconstrained guarantee to the constrained setting. In our bounds, we will set $\mu$ to be the law induced by the coupled design with $\nu = \text{Unif}(\sqrt{3d}\mathbb{S}^d)$, which is 0.14-unsaturated, and $B$-concentrated for $B(\xi) = \sqrt{3d}$ (§G.1).

### 4.3 Regret Bounds for S-COLTS

With the pieces in place, we state and discuss our main result, which is formally proved in §H.

**Theorem 9.** *Let $\mu$ be the law induced by $\text{Unif}(\sqrt{3d}\mathbb{S}^d)$ under the coupled noise design. Then* S-COLTS$(\mu, \delta/3)$ *ensures that with probability at least $1 - \delta$, for all $T$, it holds that*

$$\mathbf{S}_T = 0 \quad \text{and} \quad \mathbf{R}_T = \mathcal{R}(a_{\mathsf{safe}}) \cdot \widetilde{O}(\sqrt{d^3 T + d^2 T \log(m/\delta)}) + \widetilde{O}(d^2 \Delta(a_{\mathsf{safe}})\Gamma(a_{\mathsf{safe}})^{-2}).$$

**Comparison of Regret Bounds to Prior Results.** As noted in §1.1, prior inefficient hard enforcement SLB methods attain regret $\widetilde{O}(\mathcal{R}(a_{\mathsf{safe}})\sqrt{d^2 T})$, while efficient methods attain regret $\widetilde{O}(\mathcal{R}(a_{\mathsf{safe}})\sqrt{d^3 T})$. Our results above recover the latter bounds. The loss of $\sqrt{d}$ relative to inefficient methods is expected since it appears in all known efficient linear bandit methods (without or without unknown constraints). The $\Omega(\sqrt{T})$ dependence is necessary (even with instance-specific information) [GCS24] as is the additive $\Delta(a_{\mathsf{safe}})/\Gamma(a_{\mathsf{safe}})^2$ term [PGBJ21]. Thus, S-COLTS recovers previously known guarantees using sampling rather than frequentist bounds.

**Computational Aspects.** An advantage of S-COLTS is that it only optimises over linear constraints (beyond those of $\mathcal{A}$), instead of SOC constraints of the form $\{\forall i \in [1:m], \hat{\Phi}_t^i a + \omega_t(\delta)\|a\|_{V_t^{-1}} \leq \alpha^i\}$ imposed by prior methods. While convex, these $m$ SOC constraints can have a palpable practical slowdown on the time needed for optimisation, especially as $m$ grows (over $\mathcal{A} = [0, 1/\sqrt{d}]^d$, with the modest $d = m = 9$ we see a $> 5\times$ speedup, and with $d = 2, m = 100$, a $18\times$ speedup, in §6). In particular, when $\mathcal{A}$ is a polyhedron, S-COLTS can be implemented with just linear programming.

We explicitly note that S-COLTS is efficient for convex $\mathcal{A}$. The dominating step is the computation of $b_t$, which can be carried out to an approximation of $1/t$ with no loss in Theorem 9. With, say, interior point methods, this needs $O(\mathsf{LP} \cdot \log(t))$ computation at round $t$, where LP is the computation needed to optimise $\max\{\theta^\top a : \Phi a \leq \alpha, a \in \mathcal{A}\}$ to constant error [BV04].

**Practical Choice of Noise.** It has long been understood that while existing theoretical techniques for analysing linear TS need large noise (with $B(\xi) = \Theta(\sqrt{d})$), in practice much smaller noise (e.g., $\text{Unif}(\mathbb{S}^d)$ with $B(\xi) = \Theta(1)$) typically retains a large enough rate of unsaturation, and significantly improve regret (although not in the worst-case [HB20]). Our practical recommendation is to indeed use such a small noise, which we find to be effective in simulations (§6). We underscore that no matter the noise used, the risk guarantee for S-COLTS is maintained.

## 5 Soft Constraint Enforcement with Resampling-COLTS

Given an action $a_{\mathsf{safe}}$ with positive safety margin, S-COLTS ensures strong safety and good regret. This section studies scenarios where we do not know such an $a_{\mathsf{safe}}$. In this case, it is impossible to ensure that $\mathbf{S}_T = 0$, and we instead show $\widetilde{O}(\sqrt{T})$ bounds on $\mathbf{S}_T$, following prior work [GCS24].

S-COLTS uses forced exploration of $a_{\mathsf{safe}}$ to ensure the feasibility of perturbed programs. However, the local optimism underlying our proof of Lemma 8 gives a different way to achieve this. Indeed, the

event $\mathsf{L}_t(\delta)$ of (6) implies that $a_*$ is feasible, and so $a(\eta, H, t)$ exists. Thus, if $\mathbb{P}[\mathsf{L}_t(\delta)|\mathfrak{H}_{t-1}] \geq \pi$, then we can just resample the noise $O(\log(t))$ times and end up with feasibility. In fact, even more is true: since $\widetilde{\theta}^\top a_* \geq \theta_*^\top a_*$ under $\mathsf{L}_t$, resampling $\pi^{-1}\Theta(\log(t))$ times ensures not only feasibility, but also *optimism* of the 'best' perturbed optimum. The R-COLTS method is based on this observation.

Concretely, given a resampling parameter $r$, at time $t$ R-COLTS samples $1 + \lceil r \log{t(t+1)/\delta}\rceil$ independent $(\eta, H)$ from $\mu$, optimises the perturbed program induced by each, and picks the optimum of the one with largest value as $a_t$ (and sets it $= a_{t-1}$ if all programs were infeasible). We let $\widetilde{\theta}_t$ denote the objective of this 'winning' perturbed program: in the notation of Alg. 2, $\widetilde{\theta}_t = \widetilde{\theta}(\eta_{i_*,t}, t)$. The main idea is captured in the following simple lemma.

**Lemma 10.** *Let $\pi \in (0, 1]$, and suppose $\mu$ satisfies $\mathbb{1}_{\mathsf{Con}_t(\delta)}\mathbb{E}[\mu(\mathsf{L}_t(\delta))|\mathfrak{H}_{t-1}] \geq \pi\mathbb{1}_{\mathsf{Con}_t(\delta)}$ for every $t$. If $r \geq \pi^{-1}$, then with probability at least $1 - 2\delta$, at all $t$, the actions $a_t$ and perturbed objective $\widetilde{\theta}_t$ selected by R-COLTS$(\mu, r, \delta)$ are optimistic, i.e., they satisfy that $\theta_*^\top a_* \leq \widetilde{\theta}_t^\top a_t$.*

The 'local optimism condition' on $\mu$ above is reminiscent of the global optimism condition of Abeille & Lazaric [AL17], and indeed the same result holds under a global optimism assumption with unknown constraints. However, the analysis in this prior work does not extend to unknown constraints due to its reliance of convexity (§1.1), and resampling bypasses this issue. See §D for more details.

Lemma 10 enables the use of standard optimism based regret analyses [e.g. APS11]. By operationalising the condition on $\mu$ via the coupled design in §4.2, we show

**Theorem 11.** *If $\mu$ is the law induced by Unif$(\sqrt{3d}\mathbb{S}^d)$ under the coupled design of Lemma 8, then with probability at least $1 - \delta$, R-COLTS$(\mu, 4, \delta/2)$ ensures that for all T,*

$$\max(\mathbf{S}_T, \mathbf{R}_T) = \widetilde{O}(\sqrt{d^3 T + d^2 T \log(m/\delta)}).$$

---

**Algorithm 2** Resampling-COLTS (R-COLTS$(\mu, r, \delta)$)

1: **Input**: $\mu, \delta$, 'resampling order' $r \in \mathbb{N}$
2: **Initialise**: $I_t \leftarrow 1 + \lceil r \log{t(t+1)/\delta}\rceil$
3: **for** $t = 1, 2, \dots$ **do**
4:     **for** $i = 1, 2, \dots, I_t$ **do**
5:         Draw $(\eta_{i,t}, H_{i,t}) \sim \mu$.
6:         **if** $a(\eta_{i,t}, H_{i,t}, t)$ exists **then**
7:             $K(i,t) \leftarrow \widetilde{\theta}(\eta_{i,t}, t)^\top a(\eta_{i,t}, H_{i,t}, t)$
8:         **else**
9:             $K(i,t) \leftarrow -\infty$
10:     **if** $\max K(i,t) = -\infty$ **then**
11:         $a_t \leftarrow a_{t-1}$.
12:     **else**
13:         $i_{*,t} \leftarrow \arg\max_i K(i,t)$,
14:         $a_t \leftarrow a(\eta_{i_*,t,t}, H_{i_*,t,t}, t)$.
15:         $\widetilde{\theta}_t \leftarrow \widetilde{\theta}(\eta_{i_*,t,t}, t)$.
16:     Play $a_t$, observe $R_t, S_t$, update $\mathfrak{H}_t$.

---

**Instance-Independent Regret Bound.** The above result limits both regret and risk to $\widetilde{O}(\sqrt{d^3 T})$, with no instance-specific terms, unlike $\mathcal{R}(a_{\mathsf{safe}})$ in S-COLTS. In particular, this bound holds even if $\max_a \Gamma(a) = 0$, i.e., the problem is marginally feasible. This result is directly comparable to the $\widetilde{O}(\sqrt{d^2 T})$ bound on both regret and risk under the DOSS method [GCS24], and loses a $\sqrt{d}$-factor relative to this, a loss that appears in all known efficient linear bandit methods.

**Computational Costs.** R-COLTS with $\mu$ as above solves $\sim 4\log(t^2/\delta)$ optimisations of $\widetilde{\theta}_t^\top a$ over $\{\Phi_t a \leq \alpha\} \cap \mathcal{A}$. Again, Theorem 11 is resilient to approximation of, say, $1/t$, and so this takes $O(\mathsf{LP} \cdot \log^2 t)$ computation per round, a factor of $\log(t)$ slower than S-COLTS, but still efficient in the practical regime of $\log(T/\delta) = O(\mathrm{poly}(d, m))$. The main point of comparison, however, is DOSS, which instead needs to solve $(2d)^{m+1}$ such programs, and so uses $(2d)^{m+1}\mathsf{LP} \cdot \log(t)$ computation per round. R-COLTS is practically *much faster* even for small domains with long horizons—for instance, with $T = 1/\delta = 10^{10}$, $4\log(t^2/\delta) \leq (2d)^{m+1}$ for all $d \geq 4, m \geq 2$.

**Relationship to Posterior Quantile Indices and Safe MABs.** The resampling approach executed in R-COLTS is closely related to the posterior-quantile approach of the BAYESUCB method [KCG12], wherein it is proposed to use a quantile of the arm posteriors as a reward index instead of a frequentist upper confidence bound. Indeed, we can compute such a quantile in a randomised way by taking many samples from the posterior of each arm, and then picking the largest of the samples as the reward index. Most pertinently, this approach was proposed for safe multi-armed bandits [CGS22], wherein this posterior quantile index is used to decide on the 'plausible safety' of putative actions. The same work further argued that the usual *single-sample* TS cannot obtain sublinear regret in safe MABs. The R-COLTS approach can be viewed as an efficient extension of this principle to linear bandits with continuum actions, and differs by directly optimising the indices under each draw, and then picking the largest, instead of performing an untenable per-arm posterior quantile computation.

**R-COLTS Without Resampling.** Given the lack of a safe action to play, one cannot direct establish the feasibility of the perturbed programs by contracting the confidence radius of a single action as in S-COLTS. However, if we introduce a small amount of 'flat' exploration whenever $V_t$ is 'small', then

this ensures that any $a$ with $\Gamma(a) > 0$ will eventually be strictly feasible under perturbations. If such $a$ exists, we only need a single noise draw to attain feasibility, and can bootstrap the scaling analysis of S-COLTS to show bounds. We term this method 'exploratory-COLTS', or E-COLTS, and specify and analyse it in §I.2. This results in the following soft-enforcement guarantee.

**Theorem 12.** *If $\mu$ is the law induced by $\mathrm{Unif}(\sqrt{3d}\mathbb{S}^d)$ under the coupled noise design, then the* E-COLTS$(\mu, \delta/3)$ *method of Algorithm 3 ensures that with probability at least $1 - \delta$, for all $T$,*

$$\mathbf{S}_T = \widetilde{O}(\sqrt{d^3 T}) + \min_a \widetilde{O}\Big(\frac{d^3\|a\|^4}{\kappa^2 \Gamma(a)^4}\Big), \text{ and } \mathbf{R}_T = \min_{a:\Gamma(a)>0}\left\{\mathcal{R}(a)\widetilde{O}(\sqrt{d^3 T}) + \widetilde{O}\Big(\frac{d^3\|a\|^4}{\kappa^2 \Gamma(a)^4}\Big)\right\},$$

*where $\kappa$ is a constant depending on the geometry of $\mathcal{A}$.*

Relative to R-COLTS, the above guarantees are instance-dependent, and are only nontrivial if $\max_a \Gamma(a) > 0$, i.e., the Slater parameter of the optimisation problem induced by $\theta_*, \Phi_*, \mathcal{A}$ is nonzero. The advantage of E-COLTS lies in its reduced computation. Comparing to S-COLTS, the above loses the strong $\mathbf{S}_T = 0$ safety, but improves regret by adapting to the best possible $\mathcal{R}(a)$.

# 6 Simulations

We give a brief summary of our simulations, leaving most details, and well as deeper investigation of our methods to §J. In all cases, we utilise the coupled noise design, driven with the (uninflated) noise $\nu = \mathrm{Unif}(0.5 \cdot \mathbb{S}^d)$, in accordance with the discussion in §4. The same noise is used for SAFE-LTS.

**Resampling tradeoff in R-COLTS.** For $d = 9$, we optimise $\theta_* = \mathbf{1}_d/\sqrt{d}$ over $\mathcal{A} = [0, 1/\sqrt{d}]^d$, with a $9 \times 9$ constraint matrix (i.e., $m = 9$). In this case, the action $0$ is feasible, and so R-COLTS without any resampling is effective. Since $a = 0$ has a nontrivial safety margin, R-COLTS, even without resampling, is effective for this problem. This is borne out in Table 2,

Table 2: $\mathbf{R}_T$ and $\mathbf{S}_T$ at $T = 5 \cdot 10^4$ for R-COLTS with $1, 2, 3$ samples per round (100 trials).

| Samples | $\mathbf{R}_T$ | $\mathbf{S}_T$ |
|---------|----------------|----------------|
| 1 | $658 \pm 170$ | $2891 \pm 171$ |
| 2 | $397 \pm 116$ | $3126 \pm 137$ |
| 3 | $301 \pm 102$ | $3266 \pm 172$ |

which shows regret and risk at the terminal time $T$. We see that resampling slightly worsens risk, but significantly improves regret (although with diminishing returns). Further, both regret and risk are far below the $\sqrt{d^2 T}$ scale expected from our bounds. We note that while a single iteration of R-COLTS takes $\sim 1$ms, since $(2d)^{m+1} > 10^{12}$, this would take *years* for DOSS, and so we do not implement it. In any case, note that the computational advantage of R-COLTS is extremely strong.

**Significant Computational Advantage and Regret Parity/Improvement of S-COLTS.** We compare S-COLTS with the hard enforcement method SAFE-LTS [MAAT21], which has been shown to match the performance of alternate such methods, while being faster. Both methods are run on the $d = m = 9$ instance above, with $a_{\mathsf{safe}} = 0$. As expected, both never play unsafe actions. Further (Fig. 2, left), S-COLTS achieves an *improvement* in regret relative to SAFE-LTS, while reducing wall-clock time by a $5.1\times$. To gain a deeper understanding of S-COLTS's computational advantage, we investigate the same with growing $m \in \{1, 10, 20, \ldots, 100\}$ constraints for a simple $d = 2$ setting (see §J.2.1 for the setup). In this problem, the benefit is even starker (Fig. 2, right). For $m \geq 10$, the regret of SAFE-LTS is $2 - 4\times$ larger than that of S-COLTS, i.e., the latter has much better regret ($m = 1$ has wide confidence bands for the ratio, but mean $\sim 1.5$) Further, the computational costs of SAFE-LTS relative to S-COLTS grow roughly linearly, starting from $\approx 1.3\times$ for $m = 1$ to $> 18\times$ at $m = 100$.

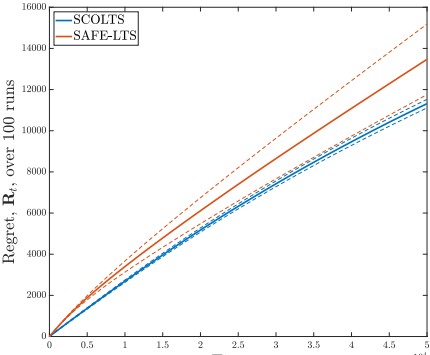 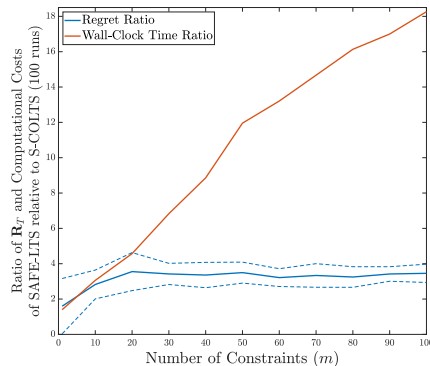

Figure 2: COMPUTATIONAL AND REGRET COMPARISONS OF S-COLTS AND SAFE-LTS. *Left*. Regret traces in the $d = 9$ instance (dashed lines are one-sigma error bars); S-COLTS mildly improves regret, and is $5\times$ faster. *Right*. Relative performance as $m$ is varied in the $d = 2$ instance. The speedup of S-COLTS grows linearly with $m$ from $1.3\times$ to $> 18\times$. Further, for $m \geq 10$, the regret of S-COLTS is 2-3$\times$ smaller than that of SAFE-LTS

## Acknowledgements

The authors would like to thank Aldo Pacchiano for helpful discussions. This research was supported by the Army Research Office Grant W911NF2110246, AFRL Grant FA8650-22-C1039, and the National Science Foundation grants CPS-2317079, CCF-2007350, and CCF-1955981.

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

# A  Glossary

| Symbol | Explanation | Expression/Comments |
|---|---|---|
| $(\theta_*, \Phi_*)$ | True objective/constraints | $\in \mathbb{R}^{d\times 1} \times \mathbb{R}^{m\times d}$ |
| $\alpha$ | Constraint level | $\in \mathbb{R}^{m\times 1}$ |
| $\mathcal{A}$ | Action domain | |
| $a_*$ | Optimal action for $(\theta_*, \Phi_*)$ | $\arg\max\{\theta_*^\top a : a \in \mathcal{A}, \Phi_* a \leq \alpha\}$ |
| $K(\theta, \Phi)$ | Value function | $\sup\{\theta^\top a : a \in \mathcal{A}, \Phi a \leq \alpha\}$, $-\infty$ if $\{\Phi a \leq \alpha\} \cap \mathcal{A} = \varnothing$. |
| $\Delta(a)$ | Reward gap | $\theta_*^\top (a_* - a)$ |
| $\Gamma(a)$ | Safety margin | $\min_i ((\alpha - \Phi_* a)^i)_+$ |
| $\mathcal{R}(a)$ | Gap-margin ratio | $1 + (\Delta(a)/\Gamma(a))$ |
| Estimation and Signal | | |
| $\mathfrak{H}_{t-1}$ | Historical filtration | See §2 |
| $\hat\theta_t, \hat\Phi_t$ | RLS-estimates of parameters | See §2 |
| $V_t$ | Action second moment | $I + \sum_{s<t} a_s a_s^\top$ |
| $\omega_t(\delta)$ | Confidence radius | See §2 |
| $\mathcal{C}_t^\theta, \mathcal{C}_t^\Phi$ | Confidence sets for $\theta_*, \Phi_*$ | |
| $\mathsf{Con}_t(\delta)$ | Consistency event at time $t$ | $\{\theta_* \in \mathcal{C}_t^\theta(\delta), \Phi_* \in \mathcal{C}_t^\Phi(\delta)\}$ |
| $\mathsf{Con}(\delta)$ | Overall consistency | $\bigcap_{t\geq 1} \mathsf{Con}_t(\delta)$ |
| COLTS in general | | |
| $\mu$ | Perturbation law | Distribution on $\mathbb{R}^{1\times d} \times \mathbb{R}^{m\times d}$ |
| $(\eta, H)$ | Perturbation noise | $\sim \mu$, independently of $\mathfrak{H}_{t-1}$ |
| $\widetilde\theta(\eta, t)$ | Pertrubed objective | $\hat\theta_t + \omega_t(\delta)\eta V_t^{-1/2}$ |
| $\widetilde\Phi(H, t)$ | Perturbed constraint | $\hat\Phi_t + \omega_t(\delta) H V_t^{-1/2}$. |
| $B(\xi)$ | Tail bound on $\|\eta\|, \max_i \|H^i\|$ | |
| $B_t$ | Noise radius bound | $\max(1, B(\delta_t))$, where $\delta_t = \delta/(t^2+t)$. |
| $M_t(a)$ | Perturbation scale at $a$ | $B_t \omega_t \|a\|_{V_t^{-1}}$ |
| $a(\eta, H, t)$ | Perturbed optimum | See (3) |
| $\mathsf{U}_t(\delta)$ | Unsaturation event | $\{(\eta, H) : \Delta(a(\eta, H, t)) \leq M_t(a(\eta, H, t)\}$ |
| $\chi$ | Unsaturation rate | |
| $\mathsf{L}_t(\delta)$ | Local optimism event | $\{(\eta, H) : \widetilde\theta(\eta,t)^\top a_* \geq \theta_*^\top a_*, \widetilde\Phi(H,t) a_* \leq \alpha\}$ |
| $\pi$ | Local optimism rate | |
| Coupled Noise Design | | |
| $\nu$ | Baseline perturbation law | Supported on $\mathbb{R}^{d\times 1}$ |
| $\zeta$ | Generic draw from $\nu$ | $\zeta \sim \nu$, independent of $\mathfrak{H}_{t-1}$ |
| $\bar B$ | Tail bound for $\nu$ | $\nu(\|\zeta\| > \bar B(\xi)) \leq \xi$ |
| $p$ | Anticoncentration parameter for $\nu$ | $\inf_u \nu(\zeta^\top u > \|u\|) \geq p$ |
| $(\zeta^\top, -\mathbf{1}_m \zeta^\top)$ | Coupled noise induced by $\nu$ | i.e., draw $\zeta$, set $\eta = \zeta^\top$ and $H = -\mathbf{1}_m \zeta^\top$. |
| S-COLTS | | |
| $a_{\mathsf{safe}}$ | A priori given safe action | $\Gamma(a_{\mathsf{safe}}) > 0$. |
| $\Gamma_0$ | Reference margin (see §H.1) for estimation | $\Gamma_0 \geq \Gamma(a_{\mathsf{safe}})/2$ and $\Gamma_0 \leq \Gamma(a_{\mathsf{safe}})$ |
| $(\eta_t, H_t)$ | Perturbation noise at $t$ | |
| $\widetilde\theta_t, \widetilde\Phi_t$ | Perturbed parameters at $t$ | $\widetilde\theta_t = \widetilde\theta(\eta_t, t), \widetilde\Phi_t = \widetilde\Phi(H_t, t)$ |
| $b_t$ | Preliminary action at time $t$ (if exists) | $b_t = a(\eta_t, H_t, t)$ |
| $\mathfrak{a}(\rho)$ | $\rho$-mixture of $b_t$ and $a_{\mathsf{safe}}$ | $\mathfrak{a}(\rho) = \rho b_t + (1-\rho) a_{\mathsf{safe}}$ |
| $\rho_t$ | Largest $\rho$ with safe $\mathfrak{a}(\rho)$ | See (4); $a_t = \mathfrak{a}(\rho_t)$. |
| $\tau(t)$ | Look-back time | §4.1.2 Lemma 7 |
| E-COLTS | | |
| $(\eta_t, H_t)$ | Perturbation noise draws at time $t$ | $(\eta_t, H_t) \sim t$ |
| $\kappa$ | Goodness factor of exploratory policy | See §I.2 |
| $u_t$ | Number of exploration steps up to time $t$ | $u_t \approx B_t \omega_t \sqrt{dt}$ |
| R-COLTS | | |
| $r$ | Resampling parameter | |
| $I_t$ | Number of resamplings at time $t$ | $I_t = \lceil r \log(1/\delta_t) \rceil + 1$. |
| $(\eta_{i,t}, H_{i,t})$ | $i$th draw of noise perturbation at time $t$ | $\sim \mu$ independently |
| $K(i, t)$ | Value under perturbation | $K(\widetilde\theta(\eta_{i,t}, t), \widetilde\Phi(H_{i,t}, t))$ |
| $i_{*,t}$ | Best index at time $t$ | $\arg\max_i K(i, t)$ |
| $a_t$ | Action picked | $a_t = a(\eta_{i_{*,t}}, H_{i_{*,t}}, t)$ |
| $\widetilde\theta_t$ | Objective for $i_{*,t}$ | $\widetilde\theta_t = \widetilde\theta(\eta_{i_{*,t}}, t)$. |

# B  Examples of Real-World Domains where the Safe Linear Bandit Problem Applies

Table 3: **Mapping real domains to the bandit linear programming.** In all three cases the reward is linear in an unknown parameter vector $\theta_*$, and the safety/fairness predicate is an *unknown linear inequality* $\Phi_* a \leq \alpha$. Feedback noise in both rewards and constraints arises through environmental or individual fluctuations.

| Domain ( ref. ) | Action $a \in \mathcal{A} \subset \mathbb{R}^d$ | Reward $\theta_*^\top a$ + noise | Constraints |
| --- | --- | --- | --- |
| **Dose-finding** [AKR21] | One-hot vector for $d$ discrete dose levels | $\theta_*^i$ = patient-level efficacy probability at dose $i$ | $\Phi_*^i$ = toxicity of dose $i$; constraint so that $P(\text{toxic}|\text{dose}) \leq \alpha$ |
| **Voltage-constrained micro-grid** [FLZY22] | Active/reactive power set-point $[P, Q]^\top$ for each bus | $\theta_*^i$ = locational marginal price vector | $\Phi_*$ = linearised network power-flow imposing nodal-voltage constraints under variable demand |
| **Fair Reccommendation in A/B testing** [Cho+24] | Distribution over $d$ items or policies | $\theta_*^i$ = revenue of item $i$ | $\Phi_*^i$ = encoding group attributes and costs; constraints demand fair exposure for each group |

# C  Further Discussion of Related Work

***Distinction of Safe Bandits From BwK.*** BwK settings are concerned with aggregate cost metrics of the form $\mathbf{A}_T := \max_i (\sum \alpha - \Phi_* a_t)^i$, without the $(\cdot)_+$ nonlinearity in $\mathbf{S}_T$ [e.g. AD14; BKS13]. This simple change has a drastic effect, in that BwK algorithms can 'bank' violation by playing very safe actions for some rounds, and then 'spend' it to gain high reward, without any net penalty in $\mathbf{A}_T$. This is appropriate for modeling aggregate cost constraints (monetary/energy/et c.), but is evidently inappropriate to model safety constraints where feasibility violation in any round cannot be offset by acting safely in another round. Notice that such behaviour is precluded by the ramp nonlinearity in $\mathbf{R}_T, \mathbf{S}_T$: playing too-conservatively does not decrease $\mathbf{S}_T$, while any violation of constraints is accumulated, and similarly, playing suboptimally causes $\mathbf{R}_T$ to rise, but playing an over-aggressive action with negative $\Delta(a)$ does not reduce $\mathbf{R}_T$.

***Pure Exploration in Safe Bandits.*** While our paper focuses on controlling regret and risk, naturally the safe bandit problem can be studied in the pure-exploration sense. These are studied in both the 'soft enforcement' sense, in which case methods can explore both within and outside the feasible region and return actions that are $\varepsilon$-safe and $\varepsilon$-optimal [e.g., Cam+22; KS19], and the 'hard enforcement', wherein exploratory actions must be restricted to the feasible region [e.g., SGBK15; Bot+22].

***More Details on Computational Costs of Prior Methods.*** Most frequentist confidence-set based hard enforcement methods pick actions by solving the program

$$\max_{\theta \in \mathcal{C}_t^\theta, a \in \mathcal{A}} \theta^\top a \text{ s.t. } \forall \Phi \in \mathcal{C}_t^\Phi, \Phi a \leq \alpha.$$

Assuming, for simplicity, that $a_{\mathsf{safe}} = 0$, due to the structure of the confidence sets the above constraint translates to

$$\forall i \in [1:m], \hat{\Phi}_t^i a + \omega_t(\delta)\|a\|_{V_t^{-1}} \mathbf{1}_m \leq \alpha.$$

Notice that this constitutes $m$ different second-order conic constraints. In fact, as discussed in §I.2, we expect $V_t^{-1}$ to have condition number scaling as $\Omega(t^{1/4})$, which adds further computational burdens to optimising under such constraints.

Of course, as written, the above program is nonconvex due to the objective $\theta^\top a$. Indeed, this is a well-established issue in linear bandits (without unknown constraints), and was first observed in this context by Dani et al. [DHK08]. Specifically, notice that due to the ellipsoidal structure of $\mathcal{C}_t^\theta$, even without unknown constraints, the program

$$\max_{\theta, a} \theta^\top a : \theta \in \mathcal{C}_t^\theta, a \in \mathcal{X}$$

is equivalent to solving

$$\max_{a \in \mathcal{X}} \omega_t(\delta)\|a_t\|_{V_t^{-1}} + \hat{\theta}_t^\top a,$$

and in general it is NP-hard to solve such programs [Sah74; PV91]. Indeed, one can see that if $\hat{\theta}_t = 0$, this is exactly equivalent to positive definite quadratic maximisation, which is known to NP-hard even if $V = I$ and $\mathcal{X}$ is only allowed to range over polytopes.[1] Note that with the aforementioned hard enforcement constraints, $\mathcal{X}$ is the intersection of $\mathcal{A}$ and these SOC constraints.

This hardness can be addressed via a standard '$\ell_1$-relaxation [DHK08], which reduces the problem to solving $2d$ optimisation problems with linear objectives and the above SOC constraints, while weakening regret to $\widetilde{O}(\mathcal{R}(a_{\mathsf{safe}})\sqrt{d^3T})$: in a nutshell, one replaces the ellipsoid $\{\|\theta - \hat{\theta}\|_V \leq \omega\}$ by a skewed $\ell_1$ ball of the form
$$\{\|V^{1/2}(\theta - \hat{\theta})\|_1 \leq \sqrt{d}\omega\}.$$
Such a ball has only $2d$ extreme points in $\theta$, and of course for any $a$, one of these extreme points optimises $\theta^\top a$ over this $\ell_1$-ball, meaning that computing $\max_{\theta,a} \theta^\top a$ can be reduced to solving $2d$ programs of the form $\max_a \theta^\top a$. The increase in regret occurs due to the $\sqrt{d}$-blowup, which is needed to ensure that the relaxed confidence set contains the original ellipsoid.

This characterises the costs of most of these 'optimistic-pessimistic' methods [e.g. PGBJ21; PGB24; AAT19]. Afsharrad et al. give a systematic and detailed account of these considerations [AML24]. There are two exceptions. The SAFE-LTS method of Moradipari et al. [MAAT21] uses sampling to select the objective, but still imposes the same SOC constraints, thus needing only one optimisation each round. The ROFUL method of Hutchinson et al.[HTA24] instead first picks an action according to (the NP-hard to implement method) DOSS, and then scales it towards $a_{\mathsf{safe}}$ as in S-COLTS. Of course, note that S-COLTS samples only one set of *linear* constraints each round, and is efficient. There are also analytical differences between ROFUL and S-COLTS, as discussed in §4.

Turning to soft enforcement, as we mentioned in the main text, no efficient method is known. The main method herein for linear bandits is DOSS [GCS24], which instead picks actions by solving
$$\max_{\theta \in \mathcal{C}_t^\theta, a \in \mathcal{A}} \theta^\top a \text{ s.t. } \exists \Phi \in \mathcal{C}_t^\Phi : \Phi a \leq \alpha.$$
This $\exists$ operator renders this problem much more challenging, since now the constraint works out to the union of polytopes
$$\bigcup_{A \in \mathcal{C}_t^\Phi} \mathcal{A} \cap \{\Phi a \leq \alpha\},$$
which is highly nonconvex, and hard to condense or relax. Indeed, Gangrade et al. [GCS24] propose using a similar $\ell_1$-relaxation as discussed above for both the objective and the constraints, but this now leads to $(2d)^{m+1}$-extreme points of the confidence sets (accounting for both $\theta$ and the $m$-rows of $\Phi$), leading to $(2d)^{m+1} \cdot \mathsf{LP} \cdot \log(t)$ compute needed per round. In contrast, R-COLTS uses $\mathsf{LP} \cdot \log^2(t)$ compute, and E-COLTS uses only $\mathsf{LP} \cdot \log(t)$ compute.

***More Details on the Failure of Prior Thompson Sampling Analyses.*** §4 discusses the point where the prior unsaturation-based analysis of linear TS due to [AG13] breaks down in the presence of unknown constraints in some detail. For the optimism-based analysis of [AL17], we only briefly touch upon this in §5, and give a more detailed look in §D. This section serves as a brief summary of the latter.

The analysis of Abeille and Lazaric relies on the convexity of the value function $J(\theta) := \max_{a \in \mathcal{A}} \theta^\top a$ to both analyse the roundwise regret $(\Delta(a_t))$ and to establish the frequency of a certain 'global optimism' event (see §D. With unknown constraints, the corresponding object of interest is the value function $K(\theta, \Phi) := \sup\{\theta^\top a : a \in \mathcal{A}, \Phi a \leq \alpha\}$. This map is *not* convex in $\Phi$, which causes both of these steps to break down. R-COLTS avoids this issue by resampling. It is also possible to give an analysis of S-COLTS (and E-COLTS) within the optimism framework, although this again utilises a scaling trick to bypass the same issue. Of course, we also establish optimism in a convexity-free way by analysing the local behaviour at $a_*$.

***Finding a Feasible Point, and Soft-Enforcement with $O(1)$ Risk.*** Notice that since there are plenty of polynomial time methods for hard enforcement in SLBs (even though the prior methods impose SOC constraints), in principle one can develop efficient soft-enforcement methods with regret scaling inversely in $\max_a \Gamma(a)$ by first discovering an action that has $\Gamma(a) \geq \text{const.} \cdot \max_a \Gamma(a)$, and then

---

[1]Of course, we could also reveal the value of $\hat{\theta}_t^\top a$ at the optimum, and then, irrespective of the value of $\hat{\theta}_t$, turn this into a positive definite quadratic maximisation problem over the set $\mathcal{X} \cap \{\hat{\theta}_t^\top a = v\}$.

plugging this into a hard enforcement method. In this case, the exploration time would be random, but a constant, so the net risk would ostensibly be $O(1)$ as $T$ explodes, far below our $\sqrt{T}$ bounds, making the performance close to that of hard enforcement.[2]

In parallelly conducted research, we (along with coauthors) have recently provided an efficient method for this problem [GPSS25], which extends a previous (inefficient) approach to testing the feasibility of LPs due to us [GGSS24]. This method, termed FAST, is also based on Thompson Sampling, and also utilises the coupled noise design of §4.2 as a core element. However, there are significant differences in the analysis of this method: because the expression $\max_a \Gamma(a) = \max_a \min_i (\alpha - \Phi_* a)^i$ can be expressed as the value of a matrix game over $\mathcal{A}$ and the probability simplex, one can exploit the boundedness of the probability simplex in 1-norm to directly analyse a Lagrangian. We leave the details to the appropriate paper. The net result is thus: let $a_{\mathrm{mm}}$ be the maximiser of the margin $\Gamma(a)$ over $\mathcal{A}$, and let $\Gamma_* := \max_a \Gamma(a)$. Then with high probability FAST finds a point $a_{\mathrm{FAST}}$ such that $\Gamma(a_{\mathrm{FAST}}) \geq \Gamma_*/2$ in $\tau = O(d^3/\Gamma_*^2)$ rounds, which incurring a net safety risk of $O(d^3/\Gamma_*)$.

Nevertheless, let us note that coupling our FAST method with S-COLTS gives the first algorithm which selects actions by only optimising over linear constraints, and gives both $O(d^3/\Gamma_*)$ risk and $\widetilde{O}(\mathcal{R}(a_{\mathrm{FAST}})\sqrt{d^3T}) \leq \widetilde{O}(\Gamma_*^{-1}\sqrt{d^3T})$ regret *without prior knowledge of any safe action such as the* $a_{\mathsf{safe}}$ assumed in this paper, as well as all prior papers on hard constraint enforcement in safe bandits. Naturally, note that if $\max_a \Gamma(a)$ is small, this regret bound can be much weaker than the unconditional $\widetilde{O}(\sqrt{d^3T})$ bound of R-COLTS: which method to prefer depends on the application, and the importance of attaining $O(1)$ instead of $O(\sqrt{d^3T})$ risk. We also note that in principle the regret bound can be tightened beyond this: instead of searching for a maximiser of $\Gamma(a)$, one could directly search for a minimiser of $\mathcal{R}(a) = \Delta(a)/\Gamma(a)$. Finding an efficient and effective way to do this is an interesting open problem.

# D   Local Optimism, Global Optimism, and Unsaturation

In §5, we (implicitly) defined a local-optimism condition on the perturbation law $\mu$ in the statement of Lemma 10, which is compared to a 'global optimism' condition suggested by the prior work of Abeille & Lazaric [AL17]. To further contextualise these, let us explicitly define them.

**Definition 13.** *Let* $K(\theta, \Phi) := \sup\{\theta^\top a : a \in \mathcal{A}, \Phi a \leq \alpha\}$ *denote the value function of optimising the objective* $\theta$ *under constraint matrix* $\Phi$ *over* $\mathcal{A}$*, with the convention that* $\sup \emptyset = -\infty$*. Recall that the* local optimism event *at* $a_*$ *is*

$$\mathsf{L}_t(\delta) := \{(\eta, H) : \widetilde{\theta}(\eta, t)^\top a_* \geq \theta_*^\top a_*, \widetilde{\Phi}(H, t)a_* \leq \alpha\},$$

*where* $a_*$ *is the constrained optimum for the true parameters* $(\theta_*, \Phi_*)$*. Further, define the* global optimism event

$$\mathsf{G}_t(\delta) := \{(\eta, H) : K(\widetilde{\theta}(\eta, t), \widetilde{\Phi}(H, t)) \geq \theta_*^\top a_* = K(\theta_*, \Phi_*)\}.$$

*For* $\pi \in (0, 1]$*, we say that a law* $\mu$ *on* $(\eta, H)$ *satisfies* $\pi$*-local optimism if*

$$\forall t, \mathbb{E}[\mu(\mathsf{L}_t(\delta))|\mathfrak{H}_{t-1}]\mathbb{1}_{\mathsf{Con}_t(\delta)} \geq \pi\mathbb{1}_{\mathsf{Con}_t(\delta)},$$

*and similarly, that* $\mu$ *satisfies* $\pi$*-global optimism if*

$$\forall t, \mathbb{E}[\mu(\mathsf{G}_t(\delta))|\mathfrak{H}_{t-1}]\mathbb{1}_{\mathsf{Con}_t(\delta)} \geq \pi\mathbb{1}_{\mathsf{Con}_t(\delta)}.$$

Notice that $\mathsf{G}$ demands perturbations such that after optimising the perturbed parameters, the value of the resulting program is larger than $\theta_*^\top a_*$, while $\mathsf{L}$ demands the stronger condition that $a_*$ is feasible, and its value increases. Evidently, $\mathsf{L} \subset \mathsf{G}$, and so $\pi$-local optimism of $\mu$ implies $\pi$-global optimism. Naturally, the entirety of §5 follows if we have a globally optimistic $\mu$ instead of locally optimistic $\mu$. We presented this section with $\mathsf{L}_t$ instead due to limited space in the main text.

As discussed in §4.2, we will also show, in §G, $\mathsf{L}_t(\delta) \cap \mathsf{Con}_t(\delta) \subset \mathsf{U}_t(\delta) \cap \mathsf{Con}_t(\delta)$, i.e., when consistency holds, local optimism implies unsaturation. Thus, $\mathsf{L}_t$ links the global-optimism based

---

[2]note that there is a cost, though: as stated before, the regret would scale inversely in the Slater gap, and until the safe point is discovered, would grow linearly.

framework of [AL17], and the unsaturation based framework of [AG13]. Nevertheless, technically, these are distinct events.

Let us briefly note that the prior work [AG13] essentially passes through the same strategy as us when establishing a good unsaturation rate, in that they argue that local-optimism holds frequently (although they do not consider unknown constraints, so their argument does not extend to our setting). On the other hand, [AL17] presents a convexity-based proof of frequent global optimism for linear TS without unknown constraints, while immediately breaks in our setting because $K(\theta, \Phi)$ is nonconvex in $\Phi$. We also reiterate that our coupled noise design of §4.2 essentially takes the same conditions on perturbations used in these prior works, and extends them to produce the *same* bounds on unsaturation or global-optimism rates by arguing that local-optimism holds. This means that these prior results do not capture the prevalence of these events beyond local optimism. Our simulations in §J suggest that this leaves a significant amount of performance on the table, capturing which theoretically would require deeper understanding of $\mathsf{U}_t \setminus \mathsf{L}_t$ and $\mathsf{G}_t \setminus \mathsf{L}_t$.

**Role of These Conditions in Our Work.** To analyse S-COLTS and E-COLTS, we used a look-back approach enabled by the unsaturation condition, while to analyse R-COLTS, we relied on a direct use of the optimism condition. It turns out that the unsaturation condition is not effective at capturing at least our strategy for analysing the resampling-based strategy R-COLTS. The reason is that while the resampling will ensure that at least one of the optima of attaining the various $K(i, t)$ values will be unsaturated, we have no guarantee that the procedure we take of picking the $i_{*,t}$ that maximises $K(i, t)$ will choose an unsaturated action. On the other hand, the optimism condition *can* be used to analyse S-COLTS and E-COLTS directly (see §H.5), but a direct execution of the previous optimism based approach [AL17] fails due to the lack of convexity of the map $K(\theta, \Phi)$. Instead, we have to directly analyse expressions of the form $\mathbb{E}[|K(\widetilde{\theta}, \widetilde{\Phi}) - K(\widetilde{\theta}', \widetilde{\Phi}')| \mid \mathfrak{H}_{t-1}]$, where $(\widetilde{\theta}, \widetilde{\Phi})$ and $(\widetilde{\theta}', \widetilde{\Phi}')$ are iid draws of the perturbation at tie $t$. Under the assumption that there is an action with positive safety margin with small $M_t$, this can be executed via a similar scaling-based analysis, albeit at a loss of some factors in the regret bound (§H.5). In our opinion the unsaturation based look-back analysis of $\Delta(a_t)$ is conceptually clearer, and we chose to present it in the main instead.

Nevertheless, in terms of their explanatory power, neither condition dominates the other. Indeed, in simulations, we find both cases where unsaturation is frequent but global optimism is not, and cases where global optimism is frequent but unsaturation is not.[3] Of course, in our analysis, both of these are connected by local optimism as detailed above, which is rendered frequent through our coupled design. Nevertheless, the local optimism rate can be significantly smaller than the unsaturation and global optimism rates, particularly when the noise is shrunk far below the theoretically analysed setting of $\Theta(\sqrt{d})$-scale noise (see §J). These observations again hint that developing a tight theory of linear TS (both with and without unknown constraints) requires a deeper understanding of the portion of these events that do not intersect with local optimism.

# E   An Informal Discussion of Contextual Safe Linear Bandits

Rather than static bandit problems, most practical scenarios are contextual, wherein the learner observes some side information $x_t$ before choosing an action, and this side information affects the reward and constraint structure at time $t$. A common setting to model this [PGB24; AG13] is to assume that there is a known feature map $\varphi : \mathcal{X} \times \mathcal{A} \to \mathbb{R}^d$ such that the reward and constraints at time $t$ are of the form

$$\theta_*^\top \varphi(x_t, a) \text{ and } \Phi_* \varphi(x_t, a) \le \alpha.$$

Throughout, we assume the same feedback structure, i.e., noisy measurements of $\theta_*^\top \varphi(x_t, a_t)$ and $\Phi_* \varphi(x_t, a_t)$. Naturally, regret is compared to the optimal policy $\mathscr{A}_* : \mathcal{X} \to \mathcal{A}$, where

$$\mathscr{A}_*(x) = \arg\max \theta_*^\top \varphi(x, a) : \Phi_* \varphi(x, a) \le \alpha, a \in \mathcal{A}.$$

It should be noted that the Lemma 1 on consistency, and the elliptical potential lemma (Lemma 14) continue to hold, with $V_t$ replaced by $I + \sum_{s \le t} \varphi(x_s, a_s)\varphi(x_s, a_s)^\top$, and $a_t$ by $\varphi(x_t, a_t)$. Notationally, we extend $\Delta(a), \Gamma(a)$ to $\Delta(x, a) = \theta_*^\top(\varphi(x, \mathscr{A}_*(x)) - \varphi(x, a))$ and $\Gamma(x, a) = \max_i((\alpha - \Phi_* \varphi(x, a))^i)_+$.

---

[3]This is most pertinent for the setting where we drive the perturbations with independent noise, where in §J.3 we observed that the unsaturation rate decayed with $m$, but the global optimism rate did not. Indeed, this is what prompted us to write the optimism-based analysis of §H.5.

A key observation is that our result on the frequency of the local optimism persists in this contextual setting. Under the hood, this essentially shows that at any $t$, and for any vector $\varphi$,

$$\mathbb{P}\left\{(\eta, H) : \widetilde{\theta}^{\top}\varphi \geq \theta_*^{\top}\varphi, \widetilde{\Phi}\varphi \leq \Phi_*\varphi \Big| \mathfrak{H}_{t-1}\right\} \mathbb{1}_{\mathsf{Con}_t(\delta)} \geq \pi \mathbb{1}_{\mathsf{Con}_t(\delta)},$$

where $\pi \geq 0.28$ for the coupled noise driven by $\mathrm{Unif}(\sqrt{3d}\mathbb{S})^d$. Consequently, frequent local optimism follows in the contextual setting by using this result for $\varphi(x_t, \mathscr{A}_*(x_t))$ at time $t$.

The above observation means that using the same coupled noise lets us extend the results of Theorem 11 on the regret of R-COLTS to the contextual case with only cosmetic changes in the analysis. This holds no matter how the sequence $x_t$ is selected, as long as the noise remains conditionally centred and subGaussian given $a_t, x_t$, the algorithmic randomness, and the history. Note, however, that the optimisation over $a$ may become harder due to the feature map $\varphi$, and efficiency requires further structural assumptions on $\varphi$.

Focusing now on S-COLTS, let us first note that if we were given a safe action $a_{\mathsf{safe}}$ that was safe no matter the context, i.e., such that $\inf_x \Gamma(x, a_{\mathsf{safe}}) \geq \Gamma_{\mathsf{safe}} > 0$, and $\varphi$ were 'nice' in terms of $a \in \mathcal{A}$,[4] then as long as we *know* $\Gamma_{\mathsf{safe}}$ a priori, no real change is required, and the guarantees of Theorem 9 for S-COLTS extend to the contextual setting,[5] since we can again guarantee the frequent choice of unsaturated actions through our persistent local optimism property. We note that previous works on safe contextual bandits [PGB24] assume exactly this existence of an 'always very safe' action. Nevertheless, this structure is unrealistic: practically, safety should depend strongly on the context, and it is unlikely that a single action would always be safe, let alone have a large safety margin. A more natural assumption is that instead of a single safe action, we are given a safe policy $\mathscr{A}_{\mathsf{safe}} : \mathcal{X} \to \mathcal{A}$. Here, again, if we know that $\inf_x \Gamma(x, \mathscr{A}_{\mathsf{safe}}(x)) \geq \Gamma_{\mathsf{safe}} > 0$, and we know the value of $\Gamma_{\mathsf{safe}}$, then we are good to go, although this is a strong assumption. Without knowing this value, we need to be able to determine a good estimate of $\Gamma(x_t, \mathscr{A}_{\mathsf{safe}}(x_t))$ in order to appropriately ensure feasiblity of perturbed programs, and to scale back the actions $b_t$. This can be a challenging task, especially if $x_t$ varies in an adversarial way, and structures enabling such estimation must be assumed.[6] Finally, note that even if we were given $\Gamma(x, \mathscr{A}_{\mathsf{safe}}(x))$ as a function explicitly, the easily forthcoming regret bounds rather pessimistically scale with $(\inf_x \Gamma(x, \mathscr{A}_{\mathsf{safe}}(x)))^{-1}$, and do not capture how variation in this margin with $x$ can be used to limit regret. A (at least somewhat) different analysis is needed to express this in a clear way. Resolving such limitations is an important open problem in the theory of SLBs.

This lacuna also affects the E-COLTS method of §I.2, but to a lesser extent. Sticking with 'nice' feature maps, again, if there *exists* an action that is always safe, i.e., if $\max_a \min_x \Gamma(x_t, a_t) > 0$, then the guarantees of Theorem 12 extend with arbitrary context sequence. Without this guarantee, the main gap is the exploration policy being utilised, which must be adapted to attain a good coverage over $\{\varphi(x, a)\}$ even as $x_t$ varies. Given such a policy, however, the results of Theorem 12 again extend to the contextual case with arbitrary $x_t$.

## F    Some Basic Tools For the Analysis

We begin with some standard tools that are repeatedly utilised in the analysis. The first of these, termed the *elliptical potential lemma* offers generic control on the accumulation of $\|a_t\|_{V_t^{-1}}$.

**Lemma 14.** *[APS11; CVA20] For any sequence of actions $\{a_t\} \subset \{\|a\| \leq 1\}$, and any $t$,*

$$\sum_{s \leq t} \|a_s\|_{V_s^{-1}}^2 \leq 2d \log(1 + t/d), \text{ and } \sum_{s \leq t} \|a_s\|_{V_s^{-1}} \leq \sqrt{2dt \log(1 + t/d)}.$$

*Further, for all $t, \delta, \omega_t(\delta) \leq 1 + \sqrt{1/2 \log((m+1)/\delta) + d/2 \log(1 + t/d)}$.*

We further explicitly write the following instantiation of the Cauchy-Schwarz inequality pertinent to our setting.

---

[4]We essentially need a way to efficiently select an action $a$ such that $\varphi(x_t, a) = \rho\varphi(x_t, b_t) + (1 - \rho)\varphi(x_t, a_{\mathsf{safe}})$, so that safety can still be attained by mixing with $a_{\mathsf{safe}}$.

[5]upto replacing $\Delta(a_{\mathsf{safe}})$ by 1

[6]For instance, if $x_t$ were drawn in some static randomised way, and $\Gamma$ were sufficiently simple, then we could learn $\Gamma(x, \mathscr{A}_{\mathsf{safe}}(x))$ using regression techniques.

**Lemma 15.** *For any positive definite matrix $V$. For pair of tuples $(\theta, \Phi)$ and $(\widetilde{\theta}, \widetilde{\Phi})$ lying in $\mathbb{R}^d \times \mathbb{R}^{m \times d}$ and any $a \in \mathbb{R}^d$, it holds that*

$$\max\left(|(\theta - \widetilde{\theta})^\top a|, \max_i |(\Phi^i - \widetilde{\Phi}^i)a|\right) \le \max(\|\widetilde{\theta} - \theta\|_V, \max_i \|\widetilde{\Phi}^i - \Phi^i\|_V) \cdot \|a\|_{V^{-1}}.$$

*Proof.* Notice that $(\widetilde{\theta} - \theta)^\top a = (\widetilde{\theta} - \theta)^\top V^{1/2} V^{-1/2} a \le \|(V^{1/2}(\widetilde{\theta} - \theta)\| \cdot \|V^{-1/2}a\|$. The claim follows by first repeating the same observation for each $(\Phi^i - \widetilde{\Phi}^i)$ (adjusting for the fact that these are row-vectors), and then recalling that (for column vectors) $\|a\|_M = \|M^{1/2}a\|$ by definition. $\square$

This immediately yields a proof of the concentration statement of Lemma 3, which motivated the definition of $M_t(a)$.

*Proof of Lemma 3.* Notice that by a union bound

$$\mathbb{P}(\exists t : \max(\|\eta_t\|, \max_i \|H_t^i\|) > B(\delta_t)) \le \sum_t \delta_t = \delta.$$

Now assume that $\max(\|\eta_t\|, \max_i \|H_t^i\|) \le B(\delta_t)$, and that the consistency event $\mathsf{Con}_t(\delta)$ holds. Then, via the triangle inequality,

$$\|\widetilde{\theta}(\eta_t, t) - \theta_*\|_{V_t} \le \|\widetilde{\theta}(\eta_t, t) - \hat{\theta}_t\|_{V_t} + \|\hat{\theta}_t - \theta_*\|_{V_t}.$$

Of course, given $\mathsf{Con}_t(\delta)$, the second term is smaller than $\omega_t(\delta)$. For the first, expanding the definition of $\widetilde{\theta}(\cdot, \cdot)$, we find that

$$\|\widetilde{\theta}(\eta_t, t) - \hat{\theta}_t\|_{V_t} = \omega_t(\delta)\|\eta_t V_t^{-1/2}\|_{V_t} = \|\omega_t(\delta)\eta_t V_t^{-1/2} \cdot V_t^{1/2}\| \le \omega_t(\delta)\|\eta_t\|,$$

and of course, $\|\eta_t\| \le B(\delta_t)$ by our assumption above. Thus, given the concentration assumption on $\|\eta_t\|$s and $\mathsf{Con}_t(\delta)$, for any $t$, it holds that

$$\|\widetilde{\theta}(\eta_t, t) - \theta_*\|_{V_t} \le (1 + B(\delta_t))\omega_t(\delta) \le B_t\omega_t(\delta).$$

Of course, entirely the same applies to $\|\widetilde{\Phi}(H_t, t)^i - \Phi_*^i\|_{V_t}$, with $\eta$ replaced by $H_t^i$. The claim now follows by Lemma 15 and the fact that $\mathsf{Con}(\delta) := \bigcap \mathsf{Con}_t(\delta)$ has chance at least $1 - \delta$. $\square$

## G   Analysis of the Coupled Noise Design

We will first execute the strategy described in §4.2 to show that under the conditions of Lemma 8, local optimism is frequent. We will then use this to show the frequency of unsaturation.

**Lemma 16.** *Let $p \in (0, 1]$, and let $\nu$ be a law on $\mathbb{R}^{d \times 1}$ such that*

$$\forall u \in \mathbb{R}^d, \nu(\{\zeta : \zeta^\top u \ge \|u\|\}) \ge p.$$

*Let $\mu$ be the the pushforward of $\nu$ under the map $\zeta \mapsto (\zeta^\top, -\mathbf{1}_m\zeta^\top)$. Then, for all $t$, $\mathbb{1}_{\mathsf{Con}_t(\delta)}\mathbb{E}[\mu(\mathsf{L}_t(\delta))|\mathfrak{H}_{t-1}] \ge p\mathbb{1}_{\mathsf{Con}_t(\delta)}$, where $\mathsf{L}_t(\delta)$ is the local optimism event (6).*

*Proof.* Observe that under a draw from $\mu$, for all $t$, we have

$$\widetilde{\theta}^\top := (\widetilde{\theta}(\eta, t))^\top = \hat{\theta}_t^\top + \omega_t(\delta)\zeta^\top V_t^{-1/2}$$
$$\widetilde{\Phi} := \widetilde{\Phi}(H, t) = \hat{\Phi}_t - \mathbf{1}_m(\omega_t(\delta)\zeta^\top V_t^{-1/2}).$$

Further, recall that if the event $\mathsf{Con}_t(\delta)$ occurs, then, for all $a$,

$$\hat{\theta}_t^\top a \ge \theta_*^\top a + \omega_t(\delta)\|V_t^{-1/2}a\|, \text{ and } \hat{\Phi}_t a \le \Phi_* a + \mathbf{1}_m(\omega_t(\delta)\|V_t^{-1/2}a\|),$$

where we have the Cauchy-Schwarz inequality, and the fact that $\|a\|_{V_t^{-1}} = \|V_t^{-1/2}a\|$. Thus, assuming $\mathsf{Con}_t(\delta)$, for any action $a$, we find that

$$\widetilde{\theta}^\top a \ge \theta_*^\top a + \omega_t(\delta)\left(\zeta^\top V_t^{-1/2}a - \|V_t^{-1/2}a\|\right),$$
$$\widetilde{\Phi}a \ge \Phi_* a + \mathbf{1}_m\omega_t(\delta)\left(\zeta^\top V_t^{-1/2}a - \|V_t^{-1/2}a\|\right).$$

Now, set $a = a_*$, and suppose that $\zeta^\top V_t^{-1/2} a_* \geq \|V_t^{-1/2} a_*\|$. Then we can conclude that

$$\widetilde{\theta}^\top a_* \geq \theta_*^\top a_* \text{ and } \widetilde{\Phi} a_* \leq \Phi_* a_* \leq \alpha,$$

the final inequality holding since $a_*$ is of course feasible for the program it optimises. Of course, by definition, this means that the ensuing noise $\eta, H$ lie in the event $\mathsf{L}_t(\delta)$

Now, it only remains to argue that $\zeta^\top V_t^{-1/2} a_* \geq \|\zeta^\top V_t^{-1/2} a_*\|$ happens with large chance given $\mathfrak{H}_{t-1}$. But notice that both $V_t^{-1/2}$ and (the constant) $a_*$ are $\mathfrak{H}_{t-1}$-measurable, and so are constant given it. It follows thus that

$$\mathbb{E}[\nu(\{\zeta : \zeta^\top V_t^{-1/2} a_* > \|V_t^{-1/2} a_*\|\}) \mid \mathfrak{H}_{t-1}] \geq \inf_{u \in \mathbb{R}^d} \nu(\{\zeta^\top u > \|u\|\}) \geq p. \qquad \square$$

To finish the proof of frequent unsaturation, we only need to determine that this local optimism induces unsaturation in the actions.

*Proof of Lemma 8.* Fix a $t$, and assume consistency. Suppose that $\max(\|\eta_t\|, \max_i \|H_t^i\|) \leq B(\delta_t)$. Note that given $\mathsf{Con}_t(\delta)$, this with chance at least $1 - \delta_t$. As a consequence, for any action $a \in \mathfrak{S}_t := \{a : \Delta(a) > M_t(a)\}$, by following the proof of Lemma 3 we can conclude that

$$\widetilde{\theta}(\eta, t)^\top a \leq \theta_*^\top a + M_t(a) = \theta_*^\top a_* - \Delta(a) + M_t(a) < \theta_*^\top a_*.$$

Now, suppose that the drawn $\zeta$ induces local optimism. We claim that then all saturated actions are suboptimal. Indeed, by the above, each unsaturated action satisfies $\widetilde{\theta}(\eta, t)^\top a < \theta_*^\top a_*$. But $\widetilde{\theta}(\eta, t)^\top a_* \geq \theta_*^\top a_*$, and further $\widetilde{\Phi}(H, t) a_* \leq \alpha$, means that there is an action that is feasible for the perturbed program with value strictly larger than that attained by any saturated action, i.e., any member of $\mathfrak{S}_t$. It thus follows that the optimum $a(\eta, H, t) \in \mathfrak{S}_t^c = \{a : \Delta(a) \leq M_t(a)\}$.

Now, we know from Lemma 16 that given $\mathfrak{H}_{t-1}$, our assumptions of $\mathsf{Con}_t(\delta)$ and the norm-control on $\|\eta_t\|, \max_i \|H_t^i\|$ imply that local optimism occurs with chance at least $p$. Since these events occur with chance at least $1 - \delta_t$, this means that unsaturation occurs with chance at least $p - \delta_t$. Since definition 4 restricts attention to $t : \delta_t \leq p/2$, the statement follows. $\qquad \square$

### G.1 Bounds for Simple Reference Laws

We argue that both the standard Gaussian, and the uniform law of the sphere of radius $\sqrt{3d}$ yield effective noise distributions for our coupled design.

For the Gaussian, recall that if $Z \sim \mathcal{N}(0, I_d)$, then $\|Z\|^2$ is distributed as a $\chi^2$-random variable. A classical subexponential concentration argument [e.g. LM00, Lemma 1] yields that for any $x$,

$$\mathbb{P}(\|Z\|^2 \geq d + 2\sqrt{dx} + 2x) \leq e^{-x}.$$

Note that $(d + 2\sqrt{dx} + 2x) \leq (\sqrt{d} + \sqrt{2x})^2$, and hence taking $x = \log(1/\xi)$ in the above yields that $B(\xi) \leq \sqrt{d} + \sqrt{2 \log(1/\xi)}$. Further, due to the isotropicity of $Z$, $Z^\top u/\|u\| \overset{\text{law}}{=} Z_1 \sim \mathcal{N}(0, 1)$, and thus $\pi \geq 1 - \Phi(1) \geq 0.158 \ldots$.

Further, notice that if $Z \sim \mathcal{N}(0, I_d)$, then $Y := \sqrt{3d} Z/\|Z\| \sim \mathrm{Unif}(\sqrt{3d} \cdot \mathbb{S}^d)$, and by isotropicity, for any $u$, $Y^\top u/\|u\| \overset{\text{law}}{=} Y_1$. As a result,

$$\mathbb{P}(Y^\top u/\|u\| \geq 1) = \mathbb{P}(Y_1 \geq 1) = \frac{1}{2}\mathbb{P}(Y_1^2 \geq 1)$$

$$= \frac{1}{2}\mathbb{P}((3d - 1)Z_1^2 \geq \sum_{i=2}^d Z_i^2) \geq \frac{1}{2}\mathbb{P}(Z_1^2 \geq 1) \cdot \mathbb{P}(\sum_{i=2}^d Z_i^2 \geq 3d - 1).$$

But notice that $d - 1 + 2\sqrt{(d-1) \cdot d/3} + 2d/3 \leq 3d - 1$, and thus, $\mathbb{P}(\sum_{i=2}^d Z_i^2 \geq 3d - 1) \leq \exp(-d/3)$. Invoking the bound on $\mathbb{P}(Z_1 \geq 1) = \frac{1}{2}\mathbb{P}(|Z_1| \geq 1)$ above, we conclude that $\pi \geq 0.15 \cdot (1 - e^{-d/3})$. Of course, $\|Y\| = \sqrt{3d}$ surely, giving the $B$ expression.

We note that while the above only shows a $0.15(1 - e^{-d/3})$ bound on the anticoncentration of the uniform law on $\sqrt{3d}\mathbb{S}^d$, it is a simple matter of simulation to find that this is actually larger than

0.28 for all $d$ - for small dimensions, the bound turns out to be very loose, while as $d$ diverges, this converges from above towards the chance that a standard Gaussian exceeds $1/\sqrt{3}$, which is $0.2818\ldots$.

# H  The Analysis of S-COLTS

We move on to the analysis of S-COLTS. Before proceeding, we recall that in our presentation of S-COLTS in Algorithm 1, we assumed access to a quantity $\Gamma_0 \in [\Gamma(a_{\mathsf{safe}})/2, \Gamma(a_{\mathsf{safe}})]$. We will first address how to obtain such a quantity by repeatedly playing $a_t = a_{\mathsf{safe}}$, and characterise how long this takes. For completenesss, the cost of this will be incorporated into our regret bound.

Beyond this, we need to characterise the subsequent time spent playing $a_{\mathsf{safe}}$ due to $M_t(a_{\mathsf{safe}})$ being large, and to prove the look-back bound of Lemma 7, along with the characterisation of $\sum M_{\tau(t)}(a_{\tau(t)})$ offered in Lemma 5. We will analyse these results in order, and finally show Theorem 9 using these results.

## H.1  Identifying $\Gamma_0$ and Sampling Rate of $a_{\mathsf{safe}}$

We first discuss the determination of $\Gamma_0$. There are two main points to make: how to ensure a correct value of $\Gamma_0$, and how many rounds of exploration this costs. To this end, we first recall the following nonasymptotic law of iterated logarithms [e.g. HRMS21].

**Lemma 17.** *Let $\{\mathfrak{F}_t\}$ be a filtration, and let $\{\xi_t\}$ be a process such that each $\xi_t$ is $\mathfrak{F}_t$-measurable, and is further conditionally centred and 1-subGaussian given $\mathfrak{F}_{t-1}$. Then*

$$\forall \delta \in (0,1], \mathbb{P}(\exists t : |Z_t| > \mathrm{LIL}(t, \delta)) \leq \delta,$$

*where $Z_t := \sum_{s \leq t} \xi_t$, and*

$$\mathrm{LIL}(t, \delta) := \sqrt{4t \log \frac{\max(1, \log(t))}{\delta}}.$$

With this in hand, the determination of $\Gamma_0$ proceeds thus: we repeatedly play $a_{\mathsf{safe}}$, and maintain the running average $\mathrm{Av}_t = \sum_{s \leq t}(\alpha - S_s)/t$. Further, we maintain the upper and lower bounds

$$u_t^i := \mathrm{Av}_t + \mathrm{LIL}(t, \delta/m)/t, \ell_t^i := \mathrm{Av}_t - \mathrm{LIL}(t, \delta)/t.$$

We stop at the first time when $\forall i, \ell_t^i \geq u_t^i/2$, and set $\Gamma_0 = \min_i \ell_t^i$. This stopping time is denoted $T_0$.

Let us first show that this procedure is correct, and bound the size of $T_0$.

**Lemma 18.** *Under the procedure specified above, it holds with probability at least $1 - \delta$ that*

$$\Gamma_0 \in [\Gamma(a_{\mathsf{safe}})/2, \Gamma(a_{\mathsf{safe}})]$$

*and that*

$$T_0 \leq \frac{8}{\Gamma(a_{\mathsf{safe}})^2} \log(8/(\delta \Gamma(a_{\mathsf{safe}})^2))$$

*Proof.* Notice that we can write

$$\mathrm{Av}_t = \alpha - \Phi_* a_{\mathsf{safe}} + \sum_{s \leq t} w_s^S/t.$$

For succinctness, let us write $\Gamma = \alpha - \Phi_* a_{\mathsf{safe}}$. Now, by our assumption on the noise $w_t^S$, we observe that each coordinate of $w_t^S$ constitutes an adapted, centred, and 1-subGaussian process. Applying Lemma 17 along with a union bound over the coordinates then tells us that with probability at least $1 - \delta$,

$$\forall t, |\mathrm{Av}_t - \Gamma| \leq \mathrm{LIL}(t, \delta/m)/t \cdot \mathbf{1}_m.$$

As a consequence, at all $t$, we have

$$u_t \geq \Gamma \geq \ell_t,$$

where $u_t$ is the vector with $i$th coordinate $u_t^i$, and similarly for $\ell_t$. It follows thus that at the stopping time $T_0$,

$$\forall i, \ell_{T_0}^i \geq u_{T_0}^i/2 \implies \ell_{T_0} \geq \Gamma/2.$$

Of course, a fortiori, it follows that $\Gamma_0 = \min_i \ell_t^i \geq \min_i \Gamma^i/2 = \Gamma(a_{\mathsf{safe}})/2$. Further, of course, $\Gamma_0 \leq \Gamma(a_{\mathsf{safe}})$ follows as well, since $\forall t \min_i \ell_t^i \leq \min_i(\Gamma^i) = \Gamma(a_{\mathsf{safe}})$.

It only remains to control $T_0$. To this end, notice that for all $t$

$$\ell_t = \mathrm{Av}_t - \mathrm{LIL}(t, m/\delta)/t \cdot \mathbf{1}_m \geq \Gamma - 2\mathrm{LIL}(t, m/\delta)/t \cdot \mathbf{1}_m,$$

and similarly,

$$u_t \leq \Gamma + 2\mathrm{LIL}(t, m/\delta)/t \cdot \mathbf{1}_m.$$

Of course, then $\ell_t^i > u_t^i/2$ for all $t$ such that

$$\forall i, \Gamma^i - \mathrm{LIL}(t, m/\delta)/t \geq \Gamma^i/2 + \mathrm{LIL}(t, m/\delta)/2t \iff \Gamma^i > 3\mathrm{LIL}(t, m/\delta)/t.$$

It follows thus that

$$T_0 \leq \inf\{t : t\Gamma(a_{\mathsf{safe}}) \geq 3\mathrm{LIL}(t, m/\delta)\}.$$

By a simple inversion, this can be bounded as

$$T_0 \leq \inf\{t : t > 8/\Gamma(a_{\mathsf{safe}})^2 \log(1/\delta) \text{ and } t > 8/\Gamma(a_{\mathsf{safe}})^2 \log(1 + \log(t))\},$$

which is bounded as

$$T_0 \leq \frac{8}{\Gamma(a_{\mathsf{safe}})^2} \log(8/(\delta\Gamma(a_{\mathsf{safe}})^2)). \qquad \square$$

***Number of Times*** $a_{\mathsf{safe}}$ ***is sampled after*** $T_0$**.** Given the behaviour of $\Gamma_0$ above, we can further bound the number of times $a_{\mathsf{safe}}$ is played after determining $\Gamma_0$.

**Lemma 19.** *For any $\Gamma_0 > 0$, and $T$, the number of times* S-COLTS *plays $a_{\mathsf{safe}}$ because $M_t(a_{\mathsf{safe}}) > \Gamma_0/3$ is bounded as* $\frac{9\omega_T^2 B_T^2}{\Gamma_0^2} + 1$.

*Proof.* Let $n_t$ denote the total number of times $a_{\mathsf{safe}}$ has been played up to time $t$. Then, of course, $V_t \succcurlyeq I + n_t a_{\mathsf{safe}} a_{\mathsf{safe}}^\top$. Now, recall that for symmetric positive definite matrices $A, B$, it holds that $A \succcurlyeq B \iff B^{-1} \succcurlyeq A^{-1}$.[7] Thus, we have

$$M_t(a_{\mathsf{safe}}) \leq \omega_t B_t \sqrt{a_{\mathsf{safe}}^\top (I + n_t a_{\mathsf{safe}} a_{\mathsf{safe}}^\top)^{-1} a_{\mathsf{safe}}}.$$

Now, by the Sherman-Morrisson formula,

$$a_{\mathsf{safe}}(I + n_t a_{\mathsf{safe}} a_{\mathsf{safe}}^\top)^{-1} a_{\mathsf{safe}} = \|a_{\mathsf{safe}}\|^2 - \frac{a_{\mathsf{safe}}^\top (n_t a_{\mathsf{safe}} a_{\mathsf{safe}}^\top) a_{\mathsf{safe}}}{1 + n_t \|a_{\mathsf{safe}}\|^2} = \frac{\|a_{\mathsf{safe}}\|^2}{1 + n_t \|a_{\mathsf{safe}}\|^2} \leq \frac{1}{n_t}.$$

It follows thus that

$$M_t(a_{\mathsf{safe}}) \leq \frac{\omega_t B_t}{\sqrt{n_t}}.$$

Thus $M_t(a_{\mathsf{safe}}) > \Gamma_0/3$ if and only if

$$n_t \leq \frac{9\omega_t^2 B_t^2}{\Gamma_0^2}.$$

Of course, each time this occurs, $n_t$ is increased by one. Consequently, the number of times $a_{\mathsf{safe}}$ is played by time $t$ is at most

$$\frac{9\omega_T^2 B_T^2}{\Gamma_0^2} + 1. \qquad \square$$

Note that since $(\omega_T B_T)^2 = \Theta(d^2 + d\log(m/\delta))$ with our choice of the coupled noise driven by $\mathrm{Unif}(\sqrt{3d}\mathbb{S}^d)$, the bound above due to playing $a_{\mathsf{safe}}$ due to too large an $M_t(a_{\mathsf{safe}})$ outstrips the bound on $T_0$ above as long as $\log(1/\Gamma(a_{\mathsf{safe}})) = o(d^2)$, as is to be expected.

---

[7]In more technical terms, inversion is monotone decreasing in the Loewner sense. A simple way to see this is to define $C = B^{-1/2}AB^{-1/2}$. Then $A \succcurlyeq B \implies C \succcurlyeq I$ (really iff), since for any $x$, $(B^{-1/2}x)^\top A(B^{-1/2}x) \geq (B^{-1/2}x)^\top B(B^{-1/2}x) \iff x^\top Cx \geq x^\top x$. Using this for $y = C^{-1/2}x$ then gives $x^\top x = (C^{-1/2}x)^\top C(C^{-1/2}x) \geq (C^{-1/2}x)^\top (C^{-1/2}x) = x^\top C^{-1}x$. Since $C^{-1} = B^{1/2}A^{-1}B^{1/2}$ (direct multiplication), the same trick yields $x^\top B^{-1}x = (B^{-1/2}x)^\top (B^{-1/2}x) \geq x^\top B^{-1/2}(B^{1/2}A^{-1}B^{1/2})B^{-1/2}x$, or in other words, $B^{-1} \succcurlyeq A^{-1}$.

## H.2 Proof of the Look-Back Bound

The main text provides a brief sketch of the approach. We will flesh out these details, as well as fill in the omitted aspects of the bound. To this end, we first state a result lower bounding $\rho_t$. Note that the second half of this result implies Lemma 6.

**Lemma 20.** *Assume that $\Gamma_0 \in [\Gamma(a_{\mathsf{safe}})/2, \Gamma(a_{\mathsf{safe}})]$, and that both $\mathsf{Con}(\delta) = \bigcap \mathsf{Con}_t(\delta)$ and the event of Lemma 3 hold true. Then for all $t$ such that $M_t(a_{\mathsf{safe}}) \leq \Gamma_0/3$, it holds that*

$$\rho_t \geq \frac{\Gamma(a_{\mathsf{safe}})}{\Gamma(a_{\mathsf{safe}}) + 3M_t(b_t)}$$

*and*

$$\rho_t \geq \frac{2M_t(a_{\mathsf{safe}})}{2M_t(a_{\mathsf{safe}}) + M_t(b_t)}.$$

*A fortiori, each of the following bounds is true:*

$$(1 - \rho_t)M_t(a_{\mathsf{safe}}) \leq M_t(a_t),$$
$$\rho_t M_t(b_t) \leq 2M_t(a_t), \text{ and}$$
$$(1 - \rho_t)\Gamma(a_{\mathsf{safe}}) \leq 6M_t(a_t).$$

*Proof.* Recall that $\rho_t$ is the largest $\rho$ in $[0, 1]$ such that

$$\hat{\Phi}_t(\rho b_t + (1 - \rho)a_{\mathsf{safe}}) + \omega_t(\delta)\|\rho b_t + (1 - \rho)a_{\mathsf{safe}}\|_{V_t^{-1}}\mathbf{1}_m \leq \alpha.$$

So, if we demonstrate a $\rho_0 \leq 1$ that satisfies this inequality, then $\rho_t \geq \rho_0$.

First note that under the assumption $M_t(a_{\mathsf{safe}}) \leq \Gamma_0/3$, we know that

$$\widetilde{\Phi}_t a_{\mathsf{safe}} \leq \alpha - \Gamma(a_{\mathsf{safe}})\mathbf{1}_m + \Gamma_0/3 \cdot \mathbf{1}_m \leq \alpha - 2\Gamma(a_{\mathsf{safe}})/3 \cdot \mathbf{1}_m,$$

and thus $b_t$ exists since the program defining it is feasible. Now,

$$\hat{\Phi}_t a_{\mathsf{safe}} + \omega_t(\delta)\|a_{\mathsf{safe}}\|_{V_t^{-1}}\mathbf{1}_m = \hat{\Phi}_t a_{\mathsf{safe}} + \frac{M_t(a_{\mathsf{safe}})}{B_t}\mathbf{1}_m$$

$$\leq \alpha - \Gamma(a_{\mathsf{safe}})\mathbf{1}_m + \frac{2M_t(a_{\mathsf{safe}})}{B_t}\mathbf{1}_m \leq \alpha - \frac{2\Gamma(a_{\mathsf{safe}})}{3}\mathbf{1}_m,$$

using the consistency of the confidence sets (and the Cauchy-Schwarz inequality), along with the fact that $B_t = 1 + \max(1, B(\delta_t)) \geq 2$. Further,

$$\hat{\Phi}_t b_t + \omega(\delta)\|b_t\|_{V_t^{-1}} \leq \widetilde{\Phi}_t b_t + \frac{B_t - 1}{B_t}M_t(b_t)\mathbf{1}_m + \frac{1}{B_t}M_t(b_t)\mathbf{1}_m \leq \alpha + M_t(b_t)\mathbf{1}_m.$$

Therefore,

$$\hat{\Phi}_t(\rho b_t + (1 - \rho)a_{\mathsf{safe}}) + \omega_t(\delta)\|\rho b_t + (1 - \rho)a_{\mathsf{safe}}\|_{V_t^{-1}}$$

$$\leq \rho\left(\hat{\Phi}_t b_t + \frac{M_t(b_t)}{B_t}\mathbf{1}_m\right) + (1 - \rho)\left(\hat{\Phi}_t a_{\mathsf{safe}} + \frac{M_t(a_{\mathsf{safe}})}{B_t}\mathbf{1}_m\right)$$

$$\leq \alpha + (\rho M_t(b_t) - (1 - \rho)\Gamma(a_{\mathsf{safe}})/3)\mathbf{1}_m.$$

It is straightforward to find that the additive term above is nonpositive for $\rho_0 = \frac{\Gamma(a_{\mathsf{safe}})}{\Gamma(a_{\mathsf{safe}}) + 3M_t(b_t)}$, and thus $\rho_t \geq \frac{\Gamma(a_{\mathsf{safe}})}{\Gamma(a_{\mathsf{safe}}) + 3M_t(b_t)}$.

Further, since $M_t(a_{\mathsf{safe}}) \leq \Gamma(a_{\mathsf{safe}})/3$, we also have

$$\alpha - 2\Gamma(a_{\mathsf{safe}})/3 \leq \alpha - 2M_t(a_{\mathsf{safe}}).$$

Thus, we can also write

$$\hat{\Phi}_t a_{\mathsf{safe}} + M_t(a_{\mathsf{safe}})/B_t\mathbf{1}_m \leq \alpha - 2M_t(a_{\mathsf{safe}})\mathbf{1}_m,$$

and carrying out the same procedure then shows that

$$\rho_t \geq \frac{2M_t(a_{\mathsf{safe}})}{2M_t(a_{\mathsf{safe}}) + M_t(b_t)}.$$

To draw the final conclusions, first observe that

$$1-\rho_t \leq \frac{M_t(b_t)}{2M_t(a_{\mathsf{safe}}) + M_t(b_t)} \implies 2(1-\rho_t)M_t(a_{\mathsf{safe}}) \leq \rho_t M_t(b_t) \leq M_t(a_t)+(1-\rho_t)M_t(a_{\mathsf{safe}}),$$

where we used the fact that $\rho_t b_t = a_t - (1 - \rho_t)a_{\mathsf{safe}}$, and that $M_t$ is a scaling of a norm. It follows that $(1 - \rho_t)M_t(a_{\mathsf{safe}}) \leq M_t(a_t)$, and of course, that $\rho_t M_t(b_t) \leq 2M_t(a_t)$. Further, by a similar calculation,

$$(1 - \rho_t) \leq \frac{3M_t(b_t)}{\Gamma(a_{\mathsf{safe}}) + 3M_t(b_t)} \implies (1 - \rho_t)\Gamma(a_{\mathsf{safe}}) \leq 3\rho_t M_t(b_t) \leq 6M_t(a_t). \qquad \square$$

***Proving the Look-Back Bound.*** The above control on $(1 - \rho_t)$ is natural in light of terms of the form $(1 - \rho_t)\Delta(a_{\mathsf{safe}})$ appearing in the bound as sketched in the main text. Let us now complete this argument.

*Proof of Lemma 7.* We assume $\Gamma_0 \in [\Gamma(a_{\mathsf{safe}})/2, \Gamma(a_{\mathsf{safe}})]$, and that the event of Lemma 3 holds, as well as $\mathsf{Con}(\delta)$. Together these occur with chance at least $1 - 3\delta$.

Now, we begin as in the main text, by observing that

$$\Delta(a_t) = \Delta(\rho_t b_t + (1 - \rho_t)a_{\mathsf{safe}}) = \rho_t\Delta(b_t) + (1 - \rho_t)\Delta(a_{\mathsf{safe}}).$$

Let $s < t$ be such that $M_s(a_{\mathsf{safe}}) \leq \Gamma_0/3$ as well. Then we further know that

$$\widetilde{\Phi}_s b_s \leq \alpha \implies \widetilde{\Phi}_t b_s \leq \alpha + (M_t(b_s) + M_s(b_s))\mathbf{1}_m.$$

As a consequence, for

$$\sigma_{s \to t} := \frac{\Gamma(a_{\mathsf{safe}})}{\Gamma(a_{\mathsf{safe}}) + 3(M_t(b_s) + M_s(b_s))},$$

we have

$$\widetilde{\Phi}_t(\sigma_{s \to t}b_s + (1 - \sigma_{s \to t})a_{\mathsf{safe}}) \leq \alpha + \left(\sigma_{s \to t}(M_t(b_s) + M_s(b_s)) - \frac{2(1 - \sigma_{s \to t})\Gamma(a_{\mathsf{safe}})}{3}\right)\mathbf{1}_m \leq \alpha.$$

Define $\bar{b}_{s \to t} = \sigma_{s \to t}b_s + (1 - \sigma_{s \to t})a_{\mathsf{safe}}$. By the above observation, $\bar{b}_{s \to t}$ is feasible for $\widetilde{\Phi}_t$, and therefore $\widetilde{\theta}_t^\top \bar{b}_{s \to t} \leq \widetilde{\theta}_t^\top b_t$. To use this, we note that

$$\Delta(b_t) = \Delta(\bar{b}_{s \to t}) + \theta_*^\top(\bar{b}_{s \to t} - b_t) = \Delta(\bar{b}_{s \to t}) + \widetilde{\theta}_t^\top(\bar{b}_{s \to t} - b_t) + (\widetilde{\theta}_t - \theta_*)^\top(\bar{b}_{s \to t} - b_t)$$
$$\leq \Delta(\bar{b}_{s \to t}) + \widetilde{\theta}_t^\top(\bar{b}_{s \to t} - b_t) + M_t(\bar{b}_{s \to t}) + M_t(b_t),$$

where we first use Lemma 3, and then bound $M_t(\bar{b}_{s \to t} - b_t)$ by using the fact that $M_t$ is a norm. The second term above is of course nonpositive, and so can be dropped while retaining the upper bound. Further,

$$\Delta(\bar{b}_{s \to t}) = \sigma_{s \to t}\Delta(b_s) + (1 - \sigma_{s \to t})\Delta(a_{\mathsf{safe}}).$$

This leaves us with the bound

$$\Delta(a_t) \leq (1 - \rho_t + \rho_t(1 - \sigma_{s \to t}))\Delta(a_{\mathsf{safe}})$$
$$+ \rho_t\left(\sigma_{s \to t}\Delta(b_s) + M_t(b_t) + \sigma_{s \to t}M_t(b_s) + (1 - \sigma_{s \to t})M_t(a_{\mathsf{safe}})\right),$$

where we used the triangle inequality and the fact that $M_t$ is a scaling of a norm to write the final two terms. We will, of course, evaluate this at $s = \tau(t)$. In the subsequent, we will just write $\tau$ instead of $\tau(t)$ for the sake of reducing the density of notation. Using the fact that $\Delta(b_\tau) \leq M_\tau(b_\tau)$, we set up the basic bound

$$\Delta(a_t) \leq (1 - \rho_t + \rho_t(1 - \sigma_{\tau \to t}))\Delta(a_{\mathsf{safe}})$$
$$+ \rho_t M_t(b_t) + \rho_t(\sigma_{\tau \to t}(M_\tau(b_\tau) + M_t(b_\tau)) + (1 - \sigma_{\tau \to t})M_t(a_{\mathsf{safe}})).$$

Now, first observe that by Lemma 20,

$$(1 - \rho_t)\Delta(a_{\mathsf{safe}}) \leq 6\frac{\Delta(a_{\mathsf{safe}})}{\Gamma(a_{\mathsf{safe}})}M_t(a_t),$$

and further
$$\rho_t M_t(b_t) \leq 2M_t(a_t).$$
We are left with terms scaling with $\sigma_{\tau \to t}$ or $(1 - \sigma_{\tau \to t})$. For this, we first observe that
$$M_t(b_\tau) = B_t \omega_t \|b_\tau\|_{V_t^{-1}} \leq \frac{B_t \omega_t}{B_\tau \omega_\tau} \cdot B_\tau \omega_\tau \|b_\tau\|_{V_\tau^{-1}} = \frac{B_t \omega_t}{B_\tau \omega_\tau} \cdot M_\tau(b_\tau),$$
where we use the fact that $V_t$ is nondecreasing (in the positive definite ordering). Let us abbreviate $J_{\tau \to t} := 1 + {}^{B_t \omega_t}/_{(B_\tau \omega_\tau)}$. Upon observing that $\rho_t \leq 1$, to finish the argument, we only need to control
$$(1 - \sigma_{\tau \to t})(\Delta(a_{\mathsf{safe}}) + M_t(a_{\mathsf{safe}})) + J_{\tau \to t} \sigma_{\tau \to t} M_\tau(b_\tau).$$
Now, notice that since $M_\tau(a_{\mathsf{safe}}) \leq \Gamma_0/3$,
$$\sigma_{\tau \to t} = \frac{\Gamma(a_{\mathsf{safe}})}{\Gamma(a_{\mathsf{safe}}) + 3(M_t(b_\tau) + M_\tau(b_\tau))} \leq \frac{\Gamma(a_{\mathsf{safe}})}{\Gamma(a_{\mathsf{safe}}) + 3M_\tau(b_\tau)} \leq \rho_\tau \leq \frac{2M_\tau(a_\tau)}{M_\tau(b_\tau)},$$
where we invoke Lemma 20 for the final two inequalities. Thus, we find that
$$\sigma_{\tau \to t} J_{\tau \to t} M_\tau(b_\tau) \leq J_{\tau \to t} \cdot \rho_\tau M_\tau(b_\tau) \leq 2 J_{\tau \to t} M_\tau(a_\tau).$$
This leaves us with the term $(1 - \sigma_{\tau \to t})(\Delta(a_{\mathsf{safe}}) + M_t(a_{\mathsf{safe}}))$. To bound this, observe that
$$(1 - \sigma_{\tau \to t}) = \frac{3(M_t(b_\tau) + M_\tau(b_\tau))}{\Gamma(a_{\mathsf{safe}}) + 3(M_t(b_\tau) + M_\tau(b_\tau))}$$
$$\implies (1 - \sigma_{\tau \to t})\Gamma(a_{\mathsf{safe}}) = 3\sigma_{\tau \to t}(M_t(b_\tau) + M_\tau(b_\tau)) \leq 3\sigma_{\tau \to t} J_{\tau \to t} M_\tau(b_\tau).$$
Recall from the discussion above that $\sigma_{\tau \to t} M_\tau(b_\tau) \leq \rho_\tau M_\tau(b_\tau) \leq 2M_\tau(a_\tau)$. Using this, and the fact that $M_t(a_{\mathsf{safe}}) \leq \Gamma(a_{\mathsf{safe}})/3$ then yields
$$(1 - \sigma_{\tau \to t})(\Delta(a_{\mathsf{safe}}) + M_t(a_{\mathsf{safe}})) \leq 6 J_{\tau \to t} \frac{\Delta(a_{\mathsf{safe}})}{\Gamma(a_{\mathsf{safe}})} M_\tau(a_\tau) + 2 J_{\tau \to t} M_\tau(a_\tau).$$
Putting everything together, then, we conclude that
$$\Delta(a_t) \leq 6 \frac{\Delta(a_{\mathsf{safe}})}{\Gamma(a_{\mathsf{safe}})} \left( M_t(a_t) + J_{\tau \to t} M_\tau(a_\tau) \right) + 2M_t(a_t) + 4 J_{\tau \to t} M_\tau(a_\tau),$$
which of course implies the bound we set out to show. $\qquad\square$

## H.3 Controlling Accumulation in the Look-Back Bound

We proceed to control the accumulation of the look-back terms.

*Proof of Lemma 5.* Since $B_t$ and $\omega_t$ are nondecreasing, for any $s \leq t \leq T$, we have
$$(1 + (B_t \omega_t(\delta)/B_s \omega_s(\delta)))M_s(a_s) = (B_s \omega_s(\delta) + B_t \omega_t(\delta))\|a_s\|_{V_s^{-1}} \leq 2B_T \omega_T(\delta)\|a_s\|_{V_s^{-1}}.$$
Let $\mathcal{T}_T = \{t \leq T : M_t(a_{\mathsf{safe}}) \leq \Gamma_0/3\}$, and $\mathcal{U}_T = \{s \in \mathcal{T}_T : \Delta(b_s) \leq M_t(b_s)\}$. Then notice that
$$\sum_{t \in \mathcal{T}_T} \|a_{\tau(t)}\|_{V_{\tau(t)}^{-1}} = \sum_{s \in \mathcal{U}_T} L_s \|a_s\|_{V_s^{-1}},$$
where $L_s = |\{t \in \mathcal{T}_T : \tau(t) = s\}|$ is the number of times $s$ serves as $\tau(t)$ for some $t$. But this is the same as the time (restricted to $\mathcal{T}_T$) between $s$ and the *next* member of $\mathcal{U}_T$, i.e., the length of the 'run' of the method playing saturated actions (plus one).

At this point, a weaker bound of the form $\frac{2}{\chi} \log(T^2/\delta) \sum_{s \in \mathcal{U}_T} \|a_s\|_{V_s^{-1}}$ is straightforward: each round has at least a chance $\chi/2$ of picking a saturated $b_t$, and so the chance that the $k$th such run has length greater than $\frac{2}{\chi} \log(k(k+1)/\delta)$ is at most $\delta/k(k+1)$. Since there are at most $T$ runs up to time $T$, union bounding over this gives $\max_{\mathcal{U}_T} L_s \leq 1 + 2\log(T(T+1)/\delta)/\chi$.

The rest of this proof is devoted to give a more refined martingale analysis that saves upon the multiplicative $\log(T)$ term above. We encapsulate this as an auxiliary Lemma below.

**Lemma 21.** *In the setting of Lemma 5, it holds that with probability at least $1 - \delta$,*

$$\sum_{s \in \mathcal{U}_T} L_s \|a_s\|_{V_s^{-1}} \leq \frac{5}{\chi} \Big( \sum_{s \in \mathcal{U}_T} \|a_s\|_{V_s^{-1}} + \log(1/\delta) \Big)$$

This result is shown below. Assuming this result, the original claim follows immediately, since due to the nonnegativity of $\| \cdot \|_{\cdot}$, $\sum_{s \in \mathcal{U}_T} \|a_s\|_{V_s^{-1}} \leq \sum_{t \leq T} \|a_t\|_{V_t^{-1}}$. $\qquad\square$

To finish the argument, we move on to showing the auxiliary lemma described above.

*Proof of Lemma 21.* We work with the reduction to $\sum_{s \in \mathcal{U}_T} \|a_s\|_{V_s^{-1}}$ established above. Let us denote $\zeta_i = \inf\{t > \zeta_{i-1} : M_t(a_{\mathsf{safe}}) \leq \Gamma_0/3, \Delta(b_t) \leq M_t(b_t)\}$ as the times that an unsaturated action is picked, with $\zeta_0 := 0$—for $i : \zeta_i \leq T$, these are precisely the elements of $\mathcal{U}_T$. Notice that this $\{\zeta_i\}$ is a sequence of stopping times adapted to the history $\{\mathfrak{H}_t\}$. Let us further denote $L_i = (\zeta_{i+1} - \zeta_i)$, for $i \geq 0$ (this corresponds to $L_s$, where $s = \zeta_i$). The object we need to control is

$$\sum_{i:\zeta_i \leq T} L_i X_i,$$

where $X_i = \|a_{\zeta_i}\|_{V_{\zeta_i}^{-1}} \in [0,1]$, the lower bound being since $X_i$ is a norm, and the upper bound since $V_{\zeta_i} \succcurlyeq I$. For notational convenience, we always set $X_0 = 1$. Now, to control this, let us first pass to the associated sigma algebrae of the $\zeta_i$ past, denoted as

$$\mathfrak{G}_i := \zeta(\mathfrak{H}_{\zeta_i}).$$

Notice that since $\zeta_i$ is nondecreasing, we know that $\{\mathfrak{G}_i\}$ forms a filtration. Of course, by definition, $X_i$ are adapted to $\mathfrak{G}_i$, while $L_i$ are adapted to $\mathfrak{G}_{i+1}$. We further know that $L_i$ is the time (including $\zeta_i$) between $\zeta_i$ and $\zeta_{i+1}$. But then for each $t > \zeta_i$, $P(\zeta_{i+1} = t | \zeta_{i+1} > t - 1, \mathfrak{H}_{t-1}) \geq \chi/2$. As a result, these $L_i$s are conditionally stochatically domainted by a geometric random variable, i.e.,

$$\mathbb{P}(L_i > 1 + k | \mathfrak{G}_i) \leq (1 - \chi/2)^k.$$

This in turn implies that for any $\lambda$ small enough,

$$\mathbb{E}[e^{\lambda(L_i - 1)X_i} | \mathfrak{G}_i] \leq \frac{\chi/2}{1 - (1 - \chi/2)e^{\lambda X_i}}.$$

In the subsequent, we will need to select a $\lambda$ that is independent of all of these $L_i, X_i$. To ensure that the calculation makes sense, we ensure that $(1 - \chi/2)e^{\lambda} \leq 1$ (which suffices since $0 \leq X_i \leq 1$). Let us define $F_i(\lambda) := -\log((1 - (1 - \chi/2)e^{\lambda X_i})/(\chi/2))$. Then by the above calculation, we find that the process $\{M_i\}$ with $M_0 := 1$ and

$$M_i := \exp\Big( \lambda \sum (L_i - 1)X_i - \sum F_i(\lambda) \Big)$$

is a nonnegative supermartingale with respect to the filtration $\{\mathfrak{F}_i\}$ with $\mathfrak{F}_i = \mathfrak{G}_{i+1}\}$ and $\mathfrak{F}_0$ defined to be the trivial sigma algebra. Thus, by Ville's inequality, $\mathbb{P}(\exists i : M_i > 1/\delta) \leq \delta$. Taking logarithms, we find that with probability at least $1 - \delta$, it holds that

$$\forall n, \sum_{i \leq n} L_i X_i \leq \sum_{i \leq n} X_i + \frac{\log(1/\delta)}{\lambda} + \sum_{i \leq n} \frac{F_i(\lambda)}{\lambda},$$

as long as $0 < \lambda < -\log(1 - \chi/2)$. All we need now is a convenient bound on $F_i(\lambda)$ and a judicious choice of $\lambda$. To this end, we observe the following simple result.

**Lemma 22.** *For any constant $u \in (0,1)$, consider the map $f(x) := -\log \frac{1 - ue^x}{1 - u}$ over the domain $[0, -\log(u))$. Then for all $x \in [0, -\frac{1}{2}\log(u)]$, we have*

$$f(x) \leq \frac{\sqrt{u}}{1 - \sqrt{u}} x.$$

*Proof.* Observe that

$$f'(x) = \frac{ue^x}{1 - ue^x} = \frac{e^f}{1-u}(1 - e^{-f}(1-u)) = \frac{e^f}{1-u} - 1 \geq 0$$

The inequalities above arise since $e^f = \frac{1-u}{1-ue^x} > 1 - u$ using the fact that $e^x \geq 1$. By taking another derivative, we may see that $f'$ itself is an increasing function. Now, suppose $g(x)$ satisfies

$$g(0) = f(0) = 0, \text{ and } \forall x, g'(x) = f'(-\frac{1}{2}\log(u)) = \frac{\sqrt{u}}{1 - \sqrt{u}}.$$

Then since $g'(x) \geq f'(x)$ for all $x \in [0, -\frac{1}{2}\log(u)]$, by the fundamental theorem of calculus it follows that for all $x \leq -\frac{1}{2}\log u$, $f(x) = \int_0^x f' \leq \int_0^x g' = g(x)$. □

Now, of course, $F_i(\lambda) = f(\lambda X_i)$, with $u = 1 - \chi/2$. Then setting $\lambda = -\frac{1}{2}\log(1 - \chi/2)$, we have

$$\forall n, \sum_{i \leq n} L_i X_i \leq \sum X_i + \frac{\log(1/\delta)}{-\log(1 - \chi/2)/2} + \sum_{i \leq n} \frac{\sqrt{1 - \chi/2}}{1 - \sqrt{1 - \chi/2}} X_i.$$

To get the form needed, we observe that

$$\frac{\sqrt{1 - v}}{1 - \sqrt{1 - v}} \leq \frac{2}{v} \iff (2 + v)^2(1 - v) \leq 4 \iff -v^3 - 3v^2 \leq 0,$$

and of course $-\log(1 - v)/2 \geq v/2$. Plugging in $v = \chi/2 > 0$, we end up at

$$\forall n, \sum_{i \leq n} L_i X_i \leq \left(1 + \frac{4}{\chi}\right) \sum_{i \leq n} X_i + \frac{4}{\chi}\log(1/\delta).$$

Note that no explicit $n$-dependent term appears in the above. This makes sense: we essentially have the $X_i$s acting as 'time steps', and so $\sum L_i X_i$ should behave as $(1 + 2/\chi)\sum X_i + O(\sqrt{\sum X_i \log(1/\delta)/\chi} + \log(1/\delta))$, via a Bernstein-type computation. In our case, the square root terms do not meaningfully help the solution,[8] and so we just pick a convenient $\lambda$ instead.[9] Now, going back to our original object of study, we have $L_i = L_{\zeta_i}, X_i = \|a_{\zeta_i}\|_{V_{\zeta_i}^{-1}}$, and these $\zeta_i$s are precisely the members of $\mathcal{U}_T$, so we conclude that

$$\forall T, \sum_{s \in \mathcal{U}_T} L_s X_s \leq \frac{5}{\chi}\left(\sum_{s \in \mathcal{U}_t} X_s + \log(1/\delta)\right). \quad □$$

### H.4 Regret and Risk Bounds for S-COLTS

With the above pieces in place, we move on to showing the final bounds on the behaviour of S-COLTS.

*Proof of Theorem 9.* We first argue the safety properties. Firstly, in the exploration phase, as well as to explore, we repeatedly play $a_{\mathsf{safe}}$. But this is, by definition, safe, and so accrues no safety cost. When not playing $a_{\mathsf{safe}}$, the selected action $a_t$ at time $t$ satisfies

$$\hat{\Phi}_t a_t + \omega_t(\delta)\|a_t\|_{V_t^{-1}}\mathbf{1}_m \leq \alpha.$$

But, given the consistency event $\mathsf{Con}_t(\delta)$,

$$\forall a, \Phi_* a \leq \hat{\Phi}_t a + \omega_t(\delta)\|a\|_{V_t^{-1}},$$

and so $\Phi_* a_t \leq \alpha$. Since $\mathsf{Con}(\delta) := \bigcap \mathsf{Con}_t(\delta)$ holds with chance at least $1 - \delta$, it follows that $a_t$ is safe at every $t$, and a fortiori, $\mathbf{S}_T = 0$ for every $T$.

Let us turn to the regret analysis. Fix any $T$. We break the regret analysis into four pieces: the regret accrued over the initial exploration, that accrued after this phase, but when $M_t(a_{\mathsf{safe}}) > \Gamma_0/3$, and

---

[8]since there will always be an additive $\log(1/\delta)$ and $\frac{1}{\chi}\sum X_i$ term

[9]and in the process, avoid the subtleties of the dependence of $\lambda$ on the $X_i$s if we optimised it

over the time $\mathcal{T}_T := \{t \geq T_0 : M_t(a_{\mathsf{safe}}) \leq \Gamma_0/3\}$, and finally the regret incurred up to the time $\inf\{t : \delta_t > \chi/2\}$.

The last of these is the most trivial to handle: the number of such rounds is bounded as $\sqrt{2\delta/\chi}$, and the regret in any round is at most 2.

For the first case, Lemma 18 ensures that with probability at least $1 - \delta$, this phase has length at most

$$\frac{8}{\Gamma(a_{\mathsf{safe}})^2} \log(8/(\delta\Gamma(a_{\mathsf{safe}})^2)),$$

and further, the output $\Gamma_0$ is at least $\Gamma(a_{\mathsf{safe}})/2$ at the end. Using this to instantiate Lemma 19, we further find that the number of times $a_{\mathsf{safe}}$ is selected beyond this initial exploration is in total bounded as

$$1 + \frac{36\omega_T^2 B_T^2}{\Gamma(a_{\mathsf{safe}})^2}.$$

Together these contribute at most

$$\Delta(a_{\mathsf{safe}}) \cdot \frac{44\omega_T^2 B_T^2}{\Gamma(a_{\mathsf{safe}})^2} \log(8/\delta\Gamma(a_{\mathsf{safe}})^2)$$

to the regret.

This leaves us with the times at which $M_t(a_{\mathsf{safe}}) \leq \Gamma_0/3$, for which we apply Lemma 7, along with the control of Lemma 5 to find that the net regret accrued thus is bounded as

$$O\left(\left(1 + \frac{\Delta(a_{\mathsf{safe}})}{\Gamma(a_{\mathsf{safe}})}\right) B_T \omega_T(\delta) \cdot \frac{5}{\chi} \left(\sum_{t \leq T} \|a_t\|_{V_t^{-1}} + \log(1/\delta)\right)\right).$$

To complete the book-keeping, the probabilistic events required for this are the consistency of the confidence sets, that for all $t, \max(\|\eta_t\|, \max_i \|H_t^i\|)$ is bounded by $B(\delta_t)$, and of course the bound on the times between unsaturated $b_t$ being constructed from Lemma 5. Together, these occur with chance at least $1 - 3\delta$, and putting the same together with the stopping time bound, we conclude that with chance at least $1 - 4\delta$, $\mathsf{S\text{-}COLTS}(\mu, \delta)$ satisfies the regret bound

$$\mathbf{R}_T \leq \left(1 + \frac{\Delta(a_{\mathsf{safe}})}{\Gamma(a_{\mathsf{safe}})}\right) \widetilde{O}\left(\frac{\omega_t(\delta)B_T}{\chi} \sum_{t \leq T} \|a_t\|_{V_t^{-1}}\right) + \frac{\Delta(a_{\mathsf{safe}})}{\Gamma(a_{\mathsf{safe}})} \cdot \widetilde{O}\left(\frac{\omega_T^2 B_T^2}{\Gamma(a_{\mathsf{safe}})}\right) + \sqrt{\frac{8\delta}{\chi}}.$$

Now, invoking Lemma 14, we can bound $\omega_T(\delta) = \widetilde{O}(\sqrt{d} + \log(m/\delta))$, and $\sum \|a_t\|_{V_t^{-1}} = \widetilde{O}(\sqrt{dT})$. Finally, for the law $\mu$ induced via the coupled noise design by $\mathrm{Unif}(\sqrt{3d}\mathbb{S}^d)$, we further know that $B_T = O(\sqrt{d})$ and $\chi \geq 0.28$. Of course, for this noise, $B_t = \sqrt{3d}$ with certainty, which boosts the probability above to $1 - 3\delta$. The claim thus follows for $\mathsf{S\text{-}COLTS}(\mu, \delta/3)$. $\qquad\square$

### H.5 An Optimism-Based Analysis of S-COLTS

We analyse $\mathsf{S\text{-}COLTS}$ under the assumption that $\mu$ satisfies $B$-concentration and $\pi$-global optimism (Definition 13). We shall be somewhat informal in executing this.

**Setting Up.** We first note that regret accrued over rounds in which $M_t(b_t) > \Gamma(a_{\mathsf{safe}})/3$ and $M_t(a_{\mathsf{safe}}) \leq \Gamma_0/3$ is small. Indeed,

$$\sum_{t \in \mathcal{T}_T} \mathbb{1}\{M_t(b_t) > \Gamma(a_{\mathsf{safe}})/3\} \leq \frac{9}{\Gamma(a_{\mathsf{safe}})^2} \sum_{t \in \mathcal{T}_T} M_t(b_t)^2$$

$$\leq \frac{16}{\Gamma(a_{\mathsf{safe}})^2} \sum_{t \in \mathcal{T}_T} \frac{M_t(a_t)^2}{\rho_t^2} = \widetilde{O}\left(\frac{d^5}{\Gamma(a_{\mathsf{safe}})^4}\right),$$

where $\mathcal{T}_T = \{t : M_t(a_{\mathsf{safe}}) \leq \Gamma_0/3\}$, and we used the bound on $M_t(b_t)$ from Lemma 20, along with the fact that since $b_t \in \mathcal{A}, M_t(b_t) \leq B_t\omega_t = \widetilde{O}(d)$, which in turn implies that $\rho_t \geq \Gamma(a_{\mathsf{safe}})/\widetilde{\Omega}(d)$.

Naturally, this additive term is much weaker than that seen in Theorem 9. Nevertheless, the optimism-based framework does recover a similar main term. In particular we will show a regret bound of $\widetilde{O}(\Gamma(a_{\mathsf{safe}})^{-1}\sqrt{d^3 T})$

The point of the above condition is that (using Lemma 20), if $M_t(b_t) \leq \Gamma(a_{\mathsf{safe}})/3$, then $\rho_t \geq \frac{1}{2}$. We will repeatedly use this fact in the subsequent.

Now, we begin similarly to the previous analysis by using

$$\Delta(a_t) = (1 - \rho_t)\Delta(a_{\mathsf{safe}}) + \rho_t\Delta(b_t) \leq (1 - \rho_t)\Delta(a_{\mathsf{safe}}) + \Delta(b_t).$$

The first term is well-controlled, as detailed in the proof of Lemma 7. So, we only need to worry about $\sum \Delta(b_t)$. Notice that for this it suffices to control $\sum \mathbb{E}[\Delta(b_t)|\mathfrak{H}_{t-1}]$. Indeed, $\Delta(b_t) \leq 1$ (and if it is $\leq 0$, we can just drop it from the sum, i.e., we could study $(\Delta(b_t))_+$ instead with no change in the argument), so the difference $\sum_{t \leq T} \Delta(b_t) - \mathbb{E}[\Delta(b_t)|\mathfrak{H}_{t-1}]$ is a martingale with increments lying in $[-1, 1]$, and the LIL (Lemma 17) ensures that for all $T$ simultaneously, the difference between these is $O(\sqrt{T \log(\log(T)/\delta)})$ with chance at least $1 - \delta$.

From the above, then, we can restrict attention to $t$ such that $M_t(a_{\mathsf{safe}}) \leq \Gamma_0/3, \rho_t \geq \frac{1}{2}$. Finally, recalling the notation $K(\theta, \Phi) = \max\{\theta^\top a : a \in \mathcal{A}, \Phi a \leq \alpha\}$ from Definition 13, we observe that

$$\Delta(b_t) = \theta_*^\top a_* - \widetilde{\theta}_t^\top b_t + (\widetilde{\theta}_t - \theta_*)^\top b_t$$
$$\leq K(\theta_*, \Phi_*) - K(\widetilde{\theta}_t, \widetilde{\Phi}_t) + M_t(b_t)$$
$$\leq K(\theta_*, \Phi_*) - K(\widetilde{\theta}_t, \widetilde{\Phi}_t) + 4M_t(a_t),$$

where we used Lemma 3, and Lemma 20 along with the fact that $\rho_t \geq 1/2$. Now note that the final term above is summable to $\widetilde{O}(\sqrt{d^3 T})$. Thus, it equivalently suffices to analyse the behaviour of $\mathbb{E}_{t-1}[K(\theta_*, \Phi_*) - K(\widetilde{\theta}_t, \widetilde{\Phi}_t)|\mathfrak{H}_{t-1}]$. In order to do so, we begin with a 'symmetrisation' lemma.

**Lemma 23.** *Let $(\widetilde{\theta}_t, \widetilde{\Phi}_t)$ and $(\bar{\theta}_t, \bar{\Phi}_t)$ denote two independent copies of parameter perturbations at time $t$. Let $\mathbb{E}_{t-1}[\cdot] := \mathbb{E}[\cdot \mid \mathfrak{H}_{t-1}]$. If $\mu$ satisfies $\pi$-global optimism, then*

$$\mathbb{1}_{\mathsf{Con}_t(\delta)}\mathbb{E}_{t-1}[(K(\theta_*, \Phi_*) - K(\widetilde{\theta}_t, \widetilde{\Phi}_t)] \leq \mathbb{1}_{\mathsf{Con}_t(\delta)} \cdot \frac{1}{\pi}\mathbb{E}_{t-1}[|K(\widetilde{\theta}_t, \widetilde{\Phi}_t) - K(\bar{\theta}_t, \bar{\Phi}_t)|].$$

*Proof.* Let $\bar{\mathsf{G}} := \{K(\bar{\theta}_t, \bar{\Phi}_t) \geq K(\theta_*, \Phi_*)\}$. Since $K(\theta_*, \Phi_*)$ is a constant, and since $(\widetilde{\theta}_t, \widetilde{\Phi}_t)$ are independent of $(\bar{\theta}_t, \bar{\Phi}_t)$ given $\mathfrak{H}_{t-1}$, we conclude that

$$\mathbb{E}_{t-1}[K(\theta_*, \Phi_*) - K(\widetilde{\theta}_t, \widetilde{\Phi}_t)] = \mathbb{E}_{t-1}[K(\theta_*, \Phi_*) - K(\widetilde{\theta}_t, \widetilde{\Phi}_t) \mid \bar{\mathsf{G}}].$$

But given $\bar{\mathsf{G}}$, $K(\theta_*, \Phi_*) \leq K(\bar{\theta}_t, \bar{\Phi}_t)$, and so

$$\mathbb{E}_{t-1}[K(\theta_*, \Phi_*) - K(\widetilde{\theta}_t, \widetilde{\Phi}_t)] \leq \mathbb{E}_{t-1}[K(\bar{\theta}_t, \bar{\Phi}_t) - K(\widetilde{\theta}_t, \widetilde{\Phi}_t) \mid \bar{\mathsf{G}}]$$
$$\leq \mathbb{E}_{t-1}[|K(\bar{\theta}_t, \bar{\Phi}_t) - K(\widetilde{\theta}_t, \widetilde{\Phi}_t)| \mid \bar{\mathsf{G}}].$$

Finally, for any nonnegative random variable $X$, and any event $\mathsf{E}$, it holds that

$$\mathbb{E}_{t-1}[X|\mathsf{E}]\mathbb{E}_{t-1}[\mathbb{1}_\mathsf{E}] = \mathbb{E}_{t-1}[X\mathbb{1}_\mathsf{E}] \leq \mathbb{E}_{t-1}[X].$$

The claim follows upon taking $X = |K(\bar{\theta}_t, \bar{\Phi}_t) - K(\widetilde{\theta}_t, \widetilde{\Phi}_t)|, \mathsf{E} = \bar{\mathsf{G}}$, and recognising that due to $\pi$-optimism, $\bar{\mathsf{G}}$ satisfies $\mathbb{E}_{t-1}[\mathbb{1}_{\bar{\mathsf{G}}}]\mathbb{1}_{\mathsf{Con}_t} \geq \pi\mathbb{1}_{\mathsf{Con}_t}$. $\qquad\square$

The main question now becomes controlling how far the deviations in $K$ can go. We control this using a similar scaling trick as in the proof of Lemma 7.

For the sake of clarity, we will denote the optimiser of $K(\widetilde{\theta}_t, \widetilde{\Phi}_t)$ as $\tilde{b}_t$ (instead of just $b_t$ as in the rest of the text), and similarly that of $K(\bar{\theta}_t, \bar{\Phi}_t)$ as $\bar{b}_t$. Our goal is to control (the conditional mean of)

$$|\bar{\theta}_t^\top \bar{b}_t^\top - \widetilde{\theta}_t^\top \tilde{b}_t|.$$

Naturally, the core issue remains that $\bar{b}_t$ and $\tilde{b}_t$ are optima in distinct feasible sets, and so it is hard to, e.g., compare $\widetilde{\theta}_t^\top \tilde{b}_t$ and $\widetilde{\theta}_t^\top \bar{b}_t$. To this end, we observe that

$$\bar{\Phi}_t\bar{b}_t \leq \alpha \implies \Phi_* b_t \leq \alpha + M_t(\bar{b}_t)\mathbf{1}_m \implies \widetilde{\Phi}_t\bar{b}_t \leq \alpha + 2M_t(b_t)\mathbf{1}_m,$$

as long as consistency and the boundedness of the noise norms holds (which occurs with high probability). Using this and the fact that $\widetilde{\Phi}_t a_{\mathsf{safe}} \leq \alpha - 2\Gamma(a_{\mathsf{safe}})/31_m$, we find that

$$\widetilde{\Phi}_t(\bar{\sigma}_t \bar{b}_t + (1 - \bar{\sigma}_t)a_{\mathsf{safe}}) \leq \alpha, \text{ where } \bar{\sigma}_t = \frac{\Gamma(a_{\mathsf{safe}})}{\Gamma(a_{\mathsf{safe}}) + 3M_t(\bar{b}_t)}.$$

Thus, we may write

$$\bar{\theta}_t^\top \bar{b}_t - \widetilde{\theta}_t^\top \tilde{b}_t = (1 - \bar{\sigma}_t)\bar{\theta}^\top \bar{b}_t + \bar{\sigma}_t(\bar{\theta}_t - \widetilde{\theta}_t^\top)\bar{b}_t + \widetilde{\theta}_t^\top(\bar{\sigma}_t \bar{b}_t - \tilde{b}_t).$$

Above, the third term is nonpositive, while the second term may be bounded by $2\bar{\sigma}_t M_t(\bar{b}_t)$, which can further be bounded by $8M_t(\bar{a}_t)$ upon recalling that $\rho_t(\bar{b}_t) \geq \frac{1}{2}$ and the bound on $\rho_t M_t(b_t)$ in Lemma 20. This leaves the first term. It is tempting to bound this directly via $\bar{\theta}_t^\top \bar{b}_t \leq \|\bar{\theta}_t\|\|\bar{b}_t\|$, but notice that the former can be as large as $B_t \sim \sqrt{d}$. Instead, we can use the related bound

$$(1 - \bar{\sigma}_t)(\bar{\theta}^\top \bar{b}_t) \leq (1 - \bar{\sigma}_t)M_t(\bar{b}_t) + (1 - \bar{\sigma}_t)\theta_*^\top \bar{b}_t.$$

Now notice that $(1 - \bar{\sigma}_t) \leq 1$, and $M_t(\bar{b}_t) \leq 4M_t(\bar{a}_t)$ controls the first term. Similarly, $\theta_*^\top \bar{b}_t \leq 1$ (both have norm bounded by 1), so the second term is bounded by $1 - \bar{\sigma}_t \leq \frac{3M_t(\bar{b}_t)}{\Gamma(a_{\mathsf{safe}})} \leq 12\frac{M_t(\bar{a}_t)}{\Gamma(a_{\mathsf{safe}})}$. Putting these together, we conclude that

$$(1 - \bar{\sigma}_t)(\bar{\theta}_t^\top \bar{b}_t) \leq 4M_t(\bar{a}_t) + \frac{12M_t(\bar{a}_t)}{\Gamma(a_{\mathsf{safe}})},$$

which in turn yields the bound

$$K(\bar{\theta}_t, \bar{\Phi}_t) - K(\bar{\theta}_t, \widetilde{\Phi}_t) \leq 12M_t(\bar{a}_t) + \frac{12M_t(\bar{a}_t)}{\Gamma(a_{\mathsf{safe}})} \leq \frac{24M_t(\bar{a}_t)}{\Gamma(a_{\mathsf{safe}})}.$$

Of course, switching the roles of $(\bar{\theta}_t, \bar{\Phi}_t)$ and $(\widetilde{\theta}_t, \widetilde{\Phi}_t)$, we have an analogous bound on $K(\widetilde{\theta}_t, \widetilde{\Phi}_t) - K(\bar{\theta}_t, \bar{\Phi}_t)$. Putting these together, we conclude that

$$|K(\bar{\theta}_t, \bar{\Phi}_t) - K(\widetilde{\theta}_t, \widetilde{\Phi}_t)| \leq \frac{24(M_t(\bar{a}_t) + M_t(\tilde{a}_t))}{\Gamma(a_{\mathsf{safe}})}.$$

Finally, notice that $\bar{a}_t$, $\tilde{a}_t$, and the actually selected action $a_t$ all have the same distribution given $\mathfrak{H}_{t-1}$. We can thus conclude that

$$\mathbb{E}_{t-1}[|K(\bar{\theta}_t, \bar{\Phi}_t) - K(\widetilde{\theta}_t, \widetilde{\Phi}_t)|] \leq 48\mathbb{E}_{t-1}\left[\frac{M_t(a_t)}{\Gamma(a_{\mathsf{safe}})}\right].$$

With this in hand, the issue returns to one of concentration. We know that $\sum M_t(a_t)$ is $\widetilde{O}(\sqrt{d^3 T})$, and each $M_t(a_t)$ is bounded as $O(d)$ and so $\sum M_t(a_t) - \mathbb{E}_{t-1}[M_t(a_t)]$ enjoys concentration at the scale $d\mathrm{LIL}(T, \delta) = \widetilde{O}(\sqrt{d^2 T}) = o(\sqrt{d^3 T})$. Thus, passing back to the the unconstrained sums, we end up with a bound of the form

$$\mathbf{R}_T = \widetilde{O}\left(\Gamma(a_{\mathsf{safe}})^{-1}\sqrt{d^3 T}\right) + \widetilde{O}(d^5 \Gamma(a_{\mathsf{safe}})^{-4}).$$

The main loss in the main term above is that instead of a $\Delta(a_{\mathsf{safe}})/\Gamma(a_{\mathsf{safe}})$, we just have a $\Gamma(a_{\mathsf{safe}})^{-1}$ term in the bound. This can be lossy, e.g., when $a_{\mathsf{safe}}$ is very close to $a_*$, but in the regime $\Delta(a_{\mathsf{safe}}) = \Omega(1)$, it recovers essentially the same guarantees as Theorem 9, albeit with a weaker additive term.

# I   The Analysis of Soft Constraint Enforcement Methods.

## I.1   The Analysis of R-COLTS

Let us first show the optimism result for R-COLTS

*Proof of Lemma 10.* Fix any $t$, and assume $\mathsf{Con}_t(\delta)$. For each $i \in [1 : I_t]$, we know that $K(i, t) := K(\widetilde{\theta}(i, t), \widetilde{\Phi}(i, t)) \geq \widetilde{\theta}(i, t)^\top a_* \geq \theta_*^\top a_*$ whenever the event L occurs, and thus this inequality holds with chance at least $\pi$ in every round. Since the draws are all independent given $\mathfrak{H}_{t-1}$, the chance that $\max K(i, t) < \theta_*^\top a_*$ is at most $(1 - \pi)^{I_t} \leq \exp(-\log(1/\delta_t)r \cdot \pi) \leq \delta_t = \delta/t(t+1)$. Thus, if we assume that $\mathsf{Con}(\delta) := \bigcap \mathsf{Con}_t(\delta)$ holds true, the chance that at any $t$, $K(i_t, t) < \theta_*^\top a_*$ is at most $\sum \delta_t = \delta$. By Lemma 1, $\mathsf{Con}(\delta)$ holds with chance at least $1 - \delta$, and we are done. $\square$

Of course, the above proof, and thus the statement of this Lemma, holds verbatim if we replace $L_t$ by $G_t$ (Definition 13).

With the optimism result of Lemma 10, the argument underlying Theorem 11 is extremely standard.

*Proof of Theorem 11.* Assume consistency, and that at every $t$, $\widetilde{\theta}_t^\top a_t \geq \theta_*^\top a_*$. Since we sample at most $2 + r\log(1/\delta_t)$ programs in round $t$, we further know that with probability at least $1 - \delta$,

$$\forall t, \max_i \left( \max \|\eta(i,t)\|, \max_j \|H_t^j(i,t)\| \right) \| \leq \beta_t := B(\delta_t/(2 + r\log(1/\delta_t)).$$

Assume that this too occurs, and define $\tilde{M}_t(a) = \omega_t(\delta)(1 + \beta_t)\|a\|_{V_t^{-1}}$. Then, using consistency,

$$\theta_*^\top a_t \geq \widetilde{\theta}_t^\top a_t - \tilde{M}_t(a_t), \Phi_* a_t \leq \widetilde{\Phi}_t a_t + \tilde{M}_t(a_t)\mathbf{1}_m \leq \alpha + \tilde{M}_t(a_t)\mathbf{1}_m.$$

So, the safety risk is bounded as

$$\mathbf{S}_T \leq \sum_t \tilde{M}_t(a_t) \leq \omega_T(\delta)(1 + \beta_T) \sum_{t \leq T} \|a_t\|_{V_t^{-1}}.$$

Further,

$$\theta_*^\top a_* - \theta_*^\top a_t \leq \theta_*^\top a_* - \widetilde{\theta}_t^\top a_t + \tilde{M}_t(a_t),$$

which implies that

$$\mathbf{R}_T \leq \omega_T(\delta)(1 + \beta_T \sum_{t \leq T} \|a_t\|_{V_t^{-1}}$$

as well. Now Lemma 14 controls $\omega_T \sum_{t \leq T} \|a_t\|_{V_t^{-1}}$ to $\widetilde{O}(\sqrt{d^2 T})$, and for our selected noise, the coupled design driven by $\mathrm{Unif}(\sqrt{3d}\mathbb{S}^d)$, we have $B(\cdot) = \sqrt{3d}$ independently of $t, r, \delta$, and thus $\beta_T = \sqrt{3d}$. The events needed to show the above were the consistency, the concentration of the sampled noise to $\beta_t$ at each time $t$, and the optimism event of Lemma 10. Again, the second happens with certainty for us, and so the above bounds hold at all $T$ with chance at least $1 - 2\delta$. Consequently, the result was stated for R-COLTS($\mu, r, \delta/2$). $\qquad\square$

## I.2 The Exploratory-COLTS Method

As discussed in §5, the Exploratory COLTS, or E-COLTS method, augments COLTS with a low-rate of flat exploration, and exploits the resulting (eventual) perturbed feasibility of actions with large safety margin to bootstrap the scaling-based analysis of S-COLTS to soft-enforcement without resampling.

The main distinction lies, of course, in the fact that in the soft enforcement setting, we do not have access to a given safe action $a_{\mathsf{safe}}$. To motivate the method, let us consider how S-COLTS uses the knowledge of $a_{\mathsf{safe}}$. This occurs in three ways: to ensure the existence of $a(\eta_t, H_t, t)$, to compute the action $a_t$ from this, and to enable the look-back analysis of Lemma 7. The second use is easy to address: we will simply play $a_t = a(\eta_t, H_t, t)$ if it exists. The key observation is that rather than explicit knowledge of any one particular safe action, as long as *some action $a$* exists such that $M_t(a) \leq \Gamma(a)/3$, the entirely of the first and third uses can be recovered, and so the machinery of §4 can be enabled.

**Forced Exploration.** We enable the *eventual* existence of such actions by introducing a small rate of *forced exploration* in our method E-COLTS. Concretely, we demand a '$\kappa$-good' exploration policy over $\mathcal{A}$, i.e., one such that after $N$ exploratory actions $e_1, \cdots, e_N$, we are assured that $\sum e_i e_i^\top \succcurlyeq \kappa \lfloor N/d \rfloor I_d$, where $\kappa > 0$ is a constant. This can, e.g., be done by playing the elements of a barycentric spanner of $\mathcal{A}$ in round-robin [AK08; DHK08]. The resulting $\kappa$ is a geometric property of $\mathcal{A}$, and we note that $\kappa$ only enters the analysis, not the algorithm.

Let us call a time step $t$ where the exploratory policy is executed an 'E-step'. In E-COLTS, we ensure tha tat any $t$, at least $B_t \omega_t \sqrt{dt}$ such E-steps have been performed, and if not, we force an E-step. Note that we expect that the majority of the learning process occurs at steps other than E-steps, since this is where the informative action $a(\eta_t, H_t, t)$ is played. Consequently, we will call such steps 'L-steps'.

---

**Algorithm 3** Exploratory-COLTS (E-COLTS($\mu, \delta$))

---
1: **Input**: $\mu, \delta$, exploration policy.
2: **Initialise**: $u_0 \leftarrow 0, B_t \leftarrow 1 + B(\delta_t)$
3: **for** $t = 1, 2, \dots$ **do**
4:      Draw $(\eta_t, H_t) \sim \mu$.
5:      **if** $u_{t-1} \leq B_t \omega_t(\delta) \sqrt{dt}$ OR $a(\eta_t, H_t, t)$ does not exist **then**
6:          Pick $a_t$ via exploration policy.
7:          $u_t \leftarrow u_{t-1} + 1$.
8:      **else**
9:          $a_t \leftarrow a(\eta_t, H_t, t), u_t \leftarrow u_{t-1}$.
10:      Play $a_t$, observe $R_t, S_t$, update $\mathfrak{H}_t$.

---

By our requirement of enough E-steps, at any L-step $t$, the sample second moment matrix $V_t$ satsifies $V_t \succcurlyeq \kappa B_t \omega_t \sqrt{t/d} I_d$, and so,

$$\forall a, M_t(a) \leq \psi(t) := \left( \frac{dB_t^2 \omega_t^2}{\kappa^2 t} \right)^{1/4} \cdot \|a\|.$$

This means that at such $t$, any $a$ with $\Gamma(a) > 2\psi(t)/3$ satisfies $M_t(a) \leq \Gamma(a)/3$, and so $a(\eta_t, H_t, t)$ exists, and we may use the analysis of §4 for such $a$.

**Regret Bound.** The above insight is the main driver of the result of Theorem 12, which we show in §I.2.1 to follow. Recall that this states that under the E-COLTS strategy, executed with a $\mu$ constructed through the coupled noise design with base measure $\mathrm{Unif}(\sqrt{3d}\mathbb{S}^d)$, the risk and regret satisfy, with high probability, the bounds

$$\mathbf{S}_T = \widetilde{O}(\sqrt{d^3 T}) + \min_a \widetilde{O}\left( \frac{d^3 \|a\|^4}{\kappa^2 \Gamma(a)^4} \right), \text{ and}$$

$$\mathbf{R}_T = \min_{a:\Gamma(a)>0} \left\{ \mathcal{R}(a)\widetilde{O}(\sqrt{d^3 T}) + \widetilde{O}\left( \frac{d^3 \|a\|^4}{\kappa^2 \Gamma(a)^4} \right) \right\},$$

where $\kappa$ is precisely the 'goodness-factor' of the exploratory policy. Let us briefly discuss this result.

**Risk bound.** Unlike S-COLTS, E-COLTS suffers nontrivial risk, which is unavoidable due to the lack of knowledge of $a_{\mathsf{safe}}$ [PGBJ21]. The $\widetilde{O}(\sqrt{d^3 T})$ risk above above is comparable to the $\widetilde{O}\sqrt{d^2 T}$ risk of the prior soft enforcement method DOSS [GCS24], with a $\sqrt{d}$ loss again attributable to efficiency. Note that compared to R-COLTS, the risk bound is essentially the same, but now incurs an extra additive term scaling, essentially, with $(\max_a \Gamma(a))^{-4}$. Thus, a nontrivial risk bound is only shown if this maximum is strictly positive, i.e., under Slater's condition. Nevertheless, the term is additive, and scales with $T$ only logarithmically (through a dependence on $\omega_t(\delta)$, and so in typical scenarios is not expected to dominate as $T$ diverges, although the fourth-power dependence on this quantity would increase the 'burn-in' time of this result.

**Regret bound.** As discussed in §5, the main term of the regret bound above improves over that of S-COLTS, since it *minimises* over $\mathcal{R}(a)$, rather than working with the arbitrary $\mathcal{R}(a_{\mathsf{safe}})$. Note that finding the minimiser of $\mathcal{R}$ may be challenging, but E-COLTS nevertheless adapts to this. However, the additive lower-order term is larger than in S-COLTS due to the 'flat' exploration of E-COLTS, and its practical effect is unclear. In simple simulations, we do observe a significant regret improvement (§J). We note that the $\kappa$-good exploration condition only affects the lower order term in $\mathbf{R}_T$, although again the fourth order dependence on $\Gamma(a)$ is nontrivial. Of course, relative to E-COLTS, the result suffers from an instance-dependence, and again, unless Slater's condition is satisfied, it is ineffective.

**Practical Role of Forced Exploration.** E-COLTS uses forced exploration to ensure that $V_t$ is large, which leads to both feasibility of the perturbed program, and the scaling-based analysis. In practice, however, one expects that low-regret algorithms satisfy $\max_a \|a\|_{V_t^{-1}} \lesssim t^{-1/4}\|a\|$ directly, the idea being that actions with larger $V_t^{-1}$-norm represent underexplored directions that would naturally be selected (recent work has made strides towards actually proving such a result, although it does not quite get there [BGCG23]). Thus we believe that this forced exploration can practically be omitted except when the perturbed program is infeasible. Indeed, in simulations, we find that this strategy already has good regret (§J).

### I.2.1 The Analysis of E-COLTS

We will essentially reuse our analysis of S-COLTS, with slight variations.

*Proof of Theorem 12.* We will first discuss the bound on the regret. Throughout, we assume consistency, and the noise concentration event of Lemma 3. We will further just write $\omega_t$ instead of $\omega_t(\delta)$. Recall the terminology that every $t$ in which we pick an action according to the exploratory policy is called an 'E-step', and every other step an 'L-step'. Here E and L stand for exploration and learning respectively, the idea being that the former constitute the basic exploration required to enable feasibility under perturbations, and so the main learning process occurs in L-steps.

Note that the number of E-steps up to time $t$ is explicitly delineated to be at most $\lceil B_t\omega_t\sqrt{dt}\rceil$. Using the $\kappa$-good assumption, then, we find that at every L-step,

$$V_t \succcurlyeq \kappa B_t\omega_t\sqrt{t/d}I \iff (\kappa B_t\omega_t\sqrt{t/d})^{-1}I \succcurlyeq V_t^{-1}.$$

Now, fix any action $a_0$ with $\Gamma(a_0) > 0$. Then notice that at any L-step,

$$\|a_0\|_{V_t^{-1}}^2 \leq \frac{\sqrt{d}\|a_0\|^2}{\kappa B_t\omega_t\sqrt{t}} \implies M_t(a_0)^2 \leq \frac{B_t\omega_t\|a_0\|^2}{\kappa}\cdot\sqrt{d/t}.$$

Thus, for all

$$t \geq t_0(a_0) := \inf\left\{t : \frac{3^4 d\|a_0\|^4 B_t^2\omega_t^2}{\kappa^2\Gamma(a_0)^4} \leq t\right\}$$

that are L-steps, we know that as long as the noises $\eta_t, H_t$ satisfy the bound of Lemma 3, $\widetilde{\Phi}_t a_0 \leq \alpha - 2\Gamma(a_0)/31_m$. Note that since $\omega_t^2 \leq d\log(t) + \log(m/\delta)$, and since under our choice of coupled noise, $B_t = \sqrt{3d}$ for all $t$, we can conclude that

$$
\begin{aligned}
t_0(a_0) &\leq \frac{Cd^3\|a_0\|^4}{\kappa^2\Gamma(a_0)^4}\log\frac{Cd^3\|a_0\|^4}{\kappa^2\Gamma(a_0)^4} + \frac{Cd^2\|a_0\|^4\log(m/\delta)}{\kappa^2\Gamma(a_0)^4}\log\frac{Cd^2\|a_0\|^4\log(m/\delta)}{\kappa^2\Gamma(a_0)^4}\\
&= \widetilde{O}\left(\frac{d^3\|a_0\|^4}{\kappa^2\Gamma(a_0)^4}\right),
\end{aligned}
$$

where $C$ is some large enough constant ($C = 4\cdot 81$ suffices). This implies that at all $t > t_0(a_0)$ at which the number of E-steps, $u_t$, is large enough, the perturbed program is feasible, and $a_t$ exists. Thus, after this time, no extraneous E-steps are accrued due to infeasibility of the perturbed program.

At this point we apply the proof of Lemma 7, with $\rho_t = 1$. Let

$$\tau = \tau(t) = \sup\{s \leq t : \Delta(a_s) \leq M_t(a_s), M_t(a_0) \leq \Gamma(a_0)/3\}.$$

Now, $a_\tau$ need not be feasible for $\widetilde{\Phi}_t$, but we know that $\widetilde{\Phi}_\tau a_\tau \leq \alpha \implies \widetilde{\Phi}_t a_\tau \leq \alpha + M_t(a_\tau) + M_\tau(a_\tau)$. So for

$$\sigma_{\tau\to t} := \frac{\Gamma(a_0)}{\Gamma(a_0) + 3(M_t(a_\tau) + M_\tau(a_\tau))},$$

we know that

$$\widetilde{\Phi}_t(\sigma_{\tau\to t}a_\tau + (1 - \sigma_{\tau\to t})a_0) \leq \alpha.$$

Let $\bar{a}_{\tau\to t} := \sigma_{\tau\to t}a_\tau + (1 - \sigma_{\tau\to t})a_0$. Then we can write

$$
\begin{aligned}
\Delta(a_t) = \Delta(\bar{a}_{\tau\to t}) + \theta_*^\top(\bar{a}_{\tau\to t} - a_t)\\
\leq \Delta(\bar{a}_{\tau\to t}) + \widetilde{\theta}_t^\top(\bar{a}_{\tau\to t} - a_t) + M_t(a_t) + M_t(\bar{a}_{\tau\to t})\\
\leq \sigma_{\tau\to t}\Delta(a_\tau) + (1 - \sigma_{\tau\to t})\Delta(a_0) + M_t(a_t) + \sigma_{\tau\to t}M_t(a_\tau) + (1 - \sigma_{\tau\to t})M_t(a_0)\\
\leq (1 - \sigma_{\tau\to t})\Delta(a_0) + M_t(a_t) + \sigma_{\tau\to t}(M_t(a_\tau) + M_\tau(a_\tau)) + (1 - \sigma_{\tau\to t})M_t(a_0),
\end{aligned}
$$

where in the end we used the fact that $\Delta(a_\tau) \leq M_\tau(a_\tau)$. Now,

$$1 - \sigma_{\tau\to t} \leq \frac{3(M_t(a_\tau) + M_\tau(a_\tau))}{\Gamma_0},$$

and of course $M_t(a_0) \leq \Gamma_0$. We end up with a bound of the form

$$\Delta(a_t) \leq C \left(1 + \frac{\Delta(a_0)}{\Gamma(a_0)}\right) (M_t(a_t) + M_\tau(a_\tau) + M_t(a_\tau)),$$

which is essentially the same as that of Lemma 7. Given this, we can immediately invoke Lemma 5 (appropritaely modifying by $a_{\sf safe} \to a_0$ and $\Gamma_0 \to \Gamma(a_0)$). We end up with the control that

$$\sum_{t \leq T, M_t(a_0) \leq \Gamma_0/3} \Delta(a_t) = \widetilde{O}\left(\left(1 + \frac{\Delta(a_0)}{\Gamma(a_0)}\right) \cdot \frac{B_T \omega_T}{\chi} \cdot \sum_{t \leq T} \|a_t\|_{V_t^{-1}}\right).$$

For our choice of noise (being the coupled design executed with $\nu = \mathrm{Unif}(\sqrt{3d}\mathbb{S}^d)$, we have $\chi = \Omega(1), B = O(\sqrt{d})$, and so this can be bounded as

$$\sum_{t \leq T, M_t(a_0) \leq \Gamma_0/3} \Delta(a_t) = \widetilde{O}\left(\mathcal{R}(a_0)d^3T\right).$$

Of course, the above holds true for all $t > t_0(a_0)$ that were not E-steps. Before $t_0(a_0)$, we may bound the per-round regret by 2. Finally, we are left with the E-steps after the time $t_0(a_0)$. Since, as argued above, no extraneous E-steps due to the infeasiblity of perturbed programs occur, we can then, for $T \geq t_0(a_0)$, simply bound the total number of E-steps by $1 + B_T \omega_T \sqrt{dT}$, and accrue roudwise regret of at most 2 in these steps. With our chosen noise, $B_t = O(\sqrt{d})$, this cost is $\widetilde{O}(\sqrt{d^3T})$. Summing these three contributions, and invoking the bound on $t_0(a_0)$ finishes the argument upon recognizing that $a_0$ is arbitrary, and so we may minimise over it.

Turning now to the risk, first observe that for any $t > T_0 := \min_a t_0(a)$, there exists at least one action such that $M_t(a) \leq \Gamma(a)/3$, and so the perturbed program is always feasible, i.e., $a(\eta_t, H_t, t)$ exists. Now, consider subsequent times. Observe that in L-steps, since $\widetilde{\Phi}_t a_t \leq \alpha$, we know by Lemma 3 that $\Phi_* a_t \leq \alpha + M_t(a_t)\mathbf{1}_m$, assuming consistency and the concentration of $\max(\|\eta_t\|, \max_i \|H_t^i\|)$. Thus, in L-steps, the risk accrued at any time is at most $M_t(a_t)$. On the other hand, in E-steps, the risk accumulated can be bounded by just 1 (using the boundedness of $\Phi_*$ and $\mathcal{A}$, and so we only need to work out the total number of these. But after time $T_0$ such an E-step only occurs to make sure that the net number of E-steps is at least $B_t \omega_t \sqrt{dt}$, and so the total number of such steps is at most $B_T \omega_T \sqrt{dT}$.

Putting these together, we conclude that the net risk accrued is bounded as

$$\mathbf{S}_T \leq T_0 + \sum_{\substack{T_0 \leq t \leq T \\ t \text{ is an E-step}}} 1 + \sum_{\substack{t \leq T, \\ t \text{ is an L-step}}} M_t(a_t) = B_T \cdot \widetilde{O}(\sqrt{d^2T} + \min_a \widetilde{O}\left(\frac{dB_t^2 \omega_t^2 \|a\|^4}{\kappa^2 \Gamma(a)^4}\right).$$

Invoking Lemma 14, as well as the fact that $B_T = B_t = \sqrt{3d}$ for our noise design, the claim follows.

Finally, let us account for the probabilistic conditions needed: we need the concentration event of Lemma 5 to hold for the regret bound, and the consistency and noise-boundedness events for both. Of course, the second is not actually needed, since our noise is bounded always. Together, then, these occur with chance at least $1 - 2\delta$ under our noise design. Of course, then, passing to E-COLTS$(\mu, \delta/2)$ yields the claimed result. □

# J  Simulation Study

We conduct simulation studies to investigate the behaviour of E-COLTS/R-COLTS, and of S-COLTS. We first study the soft and hard constraint enforcement problems with our coupled noise design. After this, we investigate the behaviour of COLTS methods using independent (or decoupled) noise in §J.3. All experiments were executed on a consumer-grade laptop computer running a Ryzen-5 chip, in the MATLAB environment, and the total time of all experiments ran to about 8 hours.

## J.1  Soft Constraint Enforcement

We begin with studying the behaviour of the soft constraint enforcement strategies E-COLTS and R-COLTS. Throughout, we treat E-COLTS as R-COLTS$(\mu, 0, \delta)$, with no exploration.

**Setting.** We set $\Phi_*$ to be a certain $9 \times 9$ directed adjacency matrix, $A$, obtained from https://sparse.tamu.edu/vanHeukelum/cage4, which is a $\approx 60\%$ populated matrix with $d = m = 9$. The rows of $\Phi_*$ were normalised to have norm 1. We study the problem of optimising $\theta_* = \mathbf{1}_d/\sqrt{d}$ over $\mathcal{A} = [0, 1/\sqrt{d}]^d$, and enforce the unkonwn constraints $\Phi_* a \le 0.8 \cdot \mathbf{1}/\sqrt{d}$. We note that the action $0$ is always safe, no matter the $\widetilde{\Phi}_t$. This choice is intentional, in that it lets us avoid the inconvenient fixed exploration present in E-COLTS and S-COLTS. Throughout, we set $\delta = 0.1$.

As stated above, for the bulk of this section, we will implement E-COLTS without forced exploration. Indeed, this is not required since $0$ is always feasible, as discussed above. This can equivalently be interpreted as R-COLTS with the resampling parameter $r = 0$.

**Effect of Noise Rate.** As previously noted, in linear TS, small perturbation noise—of the scale $1$ rather than $\Theta(\sqrt{d})$—retains sufficient rates of global optimism and unsaturation to enable good regret behaviour. Note that such a small noise directly reduces $B_T$, and thus we would expect it to improve our regret behaviour by a factor of about $\sqrt{d}$. In order to exploit this, we begin by conducting pilot experiments with our coupled noise design to determine a reasonable noise scale for us to use.

Concretely, we drive our coupled noise design with the laws $\nu_\gamma = \text{Unif}(\gamma \cdot \mathbb{S}^d)$, and run E-COLTS without exploration for $10^3$ steps 100 times. In each run, we simply record whether (i) global optimism; (ii) local optimism; and (iii) unsaturation held, and estimate their rates simply as the fraction of time over the run that this property was true. We construct these rate estimates for $\gamma \in [\sqrt{3d}^{-3}, \sqrt{3d}]$, specifically evaluating the same for 41 values of $\gamma$ chosen over an exponential grid (i.e., so that $\log(\gamma)$ has a constant step). Figure 3 shows the resulting estimates.

The core observation is that global optimism and unsaturation rates are already at $\sim 1$ for $\log(\gamma) \approx -1$, indicating good performance with this noise. Note that while such performance with small noise has been previously observed for linear TS without unknown constraints, we are unaware if an explicit observation of these rates as above has been performed. Of course, proving these properties at such small $\gamma$ is an open question, and we also note that our estimates above are not quite correct, since they integrate the events across time, while their rates could vary with $t$. In any case, the main upshot for this is that in our subsequent experiments, *we work with $\gamma = 0.5$ instead of $\sqrt{3d} \approx 5.2$.*

**The Behaviour of E-COLTS and R-COLTS.** We now study R-COLTS and E-COLTS over the long horizon $T = 5 \cdot 10^4$. We execute R-COLTS with zero resamplings (i.e., E-COLTS with no exploration), and then one and finally two resamplings in each round, all driven by the coupled perturbation noise with $\nu_{0.5}$.

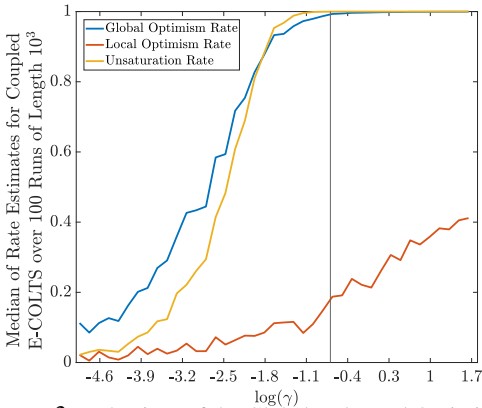

Figure 3: Behaviour of the Global and Local Optimism Rates, as well as the Unsaturation Rate. The black vertical line lies at $\gamma = 0.5$, the value selected for subsequent experimentation. The largest studied value is at $\sqrt{3d}$, which has logarithm about 1.65 Observe that the global optimism and unsaturation rates are significant, and in particular $\approx 1$ for $\gamma = 0.5$, far below $\sqrt{3d} \approx 5.2$.

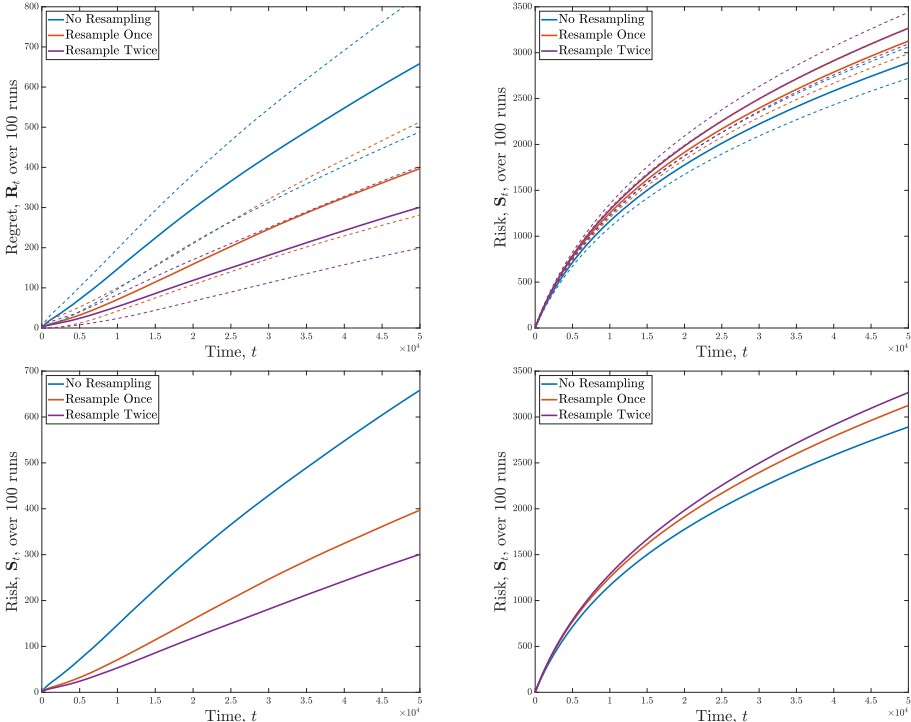

Figure 4: Regret (left) and Risk (right) of R-COLTS with zero, one, and two resamplings per round. Top includes one-sigma error bars, and for clarity, the bottom figures omit them. Note that the regret behaviour is an order of magnitude smaller than the scale $\sqrt{d^2 T} \approx 6600$, while the risk behaviour is about a factor of half of this. We further observe that resampling improves regret signficantly, while only hurting the risks slightly, although this effect appears to decelerate as resampling is increased.

*On* DOSS. We note that DOSS is not implemented. E-COLTS runs in $\sim 10^{-3}$s per round on our machine. (Relaxed)-DOSS is totally impractical: $(2d)^{m+1} > 10^{12}$, and so it needs $> 10^9$s, i.e., years, per round!

*Observations.* Figure 4 shows the observed regret and risk traces over 100 runs. The observed regret behaviour is very strong: even without resampling, the terminal median regret of $\sim 600$ is closer to $\sqrt{T \log T} \approx 750$ than to $\sqrt{d^2 T \log(T)} \approx 6600$. The risk behaviour is more significant, but still half this scale. The observation of $\mathbf{R}_T$ suggests that a stronger regret bound may hold for E-COLTS and R-COLTS, which is in line with the stronger instance-specific regret behaviour of the optimism-based method DOSS [GCS24]. Proving this is an interesting open problem.

These simulations thus bear out the strong performance of E-COLTS/R-COLTS with $r = 0$. Further, as we add resampling, risk degrades mildly, but the regret improves significantly, although the returns diminish with more resampling. This suggests that practically, a few resamplings in R-COLTS are enough to extract most of the advantage. Interestingly, resampling has a palpable effect even though the optimism rate is nearly one!

## J.2 Hard Constraint Enforcement

Next, we investigate the behaviour of S-COLTS over the same instance, supplied with the data $a_{\mathsf{safe}} = 0$. The natural point of comparison to S-COLTS is the SAFE-LTS algorithm [MAAT21], which operates in $O(\mathsf{SOCP} \log t)$ computation per round.[10]

---

[10]We do not implement other prior methods for SLBs, mainly because SAFE-LTS has previously been seen to have similar behaviour, and be about $2d = 18$ times faster than these methods. Of course, we also did not implement DOSS as a comparison for the soft constraint enforcement methods since it is impractical to execute for $d = m = 9$.

Concretely, we again drive this method with $\nu_{0.5}$ as before. For S-COLTS, we sample a perturbed objective vector with the same noise scale, and otherwise optimise over the second order conic constraints as detailed in §4.3. In both cases, we used the library methods `linprog` and `coneprog` provided by MATLAB to implement these methods.[11] Note that these methods are specifically tailored to linear and conic programming respectively. As before, we repeat runs of length $T = 5 \cdot 10^4$ for a total of 100 runs.

*Strong Safety Behaviour.* We note that in all of our runs, we did not observe any constraint violation from either S-COLTS or SAFE-LTS, despite the fact that we executed these methods with $\delta = 0.1$. This suggests both that in practice, the parameter $\delta$ can be relaxed (which would yield mild improvements in regret), and in any case verifies the strong safety properties of these methods.

*Comparison of Regret.* We show the regret traces over the 100 runs in Figure 5. We observe that S-COLTS has a slightly improved regret performance relative to SAFE-LTS, which may be attributed to the selection of stronger exploratory directions through solving the perturbed program.

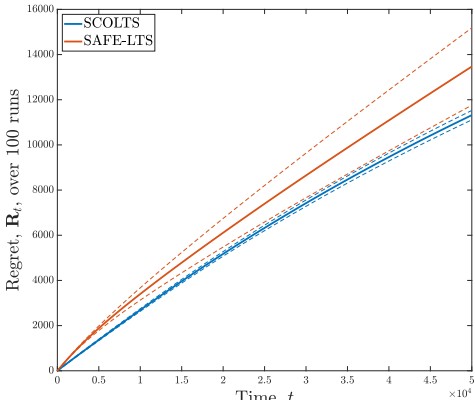

Figure 5: Regret Behaviour of S-COLTS and SAFE-LTS on the same instance as previous figures (one-sigma error curves). We note that S-COLTS offers a mild improvement in regret over SAFE-LTS. However, this comes with a $5\times$ reduction in net computational time per round, which is the main advantage of S-COLTS.

*Computational Speedup.* In wall-clock terms, each iteration of SAFE-LTS is about $5.2\times$ slower than that of S-COLTS on this 9 dimensional instance with 9 unknown constraints (over $5 \cdot 10^6$ total iterations, S-COLTS took about 0.22ms per iteration, while SAFE-LTS took about 1.16ms), a significant computational advantage even in this modest parameter setup.

*High Level Conclusions.* The main takeaway from this set of experiments is that S-COLTS offers tangible benefits in computational time relative to SAFE-LTS (and a fortiori, to other pessimism-optimism based frequentist hard constraint enforcement methods), while even obtaining a slight improvement in the regret behaviour. This demonstrates the utility of S-COLTS over these prior methodologies, and suggests that it is the natural approach that should be used in practice.

### J.2.1 Investigating Behaviour with Increasing $m$

Of course, the computational problem of optimising $m$ SOC constraints becomes harder as $m$ grows, and so we expect that the computational advantage of S-COLTS over SAFE-LTS would grow with $m$.[12] To investigate this hypothesis more closely, we turn to a slightly different setup.

*Setup.* We set $d = 2, \theta_* = (1,0), \mathcal{A} = [-1/\sqrt{d}, 1/\sqrt{d}]^d$. For $m \geq 3$, we impose $m$ unknown constraints such that the feasible region forms a regular $m$-gon with one vertex at $(0.2/\sqrt{2}, 0)$. This allows us to systematically increase $m$ (to very high values) without incurring significant computational costs. We investigate the behaviour of S-COLTS and E-COLTS on this setup with the coupled noise design as in the previous section ($\gamma = 0.5$) for $m \in \{10, 20, \cdots, 100\}$. We also

---

[11] Of course, R-COLTS/E-COLTS were also implemented using `linprog`.

[12] Note that it may be possible to mitigate this somewhat by instead imposing the convex constraint $\max_i (\hat{\Phi}_t a - \alpha)^i + \|a\|_{V_t}^{-1} \leq 0$ to exploit that the same matrix $V_t^{-1}$ appears in all constraints. However, the gradient computation of this map still grows with $m$, so the overall picture is unclear. Of course, imposing only $m$ linear constraints is bound to be faster.

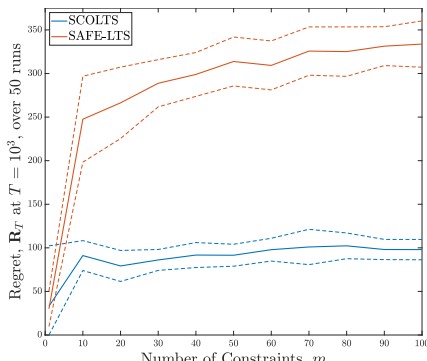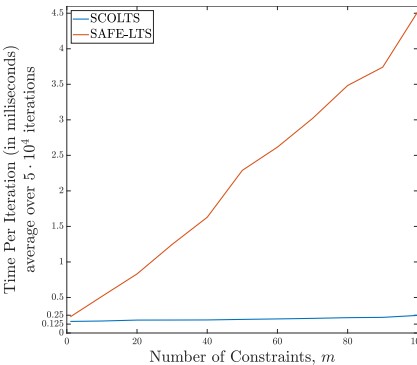

Figure 6: Comparisons of the regret (left, one-sigma error curves) and computational costs (right) of S-COLTS and SAFE-LTS in the $d = 2$ instance as $m$ varies. This is the same setting as Figure 2, right, but presented separately rather than as a ratio. The left plots the regrets at time $T = 10^3$ over 50, and the right plots the wall-clock time per iteration on our resources in milliseconds. S-COLTS needs $0.14 - 0.25$ milliseconds per iteration, while SAFE-LTS needs $> 4.5$ at $m = 100$. At the same time, for $m \geq 10$, the regret of S-COLTS is about $3\times$ smaller.

execute this for $m = 1$, where a single constraint passing through the same vertex is enforced. In all cases, we set $a_{\mathsf{safe}} = 0$, which is always feasible.

*Strong Computatational Speedup.* As seen in Figure 6, S-COLTS has a strong computational advatage, which further grows with $m$. In particular, at $m = 1$, S-COLTS is about $1.3\times$ faster to execute than SAFE-LTS, while for $m = 100$, this advantage grows to $18\times$.

*Improved Regret Performance.*[13] Further, instead of the small gain seen in the previous section, in this problem S-COLTS has a strong statistical advantage relative to SAFE-LTS for even moderate $m$. Indeed, while at $m = 1$, its regret is about $10\%$ larger than that of SAFE-LTS, for larger $m$, its regret is many times *smaller*. In particular, for $m \geq 10$, we found that the regret of S-COLTS is roughly $3\times$ smaller (ranging between $2.7\times$ and $3.4\times$.).

*Takeaways.* This investigation further bolsters the strong advantage of S-COLTS over SAFE-LTS. Note that alternative confidence-set based hard enforcement methods are at least $2d$ times slowed than SAFE-LTS, meaning that the computational advantage of S-COLTS is even stronger relative to these methods. For large $m$, this appears to be accompanied by a large statistical advantage, making this the natural method in applications of SLBs.

## J.3 Simulation Study on the Behaviour of the Decoupled Noise

Finally, we investigate the behaviour of the COLTS framework under the decoupled noise design, wherein, instead of setting $H = -\mathbf{1}_m \eta$, we draw $\eta$, and each row of $H$, independently from $\nu_\gamma$. The main impetus behind this, of course, is that this decoupled design is a natural choice to execute COLTS, although it is contraindicated by the analysis tools available to us.

*Behaviour of Event Rates with $\gamma$.* To begin with, Figure 7 shows the global optimism, local optimism, and unsaturation rates with this decoupled noise for the same instance as previously studied. Observe first that the decoupled noise design does experience a slight decrease in each of these rates compared to those seen in Figure 3. However, this effect is relatively mild, and in particular, we can see that the unsaturation rate is already up to nearly one at our previously selected value of $\gamma = 0.5$. This suggests that the decoupled noise would do nearly as well as the coupled noise in this case.

*Behaviour of Regret and Risk.* To further investigate the above claim, we execute E-COLTS without exploration (or equivalently, R-COLTS with $r = 0$) driven with this decoupled noise over the longer horizon $T = 5 \cdot 10^4$. The resulting regret and risks are plotted in Figure 8, along with the same for E-COLTS with coupled noise. Observe that the decoupled noise sees a significant loss of about $3\times$ in regret, but sees a gain of about $1.5\times$ in risk. Heuristically, we may think of the decoupled noise as

---

[13]Note: for the regret ratio in Figure 2, we perform 100 separate runs with both methods, and compute the ratio of regret for the two methods in each. That figure reports the mean over this data - in this case, the expected mean is $\sim 1.5$ at $m = 1$, but with wide confidence intervals (CIs). For $m \geq 10$, the lower confidence bounds all exceed 2. At $m = 1$, the mean regret of SAFE-LTS is about $0.91\times$ that of S-COLTS, with strongly overlapping CIs.

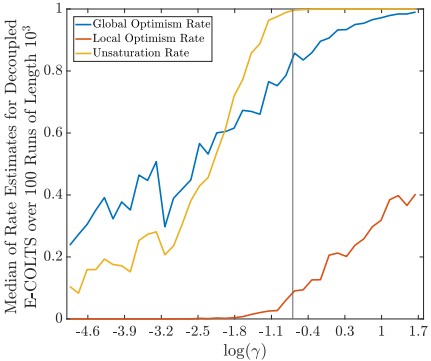

Figure 7: Behaviour of the Global Optimism, Local Optimism, and Unsaturation Rates with $\gamma$ for the Decoupled Noise in the setting of Figure 3. Observe that while these rates decay somewhat with respect to the coupled noise, they are still strong, and especially for large $\gamma$ are nearly as good as with the coupled noise.

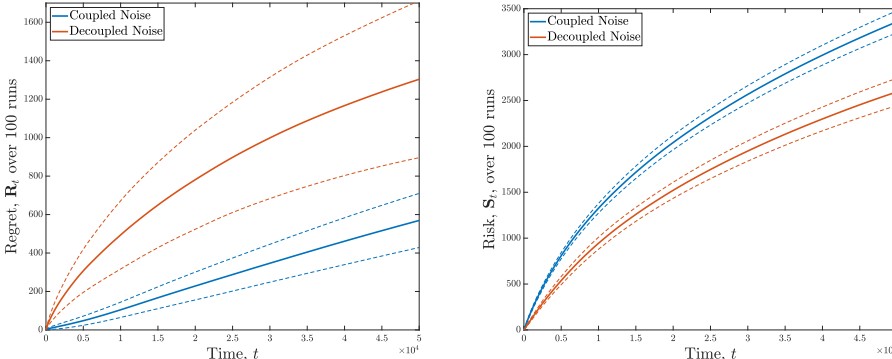

Figure 8: Behaviour of regret (left) and risk (right) for E-COLTS executed with the decoupled noise compared with E-COLTS executed with the coupled noise (one-sigma error bars). Observe that the regret behaviour sharply deteriorates, while the risk behaviour slightly improves for the decoupled noise design. Heuristically, this suggests that the decoupled noise behaves 'like' the coupled noise, but with a smaller value of $\gamma$.

behaving as if the noise is coupled but "shrunk", so that the behaviour of the risk is improved, but the behaviour of the regret worsens.

Practically speaking, our recommendation remains to use the coupled noise design, in that it attains higher rates of explanatory events, and carries theoretical guarantees. Nevertheless, establishing that $\mathbf{R}_T$ and $\mathbf{S}_T$ do scale sublinearly with the decoupled noise design, as is evident from Figure 8, is an interesting open problem.

### J.3.1 Investigation of Rates with Increasing $m$

Of course, the main obstruction with the use of the decoupled noise in §4.2 was to do with many constraints. Indeed, it should be clear that under this decoupled noise, the local optimism rate must decay exponentially with $m$, since if any row of $\widetilde{\Phi}_t$ is perturbed so that $a_*$ violates its constraints, local optimism would fail (and this would occcur with a constant chance, no matter the estimates).

To probe whether this indeed occurs, we simulate the behaviour of E-COLTS with the coupled and decoupled noise designs on a simplified setup.

*Setup.* We again take the $d = 2$ polygonal constraints investigated in §J.2.1. We investigate the behaviour of E-COLTS with both the coupled and decoupled noise designs on this instance as $m \in \{10, 20, \dots, 100\} \cup \{200, 300, \dots, 1000\}$, thus letting us probe an extremely high number of unknown constraints.

*Observations.* There are two main observations of Figure 9. Firstly, note that as shown in the main text, the rates of optimism and unsaturation under the coupled noise design are stable, and do not meaningfully vary with $m$ after it has grown at least slightly.

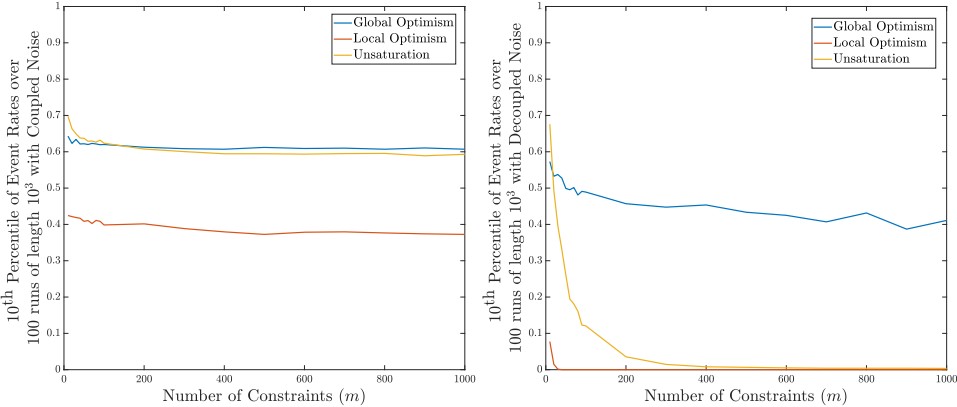

Figure 9: Behaviour of the rates of global and local optimism, and of unsaturation, in the polygonal instances as the number of constraints is increased for the coupled (left) and decoupled (right) noise designs driven by $\mathrm{Unif}(\mathbb{S}^2)$. Observed that the behaviour of these is stable with $m$ for the coupled design, but for the decoupled design, the local optimism and unsaturation rate decay with $m$. Surprisingly, the global optimism rate remains stable even for the decoupled noise design.

On the other hand, under the decoupled noise design, the local optimism rate clearly crashes exponentially. The unsaturation rate has a slower but evident decay: roughly, this is as $m^{-1.3}$ for $m \leq 100$, and appears to be exponential for large $m$. However, surprisingly, the *global optimism* rate remains stable (although lower than the same with the coupled design). This shows that there are situations with low-regret where frequent global optimism would be the 'correct' explanation for good performance of methods like S-COLTS or E-COLTS (indeed, this is what prompted us to write the optimism based analysis of these methods in §H.5). Note however that *proving* that global optimism is frequent under the decoupled design is an open problem. In fact, with unknown constraints, we do not know of any method to deal with global optimism lower bounds that does not pass through local optimism, since the approach of Abeille & Lazaric [AL17] relies on convexity properties of the value function in terms of the unknown parameters, which fails in this case.

