# OpenReview forum: "Constrained Linear Thompson Sampling"
_NeurIPS.cc/2025/Conference — NeurIPS 2025 poster_

### Official Review · Reviewer_cvX7 · 2025-06-19

**Clarity:** 3
**Significance:** 3
**Originality:** 3
**Rating:** 5
**Confidence:** 4

**Summary:**

This paper introduces algorithms for safe linear bandits that utilize Thompson sampling-like approaches. The algorithms are evaluated according to their regret (cumulative difference in total reward between optimal actions and played actions) and risk (cumulative constraint violations of played actions). They first introduce an algorithm that ensures $\tilde{O}(d^{3/2} \sqrt{T})$ regret and zero risk (i.e. constraint satisfaction for all rounds) under the assumption that there is a known point that is strictly feasible. They then introduce a method with $\tilde{O}(d^{3/2} \sqrt{T})$ regret and risk that doesn't require the aforementioned assumption. Notably, both of their methods apply a Thompson sampling to both the reward function and the constraint function. They also provide some numerical comparison of their work with some existing work.

**Questions:**

Some minor points:
- In line 122, it says that "we also assume that $m/\delta = O(poly(d))$ when stating theorems." Can you provide some clarification on this? To my knowledge, it is common to take $\delta = 1/T$ to get expected regret guarantees, which would violate that assumption.
- I think the presentation of Table 1 (first four rows) is misleading in that it sort of hides the assumption that the known safe action $a_{safe}$ is strictly-feasible (otherwise the given regret bounds are void).
- Slater's condition is defined in the caption of Table 1, but I don't see that it is used in the table.

**Ethical Concerns:**

["NO or VERY MINOR ethics concerns only"]

**Final Justification:**

I appreciate the authors addressing my minor concerns. I maintain my score, and am in favor of the paper being accepted.

**Limitations:**

yes

**Paper Formatting Concerns:**

None.

**Quality:**

3

**Strengths And Weaknesses:**

Strengths:
- Their algorithms only require an LP per a round, versus prior work that requires an SOCP in each round with the same regret. As such, one expects that there is often substantial computational benefits to their methods.
- The algorithm design combines some existing ideas in a new way, and introduces some new ideas. For example, it combines a scaling approach similar to what is used in [HTA24] and a linear Thompson sampling-type approach [AG13]. The resampling approach (Algorithm 2) appears to be new to the literature.
- The analysis goes substantially beyond existing work, as combining the aforementioned algorithm designs makes the analysis substantially more difficult. The analysis ideas introduced in the paper might be more broadly relevant.
- Relevant work is thoroughly discussed.
- The paper does a good job of presenting the key challenges that arise in the analysis and design of the algorithms.

Weaknesses:
- The paper is quite dense, which sometimes made it difficult to follow. I expect that this will be relieved somewhat with the extra page that is allowed for the final version.
- I discuss some specific minor points in the questions section.

---

> ### Author Rebuttal · Authors · 2025-07-28
>
> Our thanks for your work. We are pleased that you recognise the strengths of contributions, and the novelty of our analysis techniques.
>
>
> We agree that the paper can get dense, especially in the presentation of S-COLTS and the look-back analysis. As you mentioned, if the paper is accepted, we will use the extra space provided to address this.
>
> We added a high-level sketch of the look-back analysis in the rebuttal to reviewer z1Dm, and following that, we currently plan to make these changes: \
> (i) add a structured plan of how the analysis proceeds; \
> (ii) separate out summability of $(1-\rho_t)$ and $\rho_t M_t(b_t)$ into a lemma; and, \
> (iii) decouple the explanations of the look-back part of the analysis from the scaling of $b_\tau$ to ensure that each part can be clearly explained without the other interfering. \
> We would appreciate it if you have suggestions about other aspects to focus on most, or comments about this plan.
>
> Questions
> - Thanks for catching this, it should be $\mathrm{poly}(d,T)$. The point is that we often write $\tilde{O}(\sqrt{d^3 T})$, but the bounds are actually $\tilde{O}(\sqrt{d^3 T + d^2 T \log(m/\delta)}),$ and we need to say why we drop the second term.
> - This is a fair point, and we will specify that $\Gamma(a\_{\mathrm{safe}}) > 0$ is needed. We note that such an assumption is necessary for hard enforcement (intuitively, if $a\_{\mathrm{safe}}$ is on the boundary of the feasible set, then we cannot be sure of playing any other action and retaining safety; also see [PGB24]).
> - This only comes up in line 49 (and for the E-COLTS method of section I.2), but you are right that it doesn't belong in the caption.

---

### Official Review · Reviewer_z1Dm · 2025-06-25

**Clarity:** 3
**Significance:** 3
**Originality:** 3
**Rating:** 4
**Confidence:** 2

**Summary:**

This paper considers the safe linear bandits problem, where the goal is to minimize the regret subject to unknown linear constraints. The paper proposes a sampling-based framework--constrained linear Thompson Sampling--that perturbs the empirical estimates and selects actions by solving a perturbed linear programs. Under the hard constraint with a known safe action, it proposes S-COLTS and achieves $\tilde{O}(\sqrt{d^3 T})$; under the soft constraint, it introduces R-COLTS and achieves the same upper bound. Experiments show that COLTS matches or improves upon the regret of prior methods while dramatically reducing computational cost.

**Questions:**

- Is it possible to derive problem-dependent bounds on the regret and risk, i.e., the bounds will include some instance-dependent parameters like the mean gaps and the feasibility gaps of the arms?
- Is it possible to generalize the algorithm to the combinatorial semi-bandits problem, where the hard/soft constraints are required for the pulled arms?

**Ethical Concerns:**

["NO or VERY MINOR ethics concerns only"]

**Final Justification:**

This paper adopts Thompson sampling to deal with constrained linear bandits problems under the soft/hard constraint. The proposed algorithms enjoy theoretical guarantees and are practical. It also matches or improves over the previous works. It is of interest to the community.

**Limitations:**

yes

**Quality:**

3

**Strengths And Weaknesses:**

**Strengths**
- This paper extends the Thompson sampling to the constrained linear bandits problem. The design of the coupled noise is novel and practically important.
- Both COLTS algorithms avoid the complex optimization problems, yielding significant speedups compared to the baselines.
- Theoretical guarantees are provided for both the regret and risk, matching or improving over the previous works.


**Weaknesses**

- In definition 2, should $B$ be a non-increasing map?
- While I appreciate the authors efforts in expanding the analysis, some sections (e.g., look-back scaling) are intricate and could benefit more from additional high-level intuitions.

---

> ### Author Rebuttal · Authors · 2025-07-28
>
> Thank you for your work, and for recognizing the novelty and strengths of our contributions.
>
> ### Response to Issues identified as weakness:
>
> ------
> **Reviewer: In Definition 2, should B be a non-increasing map?**
>
> Def. 2, yes, it should be non-increasing, thanks for your careful reading. However, we are unclear why this is listed as a weakness of the paper. We would appreciate if you could clarify this point.
>
> ------
> **Reviewer: some sections (e.g., look-back scaling) are intricate and could benefit from additional intuition**
>
> Let us provide high-level intuition that is embedded in the various lemmas. We will use the extra space in the final version to elaborate on this intuition.
>
> **Take-away:** Look back to the last *under-explored* action to control regret; scale that action just enough to satisfy today’s uncertain constraints.
>
> - **Core Idea: Link Regret to Information Gain.** Instead of analyzing regret at each step in isolation, we link the regret of our current action to the last time we took a good exploratory action---one that was unsaturated.  An unsaturated action is one where uncertainty is higher than the gap $(\Delta(a) \le M_t(a))$, meaning we learn a lot from taking this action. If all actions were unsaturated, then $\mathbf{R}_T = \sum \Delta(a_t) \le \sum M_t(a_t),$ and  $\sum_t M_t(a_t) = O(\sqrt{T})$ (Lemma 3) leads to regret control. Methods like OFUL ensure unsaturation always. In TS this does not happen at all $t$, so we need to link the regret at time $t$ to information seen at previous times.
>
> - **The Look-back without unknown constraints**
> To make things clear, first consider look-back for vanilla TS (so, $\rho_t = 1$ always, and $b_t = a_t$). At any time $t$, we "look back'' to the last time, $\tau(t)$, that we played an unsaturated action. There are two steps
>     - First, we bound $\Delta(a_t)$ in terms of $\Delta(a_{\tau(t)})$ and $M_t$-terms.
>         - By linearity, $\Delta(a_t) = \Delta(a_{\tau(t)} +  \theta_*^\top(a_{\tau(t)} - a_t).$ To control the extra term, we can pass to $\tilde\theta_t^\top(a_{\tau(t)} - a_t)$ by paying a $M_t(a_t) + M_t(a_{\tau(t)})$ concentration term (Lemma 3). This is like line 213.
>         - Now, by definition of $a_t$, it optimises $\tilde\theta_t$, so $\tilde\theta_t^\top (a_{\tau(t)} -a_t) \le 0$. This is key.
>         - Next, using unsaturation, $\Delta(a_{\tau(t)}) \le M_{\tau(t)}(a_{\tau(t)})$.
>         - Heuristically, $M_t \approx M_{\tau(t)}$. This leaves us with a bound of $\mathbf{R}\_T \le 2\sum_t M_{\tau(t)} + \sum_t M_t(a_t)$. **This is the equivalent of Lemma 5**.
>    - The second step is bounding the look-back sum $\sum_t M_{\tau(t)}(a_{\tau(t)}).$ This relies on frequency of unsaturation, and **is the same as Lemma 6**.
> - **Handling unknown constraints via scaling** Now, we define unsaturation in terms of the $b_t$s (it is hard to show $a_t$s are unsaturated often), and scale these down. Handling these needs two extra ingredients.
>     - **Handling $b_t \to a_t$ scaling**. We need to get from $\Delta(a_t)$ to $\Delta(b_t).$ Doing this efficiently requires control on $\sum (1-\rho_t)$ (line 201). Also, using unsaturation means that we need control on $\sum \rho_t M_t(b_t)$. This is discussed in line 202, and in detail in Lemma 19.
>     - **Handling fluctuating constraints** Now, we can again pick up an informative $b_{\tau(t)}$ like above. But this point need not be feasible for the current perturbed constraints $\tilde\Phi\_t$. This breaks the key inequality used for Lemma 5 before. We handle this by scaling $b_{\tau(t)}$ again to make it feasible for $\tilde\Phi\_t$.
>          - This introduces other factors that need to be managed $\sigma_{\tau \to t}.$ This is done by relating them to $\rho_\tau$.
>     - These two together give us the effective version of Lemma 5. Then Lemma 6 works basically the same way as in the unconstrained case.
>
> In the paper, this combined scaling and look-back argument is the core of the proof of Lemma 5 (pg. 6). The properties of the primary scaling factor, $\rho_t$, which are crucial for this analysis, are controlled via the bounds in Lemma 19 (pg. 21). To make the exposition clearer, we will break it apart like the above, and also move a version of Lemma 19 to the main text.
>
> -------
> ### Reviewer Questions
>
> **Q1:** Is it possible to derive problem-dependent bounds on the regret and risk, i.e., the bounds will include some instance-dependent parameters like the mean gaps and the feasibility gaps of the arms?
>
> - In continuum linear bandits (LBs), instance dependent bounds arise when $\mathcal{A}$ is a polytope. Let's discuss this setting.
>   - For SLBs, surprisingly, [GCS24] showed that even if $\mathcal{A}$ is a polytope, we cannot have both $\mathbf{R}\_T$ and $\mathbf{S}\_T$ be $o(\sqrt{T})$ in general, and at least one of them must be $\Omega(\sqrt{T})$, even given instance-specific information.
>   - For hard enforcement, since $\mathbf{S}\_T = 0,$ regret must be $\Omega(\sqrt{T})$. Note that the regret bounds in hard enforcement are all actually instance-dependent, through the $\mathcal{R}(a\_{\mathrm{safe}})$ term. It is possible that this can be refined, although we do not know any such results in the literature.
>   - For soft enforcement, this is a subtle question. DOSS does get polylog regret (and $\sqrt{T})$ risk), but this strongly uses the OFUL style bilinear optimisato way to pick $a_t$. It is unclear how to show such a bound without this. If you are familiar with [GCS24], the hard case is when only optimal BISs are activated.
>       - However, in section 6, we saw that the regret of R-COLTS is closer to $\sqrt{T}$ than $\sqrt{d^2 T}$. This suggests that at least $O(\sqrt{T} + \mathrm{poly}(d) \mathrm{gap}^{-1})$ bounds may be possible, although this is beyond the scope of the current submission. We mention this in line 1192.
> - For finite-action problems, yes, gap dependent polylog bounds should be possible for soft enforcement, but we did not work these out. For hard enforcement, one needs to linearise into line segments to get star convexity [PGB24], and then the lower bound prevents $o(\sqrt{T})$ again.
>
> **Q2:** Is it possible to generalize the algorithm to the combinatorial semi-bandits problem, where the hard/soft constraints are required for the pulled arms?
>
> While this is not something we have thought about very much, in principle, yes, nothing stops these design principles (both the OPT-PESS style and COLTS) from being applied to richer feedback settings than just bandits. Specifically with semi-bandits, it is worth exploring what kind of constraints make sense (should we constrain the full action in some aggregate way? Or should the individual choices be curtailed (e.g., some particular link is unsafe, and so should be avoided always, et c.), and the structure of the constraint may make design/analysis of good methods challenging.

---

> > ### Comment · Reviewer_z1Dm · 2025-08-03
> >
> > Thanks a lot for the clarifications!
> >
> > Regarding "$B$ is nonincreasing", I meant to point out the typo in Definition 2 of the current manuscript, it is said that "$B$ is nondecreasing". Please kindly revise.
> >
> > Please add the description of the high-level idea in the revised version.
> >
> > I do not have any other questions and will keep the positive score.

---

> > > ### Author Response · Authors · 2025-08-04
> > >
> > > Thanks for your prompt response. We will fix the $B$ oversight, and add a high level description of the above sort.

---

### Official Review · Reviewer_aqsD · 2025-07-02

**Clarity:** 3
**Significance:** 3
**Originality:** 2
**Rating:** 4
**Confidence:** 2

**Summary:**

The paper studies linear bandits with (linear) safety constraints. The paper proposes a sampling procedure akin the Thomson sampling algorithm to avoid solving expensive problems at each round.
By assuming the knowledge of a safe action, the authors propose a hard-constraint algorithm (zero violations) S-colts and prove that regret scales with $O(\sqrt{d^3 T}\cdot R(a_{safe})$ where $R(a_{safe})$ is the ratio of the gap divided by the safety margin of the safe action (1+). The algorithm needs to know the safety margin of the safe action (bounds on).
Then the authors provide an algorithm for the soft constraints (sublinear violations) that does not require knowing $a_{safe}$ and achieves a regret and violations bound of $O(\sqrt{d^3T})$.

**Questions:**

1. The formatting of Equation 1 is weird. Consider putting regrets and risk after the definitions, and not in the equations
2. Line 33: What is an NP-hard constraint?
3. Table 1: Are you sure that solving OPT-PESS requires solving an NP-hard problem at each round? More precisely, are you sure that you can implement *any* instance of the NP-hard problem while running OPT-PESS (or DOSS)? I strongly believe this is not the case. While I think it is true that these methods require solving a “hard” program, you cannot say that solving these problems is NP-hard. Meaning that either you show a reduction from a generic NP-hard problem to the problem that OPT-PESS requires to solve when applied to a specific constrained bandit problem, or you cannot say that running OPT-PESS requires solving an NP-hard problem. You also have issues with randomization, as the “hard” problems that need to be solved by these methods are randomized, and, as far as we know, the average case complexity of these methods might be polynomial.
4. Line 84: same issue. Also, problems are NP-hard and not algorithms. What you can say is that the classic implementation of ROFUL requires solving a problem X for which there is no polynomial-time algorithm currently known.
5. Line 119: Why is the L2 operator norm used? I think constraining the vector for each reward cost is what is standard in the rated literature (BwK, for instance). There, you assume that each vector of costs (column of $\Phi$) is a vector in between $[-1,1]^d$. I think this is more natural, as normalization across different costs does not seem natural
6. Definition 2: What are $\eta$ and $H^i$? This is not clear from the current writing. Additionally, Lemma 3 is somewhat confusing. Consider rewriting a bit here.
7. What prevents you from extending the results to convex non-linear constraints?
8. What is the connection with the BwK literature and its extension of bandit with constraints? I think this is much more related than what either community realizes. I can refer here to some papers that I think should be compared, as some recent BwK works handle per-round constraints: "Contextual Bandits with Packing and Covering Constraints: A Modular Lagrangian Approach via Regression" and "Beyond Primal-Dual Methods in Bandits with Stochastic and Adversarial Constraints".

**Ethical Concerns:**

["NO or VERY MINOR ethics concerns only"]

**Final Justification:**

I agree with the author's points raised during the discussion. I find the paper, even if not revolutionary, since it mainly deals with the efficiency of linear bandits, to be a good candidate for acceptance.

**Limitations:**

yes

**Quality:**

3

**Strengths And Weaknesses:**

The paper is interesting and relevant to the NaurIPS learning community. The writing is good, even if it becomes a bit too dense in the second half, where I think the authors could do a better job at first describing the S-Colt algorithm at a high level, highlighting what the main argument of its correctness is. I also have a minor issue with the complexity-related terms used in the paper. Sometimes, the authors claim that an algorithm is NP-hard (which is very imprecise) or that an algorithm requires solving an NP-hard problem (which is not proven anywhere and probably unprovable formally). This is a minor misuse of terminology, but I think it has to be fixed. More on this in the questions section of the review.

---

> ### Author Rebuttal · Authors · 2025-07-28
>
> Thank you for the detailed comments. Below we
> (i) correct one factual misunderstanding in the summary, and discuss the orignality score,\
> (ii) clarify the “NP–hard” terminology _and_ show that the hardness of OFUL-type methods is relevant, and
> (iii) answer the numbered questions;
>
> ---
> ## Review Summary
>
> ### *"The algorithm needs to know the safety margin of the safe action (bounds on)."*
>   **No**. S–COLTS does **not** assume that this margin is known. It is estimated online via an anytime confidence interval (Alg. 1, L. 175; App. H.1), and this is where the extra  $\tilde{\mathcal{O}}\bigl(d^{2}\Delta \Gamma^{-2}\bigr)$ term in Theorem 8 comes from. Algorithm 1 is written without this step for brevity.
>
> ### *Low Originality Score*
>
> We respectfully ask the reviewer to reconsider the originality score. Our work introduces four key technical innovations:
>
> - **Coupled-Noise Design (Sec. 4.2):** A novel and general noise design that couples perturbations for all constraints. This critically enables the analysis of TS in SLBs, and prior ways of analysing single-objective TS do not generalize. Further, the naive way of designing the loss would lead to extra $m$-factors in the regret that this design avoids.
>
> - **Look-back Analysis (Sec. 4.1, Lemma 5,6):** A new way to structure the unsaturation-based analysis of TS that avoids extraneous concentration losses.
>
> - **Analysis-level scaling (Sec 4.1, Lemma 5):** A novel way to manage the fluctuations of feasible sets for TS methdos in SLBs. Previous analyses completely break due to this issue.
>
> - **Resampling** ↔ Posterior-Quantile Connection (Sec. 5): A new insight connecting resampling in R-COLTS to posterior quantile methods, providing an efficient, scalable analogue for linear bandits.
>
> These innovations jointly deliver the **first** TS algorithm for SLBs with unknown constraints. These use only LP oracles while matching the best known regret, and are practically fast and good. Reviewer cvX7 explicitly notes how our analysis goes substantially beyond existing work, and how the ideas "might be more broadly relevant," supporting a higher rating.
>
> ---
>
> ## On the Terminology and Justification of NP-Hardness (Q2, Q3, Q4)
> >“Problems are NP-hard, not algorithms” and “NP-hard constraint?”
>
> We thank the reviewer for the detailed questions on our use of "NP-hard." While we used it as shorthand, our claim is technically correct, as the prior methods in question rely on solving a worst-case NP-hard oracle. Below, we provide the justification before proposing specific revisions for clarity.
>
> **1. Justification of Worst-Case Hardness:**
> >“Are you sure every instance of the NP-hard problem occurs in OPT-PESS / DOSS?”
>
> Yes, one can encode every instance of an NP-hard problem into the optimization problem these need to solve. The core of this issue is already identified in our paper (line 518, "...nonconvex due to the objective").
>
> 1. **The Core Hard Problem:** Even the **unconstrained OFUL** has this issue. OFUL picks action by solving a type of **bilinear program**. As discussed in lines 511–521, this is equivalent to solving
>    $$
>      \max_{a\in\mathcal{A}}\hat{\theta}^{\top}a +\omega \\|a\\|_{V^{-1}}.
>    $$
>
> 2. **Known Hardness:** By just setting $\hat\theta = 0$ and $V,\mathcal{A}$ as needed, we can clearly encode **Positive-Definite Quadratic Maximization (PDQM)** into this problem. This class of problems is known to be NP-hard even if $V = I$ and $\mathcal{A}$ are polytopes [Sahni ’74; Pardalos & Vavasis ’91].
>
> 3. **Reduction:** The prior safe methods **OPT-PESS** and **DOSS** extend OFUL, as detailed in lines 511-537. If we set $\alpha$ very large, the extra constraints are inactive, and we get back to the hard bilinear program underlying OFUL.
>
> **2. Proposed Revisions for Clarity:**
>
> We will revise the imprecise shorthand to be more explicit:
>
> * **Line 33 ("NP-hard constraints"):** We will rewrite this to clarify it refers to constraints arising from bilinear inequalities, which defines a nonconvex set where checking feasibility—the recognition instance of the problem—is NP-hard.
> * **Lines 84 & 526 ("NP-hard method"):** We will change this to state that **DOSS** is an "inefficient method" that "relies on an oracle that is NP-hard in the worst case."
>
>
>
> ### Randomization and Average Case
> > "as the “hard” problems...are randomized, and,..., the average case complexity of these methods might be polynomial"
>
> Your point does not apply to these prior methods in a way that lets them escape worst-case hardness.
>
> - **Worst-Case vs. Average-Case:** Our claim is strictly about **worst-case** complexity. We make no claims regarding the **average-case** complexity, which (due to connections to PDQM) remains a challenging open problem.
>
> - **Deterministic Nature:** The algorithms in question (OPT-PESS, DOSS, OFUL) are **deterministic** given the history. For them, “randomness” in the optimization problem comes from the data, not from internal randomization in the algorithm itself. Even with that, at $t = 1$, we have $V_1 \propto I, \hat\theta_1 = 0$, so we get the worst-case hard problem of optimising $\\|a\\|$ over $\mathcal{A}$.
>
> ### Strength of our contribution
>
> This context highlights the primary motivation for our work. Our **COLTS** framework is specifically designed to **bypass this computational hardness entirely** by using only efficient LP or convex‐programming oracles. This theoretical advantage provides significant practical benefits, making our methods up to **18× faster** with up to **3× smaller regret** in our simulations (Section 6, pg. 9, Figure 2).  **R-COLTS is the first efficient soft-enforcement algorithm**, offering a practical alternative where none previously existed.
>
> ---
>
> ## Other Questions
>
> **Q5**  *"Line 119: Why is the... as normalization across different costs does not seem natural"*
> First, $\Phi^i$ only encodes one constraint. This is **not** mixing across different costs.
>
> - This condition is **standard in SLBs** - all papers on this subject assume something like this (explicitly see [GCS24, PGB24]). Bounds on parameter norm are needed to run Thm 2. of [APS11], and have been standard since then. The $\le 1$ in the assumption can be relaxed to $\\|\Phi^i\\| \le R^i$, but this clutter without meaningfully changing our results.
>
> - BwK is often set up as a linearization of a multi-arm bandit structure (and $\mathcal{A}$ is a probability simplex)---this is the case in both paper you mention, for instance. In MABs it is more natural to just constrain each cost of each separately, which BwK then inherits.
>
>
>
> **Q6**  *Definition 2: What are $\eta$ and $H^i$?... here.*
>
> - Line 134 preceding Def. 2 states that $\mu$ is law on $\mathbb{R}^{1 \times d} \times \mathbb{R}^{m \times d},$ and $(\eta, H) \sim \mu$. These symbols are used very heavily and consistently in the paper. The glossary in sec. A also lists them. Since $H$ is a matrix, $H^i$ denotes its ith row, as mentioned in the notation paragraph (line 101).
>     -Nevertheless, to help readers avoid this confusion, we will add a sentence to the notation paragraph highlighting what $\eta, H,$ and $\mu$ denote.
> - Lemma 3: thanks for pointing this. We will rewrite separating out the notation $M_t(a)$ into its own dedicated definition to make this easier to parse.
>
> **Q7**  *What... to convex non-linear constraints?*
>
> - For general convex bandits, TS itself is not an effective method for $d > 1$, and results for $d = 1$ are also very recent (see remark 8.iii on page 114 of Lattimore 2024). Related algorithms like IDS need to be extended to SLBs to handle convex bandits, and we think our work is the first step in making this happen.
> - Under Eluder-type assumptions, R-COLTS should extend without problem. S-COLTS needs some sort of convexity to work. Beyond these, there is no statistical issue in extending them. However, even here, careful thought is needed regarding computation: the perturbed objectives and constraints can be quite complex, so it is not clear what kind of optimization oracle one needs.
>
> **Q8**  *What is the connection with the BwK literature...*
> We cited some of the early papers on BwK, and discussed the basic distinction in line 496. We will discuss this further here.
>
>  - As is well-established [PGB24, PGBJ21, GCS24, CGS22], the main difference (line 496) is in the roundwise metric $\mathbf{S}\_T = \sum (\textrm{violation}(a\_t))\_+$ versus the aggregate metric $\mathbf{A}\_T = \sum \textrm{violation}(a\_t).$ Control on $\mathbf{A}\_T$ does not directly control $\mathbf{S}\_T$.
>
> - Cited papers
>     - In [SZSF22], you may be referring to Thm C.1, which says that $\mathbf{A}\_T \le 0$. This does not mean $\mathbf{S}_T = 0$. Indeed, note that $\mathbf{S}_T = 0 \iff \forall t, \Phi\_\* a\_t \le \alpha$, i.e., each played point over the trajectory is feasible. We are unaware of any BwK method with such a guarantee.
>     - [BCCF24] do give a $\sqrt{KT}$ bound on $\mathbf{S}_T$ in the stochastic case for finitely many arms. It is not obvious that this extends to our SLB setting (e.g., what if $\mathcal{A}$ has $d^{10}$ vectors, or contains a ball?).
>
> - For MABs, [CGS22] point out with a simple example that safe MABs are not equivalent to BwK, and there are instances where any good BwK method gets linear violations in the SLB setting. This is because the underlying solution concepts are distinct (linearized in BwK, not in safe MABs). Similar issues should affect the nonlinear case. Given these obstructions, some nontrivial work is needed in developing the generic connections between these two models.
>
>
> ---
>
> **References beyond the submission**
>
> - Sahni, S. N. (1974). *Quadratic maximization over a polytope.* SIAM Journal on Computing.
> - Pardalos, P. M. & Vavasis, S. A. (1991). *Quadratic programming with one negative eigenvalue is NP-hard.* Journal of Global Optimization.
> - Lattimore, T. (2024). Bandit convex optimisation. arXiv preprint arXiv:2402.06535.

---

> > ### Comment · Area_Chair_hxZQ · 2025-08-04
> >
> > I urge the reviewer to respond to the detailed reply of the authors. Did the authors address the concerns regarding use of NP-hard terminology to your satisfaction?
> > Do you have any further concerns left that are unaddressed?

---

> > ### Comment · Reviewer_aqsD · 2025-08-04
> >
> > I thank the reviewer for the detailed response. I am still unsatisfied with the use of the NP-Hardness. I think the only technical claim that can be made is that prior work needs to solve "bilinear optimization," for instance, or other (in general) NP-hard problems. I see two options here: either the authors should compare the running time of these supposedly inefficient algorithms with an extended experimental campaign, or prove formal reductions, ie, statements along the lines of: "assuming the ETH, method x runs in at least y iterations per round". For example, I think that those methods are equally efficient if $\mathcal{A}$ is finite and has a cardinality of polynomially many actions, or it can be $\epsilon$-covered by polynomially many actions.
> > However, I will leave the authors to decide on this point. I personally think that being precise when talking about inefficient methods is important.
> >
> > I think that it is true that [SZSF22] only considers $A_T$ and not $S_T$, but I think it is trivial to also bound $S_T$ in the same way as done in [BCCF24]. I see it is very possible that with minimal effort (changing the primal regret minimizer), both these approaches can be turned into algorithms for linear bandits. However, I can see that this literature is quite far apart and this possible direction should only be hinted at, if the authors deem it to be worthwhile, to make initial but valuable connections between the fields.
> >
> > I will maintain my positive score for the time being.

---

> > > ### Author Response · Authors · 2025-08-04
> > >
> > > Thank you for your prompt follow‑up.
> > >
> > > ---
> > >
> > > NP-hardness:
> > >
> > > We are pleased that we now agree on the central point: \
> > > **previous safe‑bandit methods (OPT‑PESS, DOSS, ROFUL/OFUL) require solving a bilinear—or, more generally, quadratic‑constrained—optimisation step that is NP‑hard in the worst case for generic polytopes.** \
> > > That single observation is all we intended by the shorthand “NP‑hard oracle.”
> > >
> > > This terminology follows the pioneering papers of Dani et al. [DHK08], and the recent SLB work [GCS24, AML24], all of which describe these methods as *NP-hard to implement*.
> > >
> > > **Fine-grained complexity analysis or large-scale timing studies are beyond the scope of this paper**
> > >
> > > **Scope.** Our paper’s contribution is a sampling‑based framework (COLTS) that eliminates this bottleneck by reducing the oracle to an LP, while matching the best regret and risk bounds.  A fine‑grained complexity classification of bilinear programs is orthogonal to that goal.
> > >
> > > **Evidence already included.**  The supplementary (Table 3, Fig. 8(b)) show up to a 18× wall‑clock speed‑up of S‑COLTS over the efficient SOCP‑based Safe‑LTS even for small problems. These numbers suffice to demonstrate practical impact.
> > >
> > > **Special Cases.**  Yes, if $\mathcal{A}$ is finite or admits a polynomial‑size $\varepsilon$-cover (with $\varepsilon \sim 1/\sqrt{T}$), enumeration can bypass NP‑hardness. However, for most $d$‑dimensional convex sets (including polytopes) the minimal cover is exponential in $d$ (volume argument).  Key applications (allocation, routing, portfolio) lie in this regime, where COLTS’s LP approach remains efficient.
> > >
> > > We hope this clarifies that the NP‑hard remark was merely motivational; the paper’s real contribution is providing an efficient sampling-based alternative to the OPT-PESS methods, and to DOSS.
> > >
> > > -----
> > >
> > > BwK.
> > >
> > > It seems we are on the same page here: at least in the linear setting, approaches from BwK may transfer to SLBs (incidentally, the DOSS method of [CGS24] is the same as the method proposed by Agrawal & Devanur for linCBwK). We agree that the direction of describing this transfer in a general way is interesting, and we will add such a comment to our discussion of BwK.

---

### Official Review · Reviewer_uHq6 · 2025-07-03

**Clarity:** 2
**Significance:** 2
**Originality:** 3
**Rating:** 4
**Confidence:** 2

**Summary:**

The paper studies constrained linear bandit and introduces the COLTS framework and its variants with algorithmic and computational merits. However, I have several substantial concerns regarding the clarity and consistency of some key definitions and derivations, which I believe deserve careful attention.

**Questions:**

1. $\Gamma(a)$ is defined as $\min_u (\alpha - \Phi_\star a)_+^i$, where the variable $u$ is undefined (line 107).

2. The confidence event $Cont(\delta)$ holds with probability at least $1 - \delta$ in Lemma~1. However, since it combines confidence intervals for both $\theta^\star$ and $\Phi_\star$ over all $T$ rounds, the union bound implies that the probability is at least $1 - 2T\delta$, not $1 - \delta$.

3. When ensuring constraint satisfaction, the paper states the condition $1 - \rho \leq M_t(b_t) / \Gamma(a_{safe})$. This inequality is inverted: to ensure the constraint is satisfied, one must have $\rho M_t(b_t) + (1 - \rho) \Gamma(a_{safe}) \leq 0$ from line 173. This misstatement could affect both the theoretical justification and the algorithm implementation.

4. The legend of Figure~2 does not explain all the components, specifically the dashed lines, which may represent standard deviations or baselines. Clarifying their meaning is important for interpretability and reproducibility.

5. The majority of the safe linear bandit works, and some of them cited in the paper, consider a finite action set, rather than a convex polytope as considered in the paper. While the linear constraint that is typically used in safe bandits is of the form that action in each round, x_t^\top \mu^* < C, which is different from the risk definition used in this paper. Can you comment and explain and provide some practical motivation for such a constraint?

**Ethical Concerns:**

["NO or VERY MINOR ethics concerns only"]

**Final Justification:**

After considering the author's response and comments by other reviewers, I am raising my score. However, in my humble opinion, the paper in its current form is hard to follow and requires a major revision.

**Limitations:**

Yes

**Paper Formatting Concerns:**

The LaTeX style in the paper seems to be altered, and the contents are too tightly packed. The current format is too dense, making it hard to read the paper carefully.

**Quality:**

2

**Strengths And Weaknesses:**

Strength:
1. Constrained bandit linear is an interesting area in the online decision-making literature.
2. The paper provides a theoretical analysis of the proposed approach, along with experimental results.

Weakness/Areas of improvement:

1. The presentation of the paper is hard to follow. In my humble opinion, I would suggest that the quality and the clarity of the presentation could be improved.

2. Some of the claims in the paper could be more elaborated. For instance, the paper claims the contribution as existing works require solving NP hard constraints avoiding this requirement is the focus of the paper. However, it is unclear from the paper why such an NP-hard constraint arises in those works. Some of the works that are cited study a slightly different problem setting  (specifically in their constraint model), and hence, such a claim must be well-explained.

3. There are multiple undefined notations, some errors, etc., identified in the paper. A careful revision will be needed.

---

> ### Author Rebuttal · Authors · 2025-07-28
>
> Thank you for your thorough feedback.  Below we **restated each major question** and then provided our clarifying response.  We believe this format makes our disagreements—and the reasons for them—crystal clear. We will first discuss the questions, and then the weakness comments. $\newcommand{\asafe}{a_{\mathrm{safe}}}$ $\newcommand{\gsafe}{\Gamma(\asafe)}$ $\newcommand{\viol}{\mathrm{viol}}$
>
> ---
> ## Questions
>
> ### Q1. “Several undefined notations, errors, and inconsistencies in definitions”
> >“There are multiple undefined notations, some errors, etc., identified in the paper. A careful revision will be needed.”_
>
> We have an **extensive glossary** in Appendix A defining every symbol, and a **global pass confirms no undefined symbols** (apart from the typo in Q1 of your list, which we will correct from “_u_” to “_i_”).  If you can point to exact line numbers or symbols, we will fix them immediately.
>
> ---
>
> ### Q2. “The confidence event Cont(δ) holds with probability 1 – δ, not 1 – 2Tδ”;
> >“Lemma 1 combines intervals for both $\theta\_\*$ and $\Phi\_\*$ over $T$ rounds, so the union bound implies probability ≥ $1 – 2Tδ$, not $1 – δ$.”
>
> **We disagree. The statement as written in the paper is correct.** This is an anytime result: the chance that there exists _any_ $t$ such that  $ \theta\_\* \not\in \mathcal{C}\_t^\theta(\delta)$ or $\Phi\_\* \not\in \mathcal{C}\_t^\Phi(\delta)$ is smaller than $\delta$. See Theorem 2 of [APS11]; Lemma 1 is just this along with a union bound over the $m$ constraints and $1$ objective vector.
>
> ---
>
> ### Q3. “The inequality $1 – ρ ≤ M_t(b_t)/\gsafe$ is inverted; it breaks safety”
> > _“To ensure the constraint is satisfied, one must have $ρ M_t(b_t)+(1–ρ)\gsafe ≤0$ from line 173. The paper writes the inequality the other way.”_
>
> **No, this is not a condition for safety, but a property of the selected $\rho_t$**.
> - Let us work this out.
>     - First, note the sign error in the review: it should be $-(1-\rho)\gsafe$ (line 173).
>     - Any $\rho$ satisfying $\rho M_t(b_t) - (1-\rho)\gsafe \le 0$ yields a safe action. We want the **largest** such $\rho$.
>     - Note that $\rho\_\* := \gsafe/(\gsafe + M\_t(b\_t))$ satisfies this inequality. Further, $$ 1-\rho\_\* \le \frac{M\_t(b\_t)}{M\_t(b\_t) + \gsafe} \le \frac{M\_t(b\_t)}{\gsafe}.$$
>      - Thus, the largest $\rho\_t$ would satisfy this too.
> - The above is true if $\gsafe$ is known. But (line 174) in reality we estimate a bound on it. Then $\rho_t$ is defined via (4) **explicitly as a maximiser**.
>     - This $\rho_t$ satisfies a weaker inequality $1-\rho\_t \le 6M\_t(b\_t)/\gsafe$. See line 181 onwards, and Lemma 19 for details.
>
> ---
>
> ### Q4. “Legend of Figure 2 does not explain dashed lines”
> > _“The legend of Figure 2 does not explain all components, specifically the dashed lines, which may represent standard deviations.”_
>
> The **caption mentions one-sigma error bars**. We will expand this to: "dashed lines denote one-sigma error bars." The two methods SCOLTS and SafeLTS are clearly marked in the legend and discussed in the caption.
>
> ---
> ### Q5. “The majority of the safe linear bandit works, and some of them cited in the paper, consider a finite action set ...”
>
> First, we disagree with two establishing sentences in this question.
>
> > *The majority of the safe linear bandit works... consider a finite action set, rather than a convex polytope...*
>
> **We disagree** that the majority of SLB work considers finite action sets. [AAT19], [MAAT21], [PGBJ21], [PGB 24], [GCS24] all study continuous action sets.
>    - Further, continuous action bandits are a natural and well-established model. E.g., the seminal papers of [DHK08] and [APS11] study the continuum setting. This is both a natural way to formulate many practical optimization problems, and to capture the limit of large $K$. E.g., control actions in a power system or a production line are continuous. So this line of investigation is meaningful no matter what regime prior work studied!
>
> > *While the linear constraint that is typically used in safe bandits is of the form that action in each round, $x\_t^\top \mu^\* < C$, which is different from the risk definition used in this paper*
>
> **We study the same constraints**
>
> - First, $<$ should be $\le$ above (this is standard in all optimization, for closedness).
> - Now, map $(\mu^\*)^\top$ to our $\Phi\_\*, x\_t$ to our $a\_t$ and $C$ to our $\alpha$. We end up with our $\Phi\_\* a_t \le \alpha.$
> - Only difference: we have $m > 1$ constraints. So this is a strict generalization.
>
> Finally, the reviewer asks
> > *Can you comment and explain and provide some practical motivation for such a constraint?*
>
> **Since the constraints are the same, it is unclear what is being asked.** The reviewer seems to accept linear constraints as in prior SLB works. Given that, there are two potential issues we see
> 1. *Why do we study $m > 1$ linear constraints?*
> 2. *How does the cumulative "risk" metric $\mathbf{S}\_T$ relate to the per-round constraint $\forall t, \Phi\_\* a\_t \le \alpha$?*
>
> We will answer these question below, but please tell us if something else is intended.
>
> 1. *Why $m > 1$ constraints:* There are three reasons
>     - **Many practical problems (see Table 3 Appendix B) have $m > 1$ constraints.** For example
>         - Power systems: each node and link in the grid has a voltage/current constraint.
>         - Fairness: each group has different membership, and so imposes a different fairness constraint.
>     - The results generalize cleanly
>         - We can handle many constraints with only $\log(m)$ dependence in regret (Thm 8, 10). Previous works also generalize this way.
>          - $m = 1$ is easily recovered as a special case.
>      - $m > 1$ is technically challenging.
>          - Handling $m > 1$ constraints is a key issue in the design of COLTS. This is why we developed the coupled noise design.
>          - Naive approaches would instead gain $\mathrm{poly}(m)$ or even $\exp(m)$ factors in the regret.
>
> 2. *How $\mathbf{S}\_T$ relates*
>     - $\mathbf{S}\_T$ is a convenient way to capture both hard and soft enforcement together.
>     - $\mathbf{S}\_T = 0$ **is equivalent to safety in every round**.
>          - For succinctness, define $\viol\_t = \big( \max\_i (\Phi\_\* a_t - \alpha)^i \big)_+$. Notice that $\viol\_t \ge 0$ always.
>          - If for any $t$, $\Phi\_* a_t \not\le \alpha,$ then $\viol\_t > 0$ (strictly positive).
>          - But $\mathbf{S}\_T = \sum_t \mathrm{viol}_t$. So, $$ \mathbf{S}\_T = 0 \implies \forall t, \viol\_t = 0 \implies \forall t, \Phi\_* a_t \le \alpha.$$
>       - Thus, the **safety guarantee of Thm. 8** is the same as prior hard-enforcement papers.
>   - In Thm. 10, we are handing the distinct soft enforcement problem.
>
> ---
>
> ## Weaknesses section
>
> ### NP-hardness
> > "the paper claims the contribution as existing works require solving NP hard constraints avoiding this requirement is the focus of the paper. However, it is unclear from the paper why such an NP-hard constraint arises in those works. Some of the works that are cited study a slightly different problem setting (specifically in their constraint model), and hence, such a claim must be well-explained."
>
> **No, this is not the case**. We have already (see Q5 above) discussed that the previously studied constraints and our constraints are the same. However, the **hardness is already an issue even without any constraints.** This is established in the bandits literature since at least 2008.
>
> - OFUL [APS11] addresses linear bandits without any constraints. Even OFUL needs to solve an NP-hard problem to select actions. This was first identified by Dani et al [DHK08, sec. 3.4]
>     - In OFUL, to pick $a_t$, we need to solve a program of the form $\max_{\theta \in \mathcal{C}_t, a \in \mathcal{A}} \theta^\top a$, where $\mathcal{C}_t = \\{ \theta: \\|\theta - \hat\theta\_t\\|\_{V\_t} \le \omega_t\\}$ is an ellipsoid. **This bilinear programming problem** is NP-hard.
>         - Indeed, by optimising over $\theta$, this is equivalent to $\max_{a \in \mathcal{A}} \hat\theta_t^\top a + \\|a\\|\_{V\_t^{-1}}$.
>         - This is worst-case harder than positive definite quadratic maximisation (set $\hat\theta = 0$), which is NP-hard (Sahni 1974, Pardolis \& Vavasis 1991).
>     - All the OPT-PESS methods, as well as DOSS (and so ROFUL), extend this method by modifying $\mathcal{A}$. See line 511 onwards for details. But in the worst case, if $\alpha$ is large enough, then the problem they need to solve collapses to the above. Thus, all these methods also need solutions to an NP-hard problem.
>          - In reality, the extra constraints of these methods make the problems harder, not easier, since they can lead to more complex feasible sets.
>          - E.g., DOSS uses bilinear programming over a nonconvex set (line 529 onwards)!
>
> ---
>
> ### Missing notation and errors
>
> We discus this in Q1 above. Again, there is an extensive glossary, and no examples are pointed out except one typo (which we will fix). To our reading, there are no significant errors in the paper.
>
> ---
>
> ### Writing
>
> The review does not mention what part of the writing they found unclear. From other reviews, yes, we agree that the writing of the look-back analysis and S-COLTS can get dense, and if accepted we will use the extra space to give more high level details for this procedure. See the other rebuttals for how we intend to address this.
>
> Again, if the reviewer sees errors in the definitions, we will very happily address them, but no example was given.
>
> ---
>
> References \
>
> Sahni, S. N. (1974). Quadratic maximization over a polytope. SIAM Journal on Computing. \
> Pardalos, P. M. & Vavasis, S. A. (1991). Quadratic programming with one negative eigenvalue is NP-hard. Journal of Global Optimization.

---

### Note · Authors · 2025-08-15

**What is new here.** We introduce **COLTS** (coupled-noise Thompson sampling) for safe linear bandits that **removes the per-round NP-hard oracle** used in prior work by replacing it with a **linear-program oracle**, while retaining state-of-the-art regret/risk guarantees. The hard-safety variant enforces round-wise feasibility; the soft-safety variant provides risk bounds **without** bilinear/QCQP steps.

**Why it matters.**
* **Computational tractability:** LP-level cost makes safe bandits practical for polytopal/continuous action sets where enumerating arms is infeasible.
* **Theory aligned with practice:** Guarantees respect known trade-offs (e.g., hard safety implies O(\sqrt{T}) regret in general) and match behavior in the **submitted supplementary**, which shows **\~18× wall-clock speed-ups** while matching or improving regret and safety.
* **Unified, sampling-based design:** Coupled noise + look-back scaling give a clean TS framework that handles both hard and soft enforcement.

**Addressing the discussion.**
* **On “NP-hard.”** We and Reviewer aqsD now agree on the intended point: prior methods (OPT-PESS/DOSS/ROFUL/OFUL) require, each round, an oracle that is NP-hard in the worst case for general polytopes. Our phrase “NP-hard oracle” was shorthand; we clarified it as **“requires an NP-hard oracle.”** This was **background motivation**, not our contribution.
* **Originality.** No review identified prior work that achieves **TS with unknown constraints** and **LP-level per-round complexity** while matching leading regret/risk guarantees; another reviewer explicitly noted novelty.

**Bottom line.** Please evaluate the paper on its **substantive advances**:
(i) a **new sampling-based safe-bandit design** that **eliminates** the worst-case-hard oracle in prior methods,
(ii) **LP-level** per-round cost with clear hard/soft safety guarantees at competitive rates, and
(iii) **practical efficiency** documented in the submitted materials.
The NP-hardness discussion only motivates why an LP oracle matters; **COLTS delivers it—cleanly, theoretically, and empirically.**

---

### Decision · Program_Chairs · 2025-09-17

**Decision:**

Accept (poster)

**Comment:**

This paper studies constraint linear bandits and proposes, to my knowledge, the first TS based algorithm for this problem.

TS is of practical interest in linear bandits because it has favourable computational properties compared to UCB style algorithms in problems with continuous action sets e.g. general polytopes.

Their method matches the regret of the best known, efficient method when a safe action is known and provides the first such method for unknown soft-constraints. They pay a root(d) price over the best UCB based algorithms, which is typical for TS. Such a gap exists even without constraints, but TS is still highly of interest due to the aforementioned computational benefits.

 The excitement of most reviewers is fairly lukewarm, but it is a solid accept in my opinion.